# Zeroth-Order Non-Log-Concave Sampling with Variance Reduction and Applications to Inverse Problems

**M. Berk Sahin** [1]  **Behzad Sharif** [1 2]  **Abolfazl Hashemi** [1]

## Abstract

Sampling from high-dimensional, non-log-concave distributions with unnormalized densities remains a fundamental challenge in machine learning, particularly in black-box settings where gradient information is inaccessible or computationally prohibitive. While Langevin dynamics provides a principled framework for sampling when gradients are accessible, its extension to the black-box settings suffers from high variance and lacks non-asymptotic convergence guarantees for non-log-concave sampling. To address these limitations, we propose a variance-reduced zeroth-order Langevin sampling method. Our method employs a gradient estimator that substantially reduces the variance of the classical batched zeroth-order estimator and eliminates the unfavorable dimensional dependence of the batch size required for accurate estimation, enabling practical and stable sampling. We establish the first non-asymptotic convergence guarantees for zeroth-order non-log-concave sampling in terms of $\varepsilon$-relative Fisher information, and, under a Poincaré inequality assumption, squared total variation distance. We further propose ZO-APMC, a posterior sampling algorithm for black-box inverse problems with pre-trained score-based generative priors, establishing the first non-asymptotic convergence guarantees for such methods. We validate our theory through synthetic experiments and demonstrate strong empirical performance on practical linear and nonlinear inverse problems.

[1]Elmore School of Electrical and Computer Engineering, Purdue University, West Lafayette, USA [2]Weldon School of Biomedical Engineering, Purdue University, West Lafayette, USA. Correspondence to: M. Berk Sahin <sahinm@purdue.edu>.

*Proceedings of the $43^{rd}$ International Conference on Machine Learning*, Seoul, South Korea. PMLR 306, 2026. Copyright 2026 by the author(s).

## 1. Introduction

We study the problem of sampling from a distribution $\pi \propto \exp(-f)$ with potential function $f : \mathbb{R}^d \to \mathbb{R}$ when we only have access to zeroth-order (ZO) evaluations of the potential function. This problem is of fundamental importance when gradients are unavailable or prohibitively expensive, and has been investigated by several recent works (Liu & Wang, 2020; Roy et al., 2022; He et al., 2024). In the special case where $f$ is strongly log-concave and has Lipschitz continuous gradient, this problem is well understood; for instance, Roy et al. (2022) establish non-asymptotic complexity bounds for *Langevin Monte Carlo (LMC)* sampling. In contrast, to the best of our knowledge, the non-log-concave setting remains largely unexplored. Our **first main contribution** is to take an initial step toward a theory of non-log-concave ZO sampling by proposing a novel ZO estimator that enables sampling from non-log-concave distributions. The main challenge is that standard ZO estimators based on finite-difference evaluations along random Gaussian directions exhibit high variance. Controlling this variance typically requires batch size scaling as $\mathcal{O}(d)$ with the dimension $d$ of the sampling variable, leading to substantial function evaluations and memory costs in high-dimensional settings. To mitigate this issue, we propose a novel variance-reduced ZO estimator that uses only $\mathcal{O}(1)$ number of function evaluations per iteration, making the batch size independent of the ambient dimension and substantially reducing the associated memory costs. Our theoretical analysis builds on the LMC framework with gradient access developed by Balasubramanian et al. (2022), and is based on a sampling analogue of *stationary point analysis*, a technique that has shown to be highly effective in non-convex optimization (Nesterov et al., 2018).

Beyond the foundational sampling setting, we further extend our method and theoretical analysis to posterior sampling for solving ill-posed inverse problems using score-based generative model (SGM) priors in black-box settings, where privileged information about the forward model such as its derivative, pseudo-inverse (Song et al., 2023), or its parametrization (Chung et al., 2023a) is unavailable or computationally prohibitive. Such scenarios arise in a wide range of applications: the forward operator may be de-

fined through large PDE-based simulators whose derivatives or pseudo-inverses are typically inaccessible or undefined (Evensen & Van Leeuwen, 1996; Oliver et al., 2008; Iglesias et al., 2013); simulators may rely on legacy code that cannot be adapted to modern auto-differentiation frameworks (Harbaugh et al., 2000); the forward model may be a proprietary system (closed-source), as in commercial MRI scanners (Karakuzu et al., 2025); the underlying physics may involve discontinuities (Moës et al., 1999; Tan et al., 2018; Lopez-Gomez et al., 2022); or the forward model may be implemented as a rule-based expert system (Rotshtein & Rakytyanska, 2012; Huang et al., 2024; Gong et al., 2025).

In such ill-posed inverse problem settings, the choice of a flexible and expressive prior is crucial. SGMs have recently emerged as powerful *plug-and-play* priors due to their ability to model high-dimensional non-log-concave data distributions, while remaining applicable across a wide range of inverse problems without re-training. They have demonstrated a strong empirical performance across a wide range of applications, including image restoration (Wang et al., 2023; Rout et al., 2023), medical imaging (Song et al., 2022; Sun et al., 2024), and music generation (Rout et al., 2025), where access to gradients of the forward operator is typically assumed. More recently, SGMs have been adopted to develop black-box posterior sampling methods for inverse problems where access to the likelihood score is unavailable, showing promising empirical performance (Tang et al., 2024; Huang et al., 2024; Zheng et al., 2025a). However, these approaches rely on heuristic approximations and currently lack rigorous convergence guarantees to the target posterior under standard probabilistic discrepancy measures such as Fisher information (FI) or total variation (TV) distance. In fact, rigorous guarantees remain rare even for posterior sampling methods with access to the likelihood score; when available, they typically rely on restrictive assumptions such as linear forward operators, which are often violated in practice (Daras et al., 2024). A more detailed discussion of both gradient-based and black-box posterior sampling methods is provided in Appendix A, along with a complementary comparison to the proposed approach in Table 3.

The **second main contribution** of this work is the development of a theoretically grounded *plug-and-play Monte Carlo (PMC)* method for solving black-box inverse problems using only forward model evaluations and a pre-trained SGM prior. We position this as an important step toward posterior sampling in black-box settings that offers an algorithm with formal convergence guarantees and a solid foundation for future advances. We encounter two key challenges in designing a practical ZO posterior sampling algorithm: standard LMC often converge slowly, and ZO estimates require batch sizes that scale with the problem dimension, leading to prohibitive computational and memory costs in high-dimensional settings. To address these challenges, we combine annealed LMC with our variance-reduced ZO estimator, enabling accurate forward model gradient approximation using a practical number of function evaluations at each iteration. We incorporate the effect of annealing schedule and SGM estimation error on convergence in our theoretical results for posterior sampling.

Specifically, the key contributions of this work are the following:

- We establish the first non-asymptotic complexity guarantees for ZO sampling, achieving an $\varepsilon$-relative FI error after $\mathcal{O}(1/\varepsilon^4)$ iterations. Under a Poincaré inequality assumption on the target distribution, this rate also yields $\varepsilon$-accuracy in squared TV distance. Furthermore, with decaying parameters, we show weak convergence to the target distribution.

- We propose a novel variance-reduced ZO gradient estimator that achieves the convergence guarantees above using only $\mathcal{O}(1)$ function evaluations per iteration, eliminating the $\mathcal{O}(d)$ batch-size scaling of standard ZO estimator.

- We propose a variance-reduced ZO annealed PMC algorithm (ZO-APMC) for black-box inverse problems, based on annealed LMC and a pre-trained SGM prior, and establish non-asymptotic and weak convergence guarantees.

- We verify our theoretical findings with numerical and statistical experiments, and further demonstrate that ZO-APMC consistently outperforms existing black-box posterior sampling methods in MRI reconstruction and black hole imaging, while delivering competitive performance on the Navier–Stokes inverse problem.

## 2. Preliminaries

### 2.1. Zeroth-Order Sampling

Traditionally, the Langevin diffusion is defined as the solution to the stochastic differential equation

$$d\boldsymbol{x}_t = -\nabla f(\boldsymbol{x}_t)dt + \sqrt{2}d\boldsymbol{B}_t, \quad (1)$$

has $\pi \propto \exp(-f)$ as its unique stationary distribution and converges to it as $t \to \infty$ under mild conditions. Here, $(\boldsymbol{B}_t)_{t\geq 0}$ denotes a standard $d$-dimensional Brownian motion. Discretizing this stochastic process with step size $\gamma > 0$ yields the standard Langevin Monte Carlo (LMC) algorithm.

$$\boldsymbol{x}_{(k+1)\gamma} := \boldsymbol{x}_{k\gamma} - \gamma\nabla f(\boldsymbol{x}_{k\gamma}) + \sqrt{2}(\boldsymbol{B}_{(k+1)\gamma} - \boldsymbol{B}_{k\gamma}), \quad (2)$$

In this work, we assume black-box access to the potential function $f$; therefore, the gradient $\nabla f(\boldsymbol{x}_{k\gamma})$ cannot be computed. Instead, we consider its ZO estimation (Nesterov &

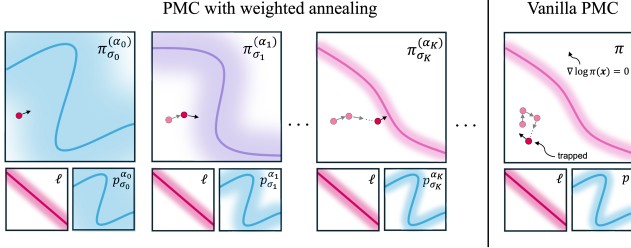

*Figure 1.* Illustration of weighted annealing in PMC through weighted posteriors $\big(\pi_{\sigma_k}^{(\alpha_k)}\big)_{k=0}^{N-1}$. Solid lines and shaded regions denote the distribution mean and density, respectively, while unshaded regions correspond to $\nabla \log \pi(\boldsymbol{x}) = 0$. By gradually reducing the prior smoothing parameter $\sigma_k$ and its weight $\alpha_k$ relative to the likelihood $\ell$, weighted annealing enables PMC to escape plateaus in $\nabla \log \pi(\boldsymbol{x})$.

Spokoiny, 2017), defined as

$$\widetilde{\nabla} f_\mu(\boldsymbol{x}, \boldsymbol{u}) := \frac{f(\boldsymbol{x} + \mu \boldsymbol{u}) - f(\boldsymbol{x})}{\mu}\, \boldsymbol{u}, \qquad (3)$$

where $\boldsymbol{u} \sim \mathcal{N}(0, I)$ and $\mu > 0$ is the smoothing parameter. We note that the ZO estimator is biased, since $\nabla f_\mu(\boldsymbol{x}) := \mathbb{E}_{\boldsymbol{u}}[\widetilde{\nabla} f_\mu(\boldsymbol{x}, \boldsymbol{u})] \neq \nabla f(\boldsymbol{x})$. However, this bias vanishes as $\mu \to 0$; see Lemma 1 in Appendix B.1. Replacing $\nabla f(\boldsymbol{x}_{k\gamma})$ in (2) with its ZO estimate $(1/b)\sum_{i=1}^b \widetilde{\nabla} f_\mu(\boldsymbol{x}_{k\gamma}, \boldsymbol{u}_k^i)$ yields a naive ZO-LMC algorithm, where $b$ is the batch size. Roy et al. (2022) analyze this approach for strongly log-concave target distributions, an assumption often violated in practice, and establish Wasserstein-2 convergence guarantees. However, their analysis requires the batch size to scale as $\mathcal{O}(d)$, resulting in prohibitively large memory requirements in high-dimensional settings. In this work, we instead provide an analysis for non-log-concave target distributions and use $\mathcal{O}(1)$ function evaluations per iteration. This property is ensured by the proposed variance-reduced ZO estimator $\boldsymbol{g}_k$, defined in Section 3.

## 2.2. Black-Box Inverse Problems

For the posterior sampling part of our work, we consider a general black-box inverse problem setting modeled as

$$\boldsymbol{y} = \boldsymbol{A}(\boldsymbol{x}) + \xi, \quad \boldsymbol{x} \in \mathbb{R}^d, \quad \boldsymbol{y}, \xi \in \mathbb{R}^m. \qquad (4)$$

where we assume only black-box access to the forward model $\boldsymbol{A}(\cdot)$. In this setting, gradient information is unavailable, and $\boldsymbol{A}$ can only be queried through input–output evaluations. The operator $\boldsymbol{A} : \mathbb{R}^d \to \mathbb{R}^m$ models the response of the imaging system, where $m \ll d$, and $\xi \in \mathbb{R}^m$ represents measurement noise. The objective is to recover the unknown signal $\boldsymbol{x}$ from the noisy measurements $\boldsymbol{y}$. In many practical settings, the mapping $\boldsymbol{x} \to \boldsymbol{y}$ is many-to-one, making the reconstruction task an ill-posed inverse problem, since $\boldsymbol{x}$ cannot be uniquely recovered from $\boldsymbol{y}$. In Bayesian framework, one could introduce $p(\boldsymbol{x}) \propto \exp(-h)$ as the *prior*,

and samples from the *posterior* $\pi(\boldsymbol{x}|\boldsymbol{y})$, which is formally established with the Bayes' rule: $\pi(\boldsymbol{x}|\boldsymbol{y}) \propto \ell(\boldsymbol{y}|\boldsymbol{x})p(\boldsymbol{x})$, where $\ell(\boldsymbol{y}|\boldsymbol{x}) \propto \exp(-f)$ is the *likelihood* distribution induced by (4). Thus, by applying Bayes' rule to the LMC iterates in (2) and replacing $\nabla f(\boldsymbol{x}_{k\gamma})$ with its naive ZO estimate, we readily obtain the following ZO posterior sampling LMC algorithm.

$$\boldsymbol{x}_{(k+1)\gamma} := \boldsymbol{x}_{k\gamma} - \gamma \bigg( \frac{1}{b} \sum_{i=1}^b \widetilde{\nabla} f_\mu(\boldsymbol{x}_{k\gamma}, \boldsymbol{u}_k^i) + \nabla h(\boldsymbol{x}_{k\gamma}) \bigg)$$

$$+ \sqrt{2}(\boldsymbol{B}_{(k+1)\gamma} - \boldsymbol{B}_{k\gamma}), \qquad (5)$$

where $f$ and $h$ denote potential functions of the likelihood and the prior, respectively. While $\widetilde{\nabla} f_\mu(\boldsymbol{x}_{k\gamma}, \boldsymbol{u}_k^i)$ can be computed via forward model function evaluations under the assumption of Gaussian measurement noise in (4), the prior score $-\nabla h(\boldsymbol{x}_{k\gamma})$ is unknown.

SGMs have been proposed as powerful models for high-dimensional non-log-concave priors. At their core, they learn the perturbed score function $-\nabla h_\sigma(\boldsymbol{x}) := \nabla \log p_\sigma(\boldsymbol{x})$, where $p_\sigma(\boldsymbol{x}) := \int_{\mathbb{R}^d} p(\boldsymbol{z}) \mathcal{N}(\boldsymbol{x}|\boldsymbol{z}, \sigma^2 I)\, d\boldsymbol{z}$ with a small perturbation $\sigma > 0$. This score is learned by a deep neural network using *score matching objective* (Hyvärinen & Dayan, 2005; Vincent, 2011) and can be estimated via *Tweedie's* formula (Robbins, 1992; Miyasawa et al., 1961; Efron, 2011). We denote the SGM estimator by $\mathcal{S}_\theta(\boldsymbol{x}, \sigma) \approx -\nabla h_\sigma(\boldsymbol{x})$, where $\mathcal{S}_\theta$ is conditioned on the noise level $\sigma$ and parametrized by $\theta$. In practice, LMC algorithms suffer from slow convergence and mode collapse when sampling from high-dimensional multimodal distributions. To alleviate this, SGMs are trained across multiple noise scales $(\sigma_k)_{k=0}^{N-1}$ where larger noise levels produce smoother approximations of the data distributions, making score estimation and sampling well-posed. Replacing $-\nabla h(\boldsymbol{x}_{k\gamma})$ in (5) by SGM prior, we obtain a naive ZO annealed LMC for posterior sampling.

$$\boldsymbol{x}_{(k+1)\gamma} := \boldsymbol{x}_{k\gamma} - \gamma \bigg( \frac{1}{b} \sum_{i=1}^b \widetilde{\nabla} f_\mu(\boldsymbol{x}_{k\gamma}, \boldsymbol{u}_k^i) \qquad (6)$$

$$- \alpha_k \mathcal{S}_\theta(\boldsymbol{x}_{k\gamma}, \sigma_k) \bigg) + \sqrt{2}(\boldsymbol{B}_{(k+1)\gamma} - \boldsymbol{B}_{k\gamma}),$$

where $(\alpha_k)_{k=0}^{N-1}$ denotes a weighted annealing schedule. This corresponds to sampling from a sequence of weighted posterior distributions $\pi_{\sigma_k}^{(\alpha_k)}(\boldsymbol{x}|\boldsymbol{y}) \propto \ell(\boldsymbol{y}|\boldsymbol{x})p_{\sigma_k}^{\alpha_k}(\boldsymbol{x})$. Initially, the posterior is dominated by the smoothed prior $p_{\sigma_k}$, and its less concentrated density facilitates escape from low-probability noisy regions of $\nabla h$. As the iterations progress, the likelihood contributes more strongly while the smoothed prior $p_{\sigma_k}$ approaches the true prior $p$, guiding the iterates toward the target posterior. An illustration of this process is provided in Fig. 1. In practice, both the noise scales

$(\sigma_k)_{k=0}^{N-1}$ and annealing weights $(\alpha_k)_{k=0}^{N-1}$ decrease over the iterations until reaching prescribed minimum values, after which they remain fixed (Song & Ermon, 2019; 2020; Sun et al., 2024). The precise definitions of these schedules are provided in (13).

## 2.3. Sampling Analogue of Stationary Point Analysis

Before introducing our proposed variance-reduced ZO sampling method with its convergence guarantees, we explain why convergence in relative FI, defined as $\mathrm{FI}(\nu\|\pi) := \int_{\mathbb{R}^n} \nu(x)\|\nabla \log \nu(x) - \nabla \log \pi(x)\|_2^2 \, dx$, serves as the sampling analogue of stationary point analysis in non-convex optimization. Consider the minimization of the *Kullback–Leibler (KL)* divergence over the Wasserstein space of probability distributions.

$$\hat{\nu} = \arg\min_{\nu} \mathrm{KL}(\nu\|\pi), \tag{7}$$

where $\mathrm{KL}(\nu\|\pi) := \int_{\mathbb{R}^d} \nu(\boldsymbol{x}) \log \frac{\nu(\boldsymbol{x})}{\pi(\boldsymbol{x})} d\boldsymbol{x}$, and $\nu$ and $\pi$ denote the estimate and target distributions, respectively. Similar to the gradient concept in Euclidean space, the Wasserstein gradient of $\mathrm{KL}(\cdot\|\pi)$ at $\nu$ is $\nabla \log(\nu/\pi)$ (Ambrosio et al., 2008) and its expected square norm gives us $\mathrm{FI}(\nu\|\pi)$. If $\nu_t$ evolves under Langevin diffusion in (1), then $\frac{d}{dt}\mathrm{KL}(\nu_t\|\pi) = -\mathrm{FI}(\nu_t\|\pi)$ (Ambrosio et al., 2008; Villani, 2009), showing that Langevin diffusion is a gradient flow in probability space. From an optimization perspective, $\mathrm{FI}(\nu_t\|\pi)$ plays a role analogous to the squared gradient norm in non-convex optimization. Unlike optimization, however, the condition $\mathrm{FI}(\nu\|\pi) = 0$ implies $\nu = \pi$, making FI a natural discrepancy measure for quantifying convergence to the target distribution. Accordingly, following Balasubramanian et al. (2022), we characterize convergence by requiring the output distribution $\nu$ to satisfy $\mathrm{FI}(\nu\|\pi) \leq \varepsilon$, analogous to reaching an $\varepsilon$-stationary point in optimization.

## 3. Methods

In this section, we propose the variance-reduced ZO gradient estimator and present theoretical guarantees for sampling from non-log-concave distributions. We then extend our analysis to posterior sampling with SGM priors, accounting for both the bias due to the annealing schedules and the SGM estimation error.

### 3.1. Variance-Reduced Zeroth-Order Sampling

As mentioned previously, the naive ZO estimate requires batch size of $\mathcal{O}(d)$ to make an accurate estimate of a gradient, which is prohibitive to calculate at each iteration. Using small batch sizes $b \ll d$, while computationally attractive, leads to noisy estimates. Thus, inspired by Li et al. (2021), we combine the large- and small-batch estimators to define

the following variance-reduced ZO estimator:

$$\boldsymbol{g}_k := \begin{cases} \dfrac{1}{b}\sum_{i=1}^{b} \widetilde{\nabla} f_\mu(\boldsymbol{x}_{k\gamma}, \boldsymbol{u}_k^i), & \text{w.p. } p, \\[2ex] \boldsymbol{g}_{k-1} + \dfrac{1}{b'}\sum_{i=1}^{b'} \Big( \widetilde{\nabla} f_\mu(\boldsymbol{x}_{k\gamma}, \boldsymbol{u}_k^i) & \text{w.p. } 1-p, \\[1ex] \qquad\qquad - \widetilde{\nabla} f_\mu(\boldsymbol{x}_{(k-1)\gamma}, \boldsymbol{u}_k^i) \Big), \end{cases} \tag{8}$$

where $k \geq 1$ is the iteration index, and $b, b' \geq 1$ denote batch sizes, with $b$ a large batch size computed with probability $p \in (0, 1]$ and $b' \ll b$ a much smaller one. For the initial step $k = 0$, we use the batch estimate $\boldsymbol{g}_0 := \frac{1}{b}\sum_{i=1}^{b} \widetilde{\nabla} f_\mu(\boldsymbol{x}_0, \boldsymbol{u}_0^i)$. The key motivation behind this construction is to mitigate the high computational cost of ZO estimation, which would otherwise require a batch size on the order of $\mathcal{O}(d)$ at every iteration. Instead, a large batch estimator is computed only intermittently, with probability $p < 1$. With the remaining probability $1 - p$, the estimation is updated recursively using a small batch of fresh function evaluations together with the previous estimate. More specifically, the update approximates the change in the gradient between consecutive iterates, exploiting the strong correlation between $\boldsymbol{x}_{(k-1)\gamma}$ and $\boldsymbol{x}_{k\gamma}$. Under a suitable regularity condition on $f$, this gradient variation can be accurately estimated using only a small batch size $b' \ll b$, thereby substantially reducing the per-iteration computational cost. We formalize this condition in the following assumption.

**Assumption 1.** The potential function $f : \mathbb{R}^d \to \mathbb{R}$ is $L_1$-Lipschitz continuous: $\|f(\boldsymbol{x}_1) - f(\boldsymbol{x}_2)\|_2 \leq L_1\|\boldsymbol{x}_1 - \boldsymbol{x}_2\|_2$, for all $\boldsymbol{x}_1, \boldsymbol{x}_2 \in \mathbb{R}^d$ and for some $L_1 > 0$.

Assumption 1 is restrictive since it does not hold globally even for simple distributions such as Gaussians. However, it is satisfied by differentiable potentials on compact domains, which commonly arise in practice through normalization or gradient clipping. In the optimization literature, variance-reduction techniques achieving $\mathcal{O}(1)$ per-iteration cost typically rely on Lipschitz continuity condition on stochastic gradients to control their variance (Cutkosky & Orabona, 2019; Li et al., 2021; Liu et al., 2025). Since gradients are unavailable in our setting and must instead be approximated using ZO estimators, we impose Lipschitz continuity on the potential function $f$ to obtain analogous variance-control properties.

Using the proposed estimator, we obtain the following variance-reduced ZO-LMC algorithm

$$\boldsymbol{x}_{(k+1)\gamma} := \boldsymbol{x}_{k\gamma} - \gamma\boldsymbol{g}_k + \sqrt{2}(\boldsymbol{B}_{(k+1)\gamma} - \boldsymbol{B}_{k\gamma}). \tag{9}$$

We state our main results for the following continuous time interpolation of LMC, defined for $t \in [k\gamma, (k+1)\gamma]$:

$$\boldsymbol{x}_t := \boldsymbol{x}_{k\gamma} - (t - k\gamma)\boldsymbol{g}_k + \sqrt{2}(\boldsymbol{B}_t - \boldsymbol{B}_{k\gamma}), \tag{10}$$

and we write $\nu_t$ for the law of $\boldsymbol{x}_t$. Before presenting our main results, we first provide intuition for the proposed variance-reduction mechanism through the following bound on the estimation error.

**Proposition 1.** *Suppose Assumption 1 holds, and let* $(\boldsymbol{x}_{k\gamma})_{k\geq 0}$ *be generated by (9), where* $\boldsymbol{g}_k$ *is given by (8). Define* $e_k^2 := \mathbb{E}[\|\boldsymbol{g}_k - \nabla f_\mu(\boldsymbol{x}_{k\gamma})\|_2^2]$. *Then, for any* $\gamma > 0$,

$$e_{k+1}^2 \leq \frac{p\sigma_{k+1}^2}{b} + (1-p)e_k^2 + \frac{4(1-p)dL_1^2\Delta_k}{\mu^2 b'}, \quad (11)$$

*where* $\sigma_{k+1}^2 := \mathbb{E}[\|\widetilde{\nabla} f_\mu(\boldsymbol{x}_{(k+1)\gamma}, \boldsymbol{u}_{k+1}^i)\|^2]$ *and* $\Delta_k := \mathbb{E}[\|\boldsymbol{x}_{(k+1)\gamma} - \boldsymbol{x}_{k\gamma}\|^2]$.

Proposition 1 (proof provided in Appendix B.2) gives an intuition for the variance-reduction mechanism. When $p = 1$, the estimation error $e_{k+1}^2$ is controlled solely by the variance term $\sigma_{k+1}^2/b$, which can only be reduced by increasing the batch size $b$. In contrast, when $p < 1$, the variance term can be reduced by $p$ while keeping $b$ fixed, at the cost of introducing an error-propagation term $(1-p)e_k^2$ together with an additional term reflecting how similar consecutive iterates remain. Consequently, smaller values of $p$ reduce the variance term but increase error propagation, leading to a trade-off between variance reduction and convergence speed. Unlike stochastic optimization (Li et al., 2021), the additional error term depends on the discretization error of the Langevin diffusion and is amplified by the ZO smoothing parameter $\mu$, resulting in an additional trade-off between ZO estimation bias and discretization error. As shown in our main results, these trade-offs can be controlled through suitable choices of the parameters $\mu$, $\gamma$, and $p$. Furthermore, controlling the discretization error requires the following assumption, which is standard in LMC analysis.

**Assumption 2.** The gradient of $f$ is $L_2$-Lipschitz continuous: $\|\nabla f(\boldsymbol{x}_1) - \nabla f(\boldsymbol{x}_2)\|_2 \leq L_2\|\boldsymbol{x}_1 - \boldsymbol{x}_2\|_2$, for all $\boldsymbol{x}_1, \boldsymbol{x}_2 \in \mathbb{R}^d$ and for some $L_2 > 0$.

We are now ready to state our first main result.

**Theorem 1.** *Let* $\pi \propto \exp(-f)$ *be the target distribution, where the potential* $f$ *satisfies Assumptions 1 and 2 with Lipschitz constants* $L_1$ *and* $L_2$, *respectively. Define* $L_m := \max\{L_1, L_2\}$. *Let* $N \geq 1$ *denote the total number of iterations, and let* $(\nu_t)_{t\geq 0}$ *denote the law of the continuous-time interpolation (10) generated by step size* $\gamma = L_m^{-1}N^{-3/4}d^{-7/4}$, *probability* $p = L_m N^{-1/4}d^{-1/4}$, *batch size* $b = \lceil p^{-1} \rceil$, *and smoothing parameter of ZO estimates* $\mu = L_m^{-1/2}N^{-1/8}d^{-5/8}$. *Then, the time-averaged law* $\bar{\nu}_{N\gamma} := \frac{1}{N\gamma}\int_0^{N\gamma} \nu_t \, dt$ *satisfies* $\text{FI}(\bar{\nu}_{N\gamma}\|\pi) \leq \varepsilon$ *after* $\mathcal{O}(d^7 L_m^4/\varepsilon^4)$ *iterations with* $\mathcal{O}(1)$ *function evaluations per iteration.*

*Remark* 1. In practice, sampling from $\bar{\nu}_{N\gamma}$ can be carried out as follows. First, draw $t \in [0, N\gamma]$ uniformly at random

and determine the largest integer $k$ such that $k\gamma \leq t$. Then, perform a linear interpolation over the interval $[k\gamma, t]$ by using (10) to obtain $\boldsymbol{x}_t$. The resulting sample follows the distribution $\bar{\nu}_{N\gamma}$.

Theorem 1 (the full statement and the proof are provided in Appendix B.3) shows that the proposed variance-reduced ZO-LMC converges with only $\mathcal{O}(1)$ function evaluations per iteration, substantially reducing the memory and per-iteration computational costs compared to naive ZO-LMC in high-dimensional settings. The high polynomial dependence on $d$ in number of iterations arises from the interaction between the ZO approximation error and the Langevin discretization error, which is captured by the last term in the upper bound of Proposition 1 and reflects the intrinsic difficulty of sampling from non-log-concave distributions using ZO information. In practice, however, our experiments show that accurate samples can be obtained using significantly fewer iterations.

We next present a stronger convergence guarantee under the following Poincaré inequality assumption.

**Assumption 3.** For every smooth, compactly supported function $\phi : \mathbb{R}^d \to \mathbb{R}$, the target distribution $\pi \propto \exp(-f)$ satisfies the Poincaré inequality: $\text{Var}_\pi(\phi) \leq C_{\text{PI}}\mathbb{E}_\pi[\|\nabla\phi\|_2^2]$ for some $C_{\text{PI}} > 0$.

Assumption 3 enforces concentration by requiring sufficiently growth of the potential at infinity, thereby preventing heavy tails, and is analogous to Polyak-Łojasiewicz inequality in optimization. This assumption holds for a broad class of distributions, including log-concave distributions and certain non-log-concave cases such as Gaussian convolutions of bounded-support distributions (Chewi et al., 2024). Combining the Poincaré inequality with Theorem 1, we obtain the following convergence guarantee in squared TV distance.

**Corollary 1.** *Let* $\pi \propto \exp(-f)$ *be the target distribution, where the potential* $f$ *satisfies Assumptions 1, 2 with Lipschitz constants* $L_1$, $L_2$, *respectively, and Assumption 3 with constant* $C_{\text{PI}}$. *Define* $L_m := \max\{L_1, L_2\}$. *Let* $N \geq 1$ *denote the total number of iterations, and let* $(\nu_t)_{t\geq 0}$ *denote the law of the continuous-time interpolation (10) generated by the same step size* $\gamma$, *probability* $p$, *batch size* $b$, *and smoothing parameter of ZO estimates* $\mu$ *as in Theorem 1. Then, the time-averaged law* $\bar{\nu}_{N\gamma} := \frac{1}{N\gamma}\int_0^{N\gamma} \nu_t \, dt$ *satisfies* $\|\bar{\nu}_{N\gamma} - \pi\|_{\text{TV}}^2 \leq \varepsilon$ *after* $\mathcal{O}(d^7 L_m^4 C_{\text{PI}}^4/\varepsilon^4)$ *iterations with* $\mathcal{O}(1)$ *function evaluations per iteration.*

The complete statement and the proof are provided in Appendix B.4. Building on Theorem 1 and the fact that $\text{FI}(\nu\|\pi) = 0$ implies $\nu = \pi$, we obtain the following result with its formal statement and proof provided in Appendix B.5.

**Theorem 2.** *Let $\pi \propto \exp(-f)$ be the target distribution, where the potential $f$ satisfies Assumptions 1 and 2 with Lipschitz constants $L_1$ and $L_2$, respectively. Define $L_m :=$ $\max\{L_1, L_2\}$. Let $(\nu_t)_{t\geq 0}$ denote the law of continuous-time interpolation (10) generated by time-varying step size $\gamma_k = \frac{C_\gamma}{k^{3/2}}$, probability $p_k = \frac{1}{2k^{1/2}}$, batch size $b_k = \lceil p_k^{-1} \rceil$, and smoothing parameter of ZO estimates $\mu_k = \frac{C_\mu}{k^{1/8}}$ for $k \geq 1$, where $C_\gamma, C_\mu > 0$ are constants. Then, the time-averaged law $\bar{\nu}_{\tau_n} := \frac{1}{\tau_n} \int_0^{\tau_n} \nu_t \, dt$, where $\tau_n := \sum_{k=1}^{n} \gamma_k$, converges weakly to $\pi$.*

### 3.2. Zeroth-Order Posterior Sampling with SGM Prior

In this section, we propose ZO-APMC algorithm for solving black-box inverse problems with an SGM prior and extend the theoretical analysis of the previous section to this setting by incorporating the bias due to the annealing schedules and the SGM estimation error. We replace the naive ZO gradient estimator in (6) with the proposed variance reduction mechanism in (8) yielding the following ZO-APMC iterates:

$$\boldsymbol{x}_{(k+1)\gamma} := \boldsymbol{x}_{k\gamma} - \gamma(\boldsymbol{g}_k - \alpha_k \mathcal{S}_\theta(\boldsymbol{x}_{k\gamma}, \sigma_k)) \qquad (12)$$
$$+ \sqrt{2}(\boldsymbol{B}_{(k+1)\gamma} - \boldsymbol{B}_{k\gamma}),$$

where $(\alpha_k)_{k=0}^{N-1}$ and $(\sigma_k)_{k=0}^{N-1}$ are annealing and noise schedules, respectively. Following Song & Ermon (2020) and Sun et al. (2024), we define, for all $k \geq 0$,

$$\alpha_k := \max\{\alpha_0 \rho_1^k, 1\} \text{ and } \sigma_k := \max\{\sigma_0 \rho_2^k, \sigma_{\min}\}, \quad (13)$$

where $\rho_1, \rho_2 \in (0, 1)$ denote decay rates, $\sigma_0 \geq \sigma_{\min}$ and $\alpha_0 \geq 1$ are initial values, and $\sigma_{\min} > 0$ is the minimum noise level. These parameters are selected using the principled techniques of Song & Ermon (2020) such that there exist indices $K_\alpha, K_\sigma < N - 1$ satisfying $\alpha_k = 1, \forall k \geq K_\alpha$ and $\sigma_k = \sigma_{\min}, \forall k \geq K_\sigma$. Our analysis relies on these properties together with a continuous-time interpolation of the ZO-APMC iterates incorporating the annealing and noise schedules. For $t \in [k\gamma, (k+1)\gamma]$, the interpolation is defined by

$$\boldsymbol{x}_t := \boldsymbol{x}_{k\gamma} - (t - k\gamma)(\boldsymbol{g}_k - \alpha_k \mathcal{S}_\theta(\boldsymbol{x}_{k\gamma}, \sigma_k)) \qquad (14)$$
$$+ \sqrt{2}(\boldsymbol{B}_t - \boldsymbol{B}_{k\gamma}).$$

Here, $\mathcal{S}_\theta(\boldsymbol{x}_{k\gamma}, \sigma_k)$ estimates the perturbed score $-\nabla h_{\sigma_k}(\boldsymbol{x}_{k\gamma})$, rather than the true score $-\nabla h(\boldsymbol{x}_{k\gamma})$. Following Sun et al. (2024), we therefore impose the following assumption to quantify the discrepancy between the perturbed and true scores.

**Assumption 4.** Let $h_{\sigma_k}$ be the potential function of the perturbed prior, defined as $p_{\sigma_k}(\boldsymbol{x}) \propto \exp(-h_{\sigma_k}(\boldsymbol{x}))$, $p_{\sigma_k}(\boldsymbol{x}) := \int_{\mathbb{R}^d} p(\boldsymbol{z}) \mathcal{N}(\boldsymbol{x}|\boldsymbol{z}, \sigma_k^2 I) \, d\boldsymbol{z}$, where $p(\boldsymbol{x})$ denotes the unperturbed prior. We assume that for any $\sigma_k > 0$ and $\boldsymbol{x} \in \mathbb{R}^d$, $\|\nabla h_{\sigma_k}(\boldsymbol{x}) - \nabla h(\boldsymbol{x})\|_2 \leq C_1 \sigma_k$ for some constant $C_1 > 0$.

Under Assumption 4, it follows that $\nabla h_{\sigma_k} \to \nabla h$ as $\sigma_k \to 0$. For special cases, such as Gaussian priors, the discrepancy between the perturbed and true scores can be characterized analytically. However, deriving a closed-form bound for this discrepancy is generally intractable for arbitrary prior distributions.

To account for the SGM estimation error, $\mathcal{S}_\theta(\boldsymbol{x}_{k\gamma}, \sigma_k) \approx -\nabla h_{\sigma_k}(\boldsymbol{x}_{k\gamma})$, we require an additional assumption. Since SGMs are trained using the denoising score matching objective (Song et al., 2021b; Ho et al., 2020), their optimal score network satisfies $\mathcal{S}_\theta(\boldsymbol{x}_{k\gamma}, \sigma_k) = -\nabla h_{\sigma_k}(\boldsymbol{x}_{k\gamma})$ with probability 1. Following this characterization and prior work (Sun et al., 2024), we impose the following assumption.

**Assumption 5.** For any $\sigma_k > 0$ and all $\boldsymbol{x} \in \mathbb{R}^d$, the score network satisfies $\|\mathcal{S}_\theta(\boldsymbol{x}, \sigma_k) + \nabla h_{\sigma_k}(\boldsymbol{x})\|_2 \leq \varepsilon_{\sigma_k} < \infty$ and $\|\mathcal{S}_\theta(\boldsymbol{x}, \sigma_k)\|_2 \leq C_2 \sigma_k^{-1}$.

Unlike Sun et al. (2024), we relax the uniform norm bound on the score network to the noise-scale-dependent bound $\|\mathcal{S}_\theta(\boldsymbol{x}, \sigma_k)\|_2 \leq C_2 \sigma_k^{-1}$. This assumption is consistent with empirical observations in seminal score-based generative modeling works (Song & Ermon, 2019; Song et al., 2021b; Song & Ermon, 2020), where the score magnitude is observed to scale as $\|\mathcal{S}_\theta(\boldsymbol{x}, \sigma_k)\|_2 \propto \sigma_k^{-1}$. Additionally, in contrast to prior work (Yang & Wibisono, 2022; Lee et al., 2022; Sun et al., 2024), our analysis does not require the score network to be Lipschitz continuous, which is often violated in practice.

We now present our main result for the ZO-APMC posterior sampling algorithm (12) with SGM prior.

**Theorem 3.** *Let $\pi \propto \ell(\boldsymbol{y}|\boldsymbol{x})p(\boldsymbol{x})$ be the posterior with the likelihood $\ell(\boldsymbol{y}|\boldsymbol{x}) \propto \exp(-f(\boldsymbol{x}))$ and the prior $p(\boldsymbol{x}) \propto \exp(-h(\boldsymbol{x}))$. Suppose the likelihood potential $f$ satisfies Assumptions 1, 2 with Lipschitz constants $L_{f_1}$, $L_{f_2}$, respectively, the prior potential $h$ satisfies Assumption 2 with Lipschitz constant $L_h$ and Assumption 4, and SGM satisfies Assumption 5 with decreasing error $\varepsilon_{\sigma_k} = \mathcal{O}(k^{-1/2})$ for $\sigma_k > 0$, and $k \geq 1$. Define $L_m := \max\{L_{f_2} + L_h, L_{f_1}\}$. Let $(\nu_t)_{t\geq 0}$ denote the law of interpolation in (14) generated by the step size $\gamma$, probability $p$, batch size $b$, smoothing parameter of ZO estimates $\mu$ stated in Theorem 1 with the annealing and noise schedules defined in (13). Then, the time-averaged law $\bar{\nu}_{N\gamma} := \frac{1}{N\gamma} \int_0^{N\gamma} \nu_t \, dt$ satisfies $\mathrm{FI}(\bar{\nu}_{N\gamma} \| \pi) \leq \varepsilon$ after $\mathcal{O}(d^7 L_m^4/\varepsilon^4)$ iterations with $\mathcal{O}(1)$ function evaluations per iteration.*

The full theorem statement and the proof are provided in Appendix B.6. Intuitively, the theorem guarantees that the sample distribution produced by ZO-APMC increasingly captures the posterior score information that governs directions toward high-probability, measurement-consistent reconstructions in inverse problems. This is achieved using only forward model evaluations without any gradient com-

putation in (4). Moreover, ZO-APMC achieves $\varepsilon$-accuracy with only $\mathcal{O}(1)$ function evaluations per iteration, compared to the $\mathcal{O}(d)$ batch sizes typically required by standard batched ZO estimator. This substantially reduces the per-iteration computational and memory costs, enabling scalable posterior sampling for black-box inverse problems. Although the iteration complexity exhibits a high-order polynomial dependence on the dimension, our empirical results on high-dimensional tasks such as MRI reconstruction show that ZO-APMC converges in practice with substantially fewer iterations, matching the iteration count of its gradient-based counterpart APMC (Sun et al., 2024).

Leveraging this result and assuming that the target posterior $\pi$ satisfies a Poincaré inequality, we obtain a stronger convergence guarantee.

**Corollary 2.** *Let $\pi \propto \ell(\boldsymbol{y}|\boldsymbol{x})p(\boldsymbol{x})$ be the posterior with the likelihood $\ell(\boldsymbol{y}|\boldsymbol{x}) \propto \exp(-f(\boldsymbol{x}))$ and the prior $p(\boldsymbol{x}) \propto \exp(-h(\boldsymbol{x}))$. Suppose the likelihood potential $f$ satisfies Assumptions 1, 2 with Lipschitz constants $L_{f_1}, L_{f_2}$, respectively, the target posterior $\pi$ satisfies Assumption 3 with constant $C_{\mathrm{PI}} > 0$, the prior potential $h$ satisfies Assumption 2 with Lipschitz constant $L_h$ and Assumption 4, and SGM satisfies Assumption 5 with decreasing error $\varepsilon_{\sigma_k} = \mathcal{O}(k^{-1/2})$ for $\sigma_k > 0$, and $k \geq 1$. Define $L_m := \max\{L_{f_2}+L_h, L_{f_1}\}$. Let $(\nu_t)_{t\geq 0}$ denote the law of interpolation (14) generated by the step size $\gamma$, probability $p$, batch size $b$, smoothing parameter of ZO estimates $\mu$ stated in Theorem 1 with the annealing schedule defined in (13). Then, the time-averaged law $\bar{\nu}_{N\gamma} := \frac{1}{N\gamma} \int_0^{N\gamma} \nu_t \, dt$ satisfies $\|\bar{\nu}_{N\gamma} - \pi\|_{\mathrm{TV}}^2 \leq \varepsilon$ after $\mathcal{O}(d^7 L_m^4 C_{\mathrm{PI}}^4 / \varepsilon^4)$ iterations with $\mathcal{O}(1)$ function evaluations per iteration.*

Corollary 2 (full statement and proof are in Appendix B.7) intuitively implies that, for posterior distributions without flat valleys, widely separated modes, or heavy tails, the reconstructions produced by ZO-APMC are statistically indistinguishable from samples drawn from the true posterior, up to an $\varepsilon$ error, after $\mathcal{O}(d^7 L_m^4 C_{\mathrm{PI}}^4 / \varepsilon^4)$ iterations using $\mathcal{O}(1)$ memory on average.

*Remark 2.* The proposed estimator in (8) is randomized, yielding an expected per-iteration cost of $\mathcal{O}(1)$. In applications requiring deterministic bounds on memory usage or per-iteration complexity, our estimator can be replaced with a ZO variant of the STORM estimator (Cutkosky & Orabona, 2019). By choosing $\beta = 1 - p$, where $\beta \in [0, 1)$ denotes the STORM momentum parameter, one obtains the same recursive error bound as in Proposition 1. Consequently, the convergence guarantees established in this work continue to hold under this alternative estimator.

Finally, we establish weak convergence of the generated samples to the target posterior, showing that they asymptotically recover the desired posterior law.

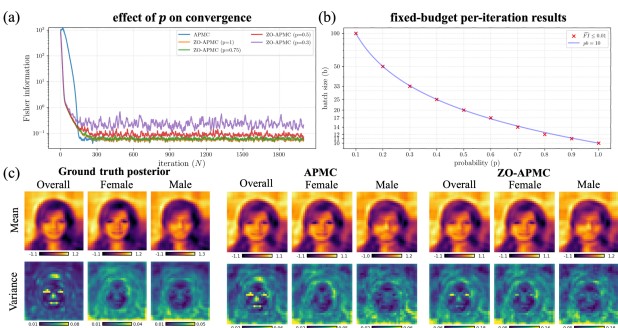

*Figure 2.* (a) Convergence of ZO-APMC with $b = 10$, $b' = 5$, and $\varepsilon_{k^*} = 2.5$ for various $p$, alongside APMC convergence with gradient access. (b) Convergence results for fixed per-iteration cost. Each $\times$ indicate parameter settings $(p, b)$ for which the FI drops below 0.01 after 2000 iterations. (c) Comparison of sample statistics generated by ZO-APMC and APMC with those of the analytical ground truth posterior.

**Theorem 4.** *Let $\pi \propto \ell(\boldsymbol{y}|\boldsymbol{x})p(\boldsymbol{x})$ be the posterior with the likelihood $\ell(\boldsymbol{y}|\boldsymbol{x}) \propto \exp(-f(\boldsymbol{x}))$ and the prior $p(\boldsymbol{x}) \propto \exp(-h(\boldsymbol{x}))$. Suppose the likelihood potential $f$ satisfies Assumptions 1, 2 with Lipschitz constants $L_{f_1}, L_{f_2}$, respectively, the prior potential $h$ satisfies Assumption 2 with Lipschitz constant $L_h$, and Assumption 4, and SGM satisfies Assumption 5 with decreasing error $\varepsilon_{\sigma_k} = \mathcal{O}(k^{-1/2})$ for $\sigma_k > 0$, and $k \geq 1$. Define $L_m := \max\{L_{f_2} + L_h, L_{f_1}\}$. Let $(\nu_t)_{t\geq 0}$ denote the law of interpolation (14) generated by time-varying step size $\gamma_k$, probability $p_k$, batch size $b_k$, and smoothing parameter of ZO estimates $\mu_k$ stated in Theorem 2 with the annealing schedule defined in (13). Then, the time-averaged law $\bar{\nu}_{\tau_n} := \frac{1}{\tau_n} \int_0^{\tau_n} \nu_t \, dt$, where $\tau_n := \sum_{k=1}^n \gamma_k$ converges weakly to $\pi$.*

The complete statement and the proof are provided in Appendix B.8. Next, we validate our theoretical results through statistical and numerical experiments, and demonstrate the performance of ZO-APMC in practical settings.

## 4. Experiments

**Baselines.** Our primary focus is on methods that assume black-box access to the forward model. Accordingly, we compare ZO-APMC against SCG (Huang et al., 2024), DPG (Tang et al., 2024), EnKG (Zheng et al., 2025a), and Forward-GSG and Central-GSG proposed by Zheng et al. (2025a). The GSG methods resemble DPS (Chung et al., 2023b), but approximate the likelihood score via Tweedie's formula combined with ZO approximations. For completeness, we additionally evaluate gradient-based methods in settings where the forward model gradient are available. Specifically, we compare against DPS (Chung et al., 2023b), PnPDM (Wu et al., 2024), and APMC (Sun et al., 2024), an annealed LMC posterior sampling method with gradient access and the gradient-based counterpart of ZO-APMC.

*Table 1.* Quantitative comparison with baselines for MRI reconstruction (SD: sample standard deviation). The best values of each metric for black-box and gradient-access settings are highlighted in **bold** and underline, respectively.

|  | PSNR (dB)↑ | SSIM↑ | NRMSE↓ | SD↓ | MSE↓ |
|---|---|---|---|---|---|
| PnPDM | 30.81 | 0.946 | 3.76e-2 | 2.16e-2 | 8.46e-4 |
| DPS | 34.38 | 0.965 | 2.54e-2 | 2.06e-2 | 4.07e-4 |
| APMC | 36.55 | 0.973 | 1.99e-2 | 2.0e-2 | 2.55e-4 |
| Forward-GSG | 27.8 | 0.918 | 5.42e-2 | 3.26e-2 | 19.1e-4 |
| Central-GSG | 27.78 | 0.917 | 5.43e-2 | 3.27e-2 | 19.2e-4 |
| SCG | 7.1 | 0.711 | 7.67 | 1.38 | 0.21 |
| DPG | 32.17 | 0.953 | 5.4e-2 | **2.69e-2** | 6.5e-4 |
| EnKG | 31.32 | 0.934 | 5.72e-2 | 2.92e-2 | 6.72e-4 |
| ZO-APMC (ours) | **35.29** | **0.966** | **2.28e-2** | 2.99e-2 | **3.29e-4** |

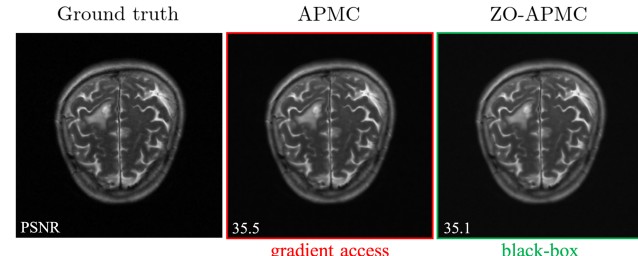

*Figure 3.* Visualization of averaged reconstructions generated by APMC in the setting with gradient access and ZO-APMC in the black-box setting. Despite relying solely on function evaluations, ZO-APMC achieves reconstruction quality comparable to APMC.

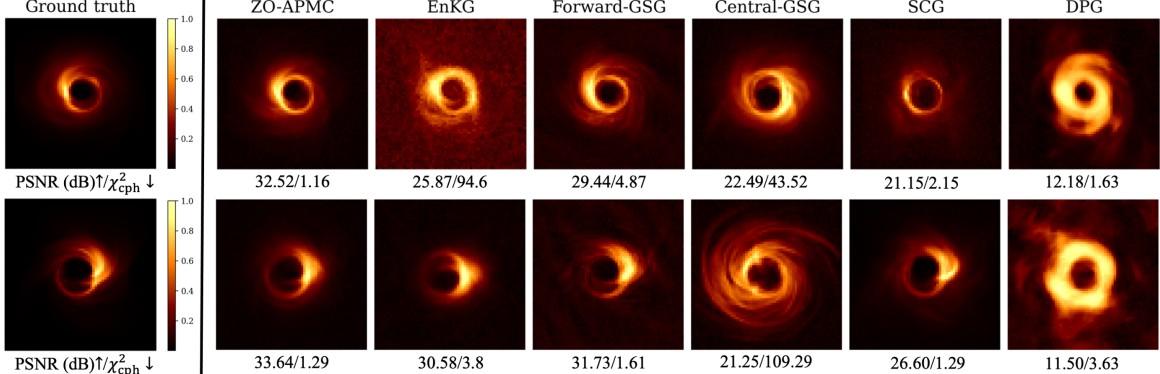

*Figure 4.* Visualization of mean reconstructions for the black-hole imaging inverse problem. Two representative test cases are shown (top and bottom rows), with ground truths in the left column and black-box method reconstructions in the remaining columns. PSNR (dB) and closure-phase error ($\chi^2_{\mathrm{cph}}$, lower is better) are reported below each reconstruction.

### 4.1. Toy Experiments

**Numerical Validation.** To validate the convergence behavior predicted by our theory, we consider a synthetic inverse problem with bimodal 2D Gaussian mixture prior, random forward model $A$, and measurement noise $\xi \sim \mathcal{N}(0, I)$. To mimic SGM estimation error, we use the analytical prior score corrupted by additive Gaussian noise with std. dev. $\varepsilon_{k^*} = 2.5$. We generate 1000 samples with ZO-APMC from 20 random initializations of $A$ and report the mean FI relative to the analytical posterior. Fig. 2a shows that, with $b = 10$, $b' = 5$, ZO-APMC converges to near-zero FI for $p \in \{1, 0.75, 0.5\}$, but becomes unstable at $p = 0.3$ due to the reduced total number of function evaluations. To validate converges under $\mathcal{O}(1)$ batch complexity, we vary $p$ and $b$ while keeping the per-iteration cost fixed ($pb = 10$) in Fig. 2b. Across all parameter pairs, ZO-APMC converges below 0.01 FI, consistent with our theoretical predictions.

**Statistical Validation.** To validate ZO-APMC's recovery of posterior mode statistics, we consider a compressed sensing problem with a bimodal Gaussian mixture prior constructed from $32 \times 32$ CelebA (Liu et al., 2015) images, a random forward model $A \in \mathbb{R}^{115 \times 1024}$, and a Gaussian measurement noise $\xi \sim \mathcal{N}(0, 0.01I)$. The prior modes correspond to the "male" and "female" attributes shifted by $+1$ and $-1$, respectively, to ensure clear separation. We train a shallow

customized SGM (Nichol & Dhariwal, 2021) on this prior and generate 1000 samples using ZO-APMC and APMC, where ZO-APMC uses $p = 0.5$, $b = 50$, and $b' = 5$. Fig. 2c shows that ZO-APMC accurately recovers the mean and variance of both posterior modes, closely matching gradient-based APMC up to a slight increase in variance due to ZO estimations. This can be further reduced by increasing $b$. Additional results and details are provided in Appendix C.

### 4.2. Magnetic Resonance Imaging (MRI)

Imaging inverse problems (e.g., MRI recon.) are widely used benchmarks. Although our primary focus is on more challenging black-box forward models, we additionally evaluate our method on linear MRI recon. problem for completeness and to demonstrate the effectiveness of our variance-reduction mechanism in a high-dimensional setting.

**Problem Setting.** We consider a radial subsampling mask with acceleration factor of $4\times$. For evaluation, we use the SGM prior from Sun et al. (2024), which is pre-trained on the FastMRI brain dataset (Zbontar et al., 2019), and evaluate all methods on a separate test set provided in that work to ensure a consistent comparison. We randomly select 40 images of size $256 \times 256$ and, for each method, generate 20 reconstructions per image, average them, and report the

*Table 2.* Quantitative comparison with baselines for black-hole imaging (SD: sample standard deviation). Best values for black-box and gradient-access settings are shown in **bold** and *underline*, respectively.

| | PSNR↑ | Blurred PSNR↑ | $\chi^2_{\text{cph}} \downarrow$ | $\chi^2_{\text{camp}} \downarrow$ | SD↓ |
|---|---|---|---|---|---|
| PnPDM | 26.48 | 32.31 | 11.48 | 23.54 | 4.5e-2 |
| DPS | 25.61 | 30.84 | 12.39 | 17.72 | 4.32e-2 |
| APMC | 26.23 | 31.32 | 11.78 | 19.23 | 4.34e-2 |
| Forward-GSG | 26.21 | 31.47 | 6.77 | 14.06 | 2.99e-2 |
| Central-GSG | 21.63 | 23.73 | 80.31 | 78.5 | 4.5e-2 |
| SCG | 22.21 | 25.51 | 23.72 | 14.23 | 1.7e-2 |
| DPG | 12.33 | 14.02 | 8.17 | 30.44 | **1.6e-2** |
| EnKG | 22.86 | 27.69 | 64.37 | 33.44 | 0.925 |
| ZO-APMC (Ours) | **26.71** | **32.86** | **5.42** | **11.23** | 3.02e-2 |

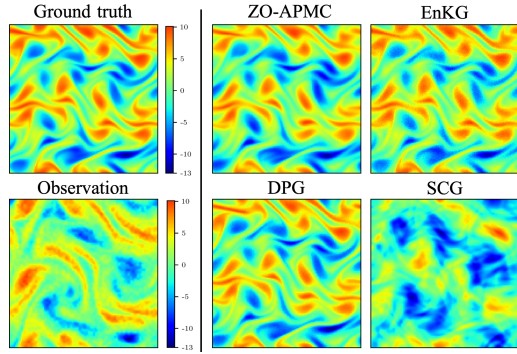

*Figure 5.* Visualization of mean samples for the Navier–Stokes inverse problem across black-box methods.

mean image-quality metrics across the test set. For ZO-APMC, we use $p = 0.2$, $b = 10^4$, and $b' = 10^3$.

**Results.** Fig. 3 demonstrates that both ZO-APMC and APMC yield visually indistinguishable reconstructions for representative brain MRI cases showing pathology, with ZO-APMC accurately capturing fine details without gradient information. Table 1 shows that ZO-APMC consistently achieve higher reconstruction quality than other black-box baselines in all image quality metrics and closely matches the APMC with gradient access. Our method yields slightly higher SD than DPG but this can be alleviated by increasing $p$ or $b$, albeit at larger computational cost.

### 4.3. Black-Hole Imaging

**Problem Setting.** Black-hole interferometric imaging reconstructs black-hole images from "visibility" measurements acquired by Earth-based telescope arrays. We use the SGM prior pre-trained on GRMHD (Wong et al., 2022) images of size $64 \times 64$, the highly nonlinear forward model, and 100 test images provided by InverseBench (Zheng et al., 2025b). For each method, we generate 5 reconstructions per image, average them, and report the resulting mean metrics across the test set. Since the image sizes are small, we use $p = 1$ with $b = 1024$. Evaluation is based on the chi-square errors of the closure phases ($\chi^2_{\text{cph}}$) and closure amplitudes ($\chi^2_{\text{camp}}$), which quantify how well the reconstructions fit the measurements. Because the black-hole imaging system captures only low spatial frequencies, we follow Akiyama et al. (2019) and compute PSNR for both the original and blurred reconstructions at the system's intrinsic resolution.

**Results.** Fig. 4 shows two representative cases of black-hole reconstructions generated by ZO-APMC and other black-box baselines, alongside the ground truth. ZO-APMC produces reconstructions that most closely match the ground truth, whereas baselines introduce artifacts and lose fine details. Table 2 shows that ZO-APMC outperforms all baselines across all metrics except SD, which can be mitigated by increasing batch size $p$ or $b$ at additional cost.

### 4.4. Navier–Stokes Equation

**Problem Setting.** The Navier–Stokes equation is a standard fluid-dynamics benchmark (Iglesias et al., 2013), widely used from ocean dynamics to climate modeling, where atmospheric observations calibrate initial conditions for numerical forecasts. Computing forward model gradients via auto-differentiation is impractical because it requires differentiating through a PDE solver. We use the Navier–Stokes forward model, 10 test images, and the pre-trained SGM prior provided by InverseBench (Zheng et al., 2025b). For each black-box method, we generate 5 samples per test image, average them, and report the mean NRMSE across the 10 test images. We repeat this procedure for 3 noise levels $\sigma_{\text{noise}} \in \{0, 1, 2\}$. Additional experimental details are provided in Appendix C.5.

**Results.** Fig. 5 shows results obtained with $\sigma_{\text{noise}} = 1$, demonstrating that ZO-APMC generates samples that qualitatively preserve key flow features, comparable to EnKG and DPG, whereas SCG fails to do so. Moreover, EnKG produces noticeably noisier samples than ZO-APMC. Quantitative results in Appendix C.5 show that, although ZO-APMC does not outperform EnKG and DPG in NRMSE, it achieves comparable performance to DPG while providing convergence guarantees and a principled understanding of how hyperparameters affect performance, unlike competing methods that rely on heuristic approximations.

## 5. Conclusion

We established a theoretically grounded framework for variance-reduced zeroth-order (ZO) sampling in non-log-concave settings, with applications to inverse problems. We derived the first non-asymptotic complexity guarantees for non-log-concave ZO sampling and for black-box posterior sampling with SGM priors. Empirically, our proposed ZO-APMC method achieved state-of-the-art performance among black-box methods in MRI reconstruction and black-hole imaging, while remaining competitive on the Navier–Stokes inverse problem. Future work includes extending our method to underdamped Langevin dynamics and latent diffusion models for improved sampling efficiency.

## Acknowledgements

This work was partially supported by National Insititutes of Health grant NIH/NHLBI R01-HL153430 (PI: Sharif), and supported in part by NSF DMS-2502560 and CNS-2313109.

## Impact Statement

This work establshes the first theoretically grounded framework for zeroth-order sampling from non-log-concave distributions, addressing a fundamental gap in derivative-free probabilistic inference. We further develop the first zeroth-order posterior sampling framework with convergence guarantees for black-box inverse problems using score-based generative priors. These advances broaden the applicability of sampling methods to settings where gradients are inaccessible, undefined, or prohibitively expensive, including scientific simulators, medical imaging systems, and physics-based inverse problems. By enabling principled probabilistic inference under limited forward model access, our work may expand the use of machine learning in scientific and engineering domains. As with other sampling and generative methods, practical deployment in application domains requires careful empirical validation and appropriate domain expertise.

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

# A. Comparison to Related Work

In this section, we compare ZO-APMC against existing posterior sampling methods based on SGM priors, highlighting their strengths and limitations.

## A.1. Black-box Posterior Sampling Algorithms

There are three other prior works that studied posterior sampling with SGM priors for black-box (derivative-free) forward models. The most recent is EnKG, which estimates the likelihood score using the ensemble-based statistical linearization (Zheng et al., 2025a). Unlike our algorithm, EnKG is not designed to sample from the full posterior and therefore offers limited uncertainty quantification. In our experiments, we observed that EnKG also becomes highly inefficient when the evaluation of the forward model is cheaper than computation of the score network, which is the case in MRI reconstruction with linear forward model. In addition, as reflected in our results, in black hole imaging and MRI reconstruction, it produced noticeably lower reconstruction quality than our method. However, it performed better than our method for Navier–Stokes inverse problem because noisy function evaluations of forward model makes the guidance term unstable. In EnKG paper, authors proposed Forward-GSG and Central-GSG baselines, which are originated from DPS algorithm but the guidance term is estimated by mini-batch ZO estimate. Since they have additional approximation error due to heuristic approximation of the guidance term (Chung et al., 2023b), they generate samples with larger reconstruction errors, as reflected in our results. However, they require less iteration as they use SDE from diffusion models (Song et al., 2021b) as opposed to Langevin sampling, which requires longer iterations.

SCG (Huang et al., 2024) and DPG (Tang et al., 2024), which cast diffusion guidance in a stochastic control framework and steer the sampling process via an estimated value function. Similar to EnKG, neither of the targets to sample from the posterior distribution due to intractable score likelihood term, so uncertainty quantification cannot be done as opposed to our algorithm. Moreover, SCG relies on a threshold parameter that disables the likelihood-based guidance in certain iterations. We found that improper tuning of this threshold leads to severely degraded performance, yet the method provides no theoretical guidance for selecting it. As a result, practitioners must rely on grid search or extensive hyperparameter tuning. However, in our setting, $p$ and $b$ can be adjusted easily once a budget for per-iteration is determined. Although we did not experiment with, since SCG is based on value function (not approximation of gradient), it could perform better than our method for rule-based inverse problems.

Similar to our approach, DPG also relies on Monte Carlo mini-batch estimates of the posterior score. However, unlike our method, DPG does not include any variance-reduction mechanism, and incorporating such mechanisms is nontrivial due to the DDPM (Ho et al., 2020) and DDIM (Song et al., 2021a) sampling procedures it depends on. In addition, their update rule couples the likelihood and prior score terms, preventing the decoupling that enables our variance-reduced design. Empirically, these differences lead to clear performance gaps: in both the black hole and MRI reconstruction tasks, our method achieves substantially better reconstruction quality, whereas on the Navier–Stokes example DPG performs slightly better, which may be due to instability in the ZO score approximations for this specific problem.

## A.2. Gradient-based Posterior Sampling Algorithms

There are many gradient-based posterior sampling with SGM priors proposed for solving inverse problems. As can be seen from our MRI reconstruction experiments (Table 1), APMC (Sun et al., 2024), which is the closest work to ours, generated slightly better reconstructions than the proposed ZO-APMC algorithm. This shows that when one can compute the gradient of the forward model exactly, it is more preferable. However, this may not be possible in commercial MRI applications or closed-source systems (Karakuzu et al., 2025), where one can only have access to input and output of the forward operator. In that scenario, our model performs the best among black-box posterior samplers with comparable performance to the gradient-based model. Similar trend can be seen in black hole imaging experiments as well. Similar to our work, the APMC paper also derived an upper bound on the FI. However, beyond establishing an FI bound in the black-box setting, we additionally prove convergence in squared total variation distance and show weak asymptotic convergence to the target distribution without using Lipschitzness assumption of the score network.

Besides APMC, we used DPS (Chung et al., 2023b) and PnP-DM (Wu et al., 2024) posterior sampling algorithms as gradient-based baselines. Although our method treats the forward model as black-box, it performed better than DPS in experiments. We attributed this to the heuristic approximation of the guidance term in DPS, which is called Jensen gap which cannot be controlled explicitly as we control the estimator error via PAGE variance reduction mechanism. PnP-DM

*Table 3.* A conceptual overview of posterior sampling approaches for probabilistic imaging. The *"Annealing"* column highlights distinctions among MCMC-based methods. The *"Black-box access"* column shows whether the corresponding method assumes black-box access and works when gradients of the forward model is unavailable. The $\boldsymbol{A}(\cdot)$ column shows the assumption on the type of forward model. This table extends that of Sun et al. (2024) by incorporating additional black-box posterior sampling algorithms.

| Category | Reference | Generative prior | Model agnostic | $\boldsymbol{A}(\cdot)$ | Convergence guarantees | Annealing | Black-box access |
|---|---|---|---|---|---|---|---|
| Variational Bayesian | (Sun & Bouman, 2021) | ✗ | ✗ | General | ✗ | – | ✗ |
| | (Feng et al., 2023) | ✓ | ✗ | General | ✗ | – | ✗ |
| DM-based | (Song et al., 2022) | ✓ | ✓ | Linear | ✗ | – | ✗ |
| | (Chung et al., 2023b) | ✓ | ✓ | General | ✗ | – | ✗ |
| | (Liu et al., 2023) | ✓ | ✗ | Linear | ✗ | – | ✗ |
| | (Tang et al., 2024) | ✓ | ✗ | General | ✗ | – | ✓ |
| | (Huang et al., 2024) | ✓ | ✗ | General | ✗ | – | ✓ |
| | (Zheng et al., 2025a) | ✓ | ✗ | General | ✗ | – | ✓ |
| MCMC-based | (Jalal et al., 2021) | ✓ | ✓ | Linear | ✓[1] | ✓ | ✗ |
| | (Kawar et al., 2021) | ✓ | ✓ | Linear | ✗ | ✓ | ✗ |
| | (Laumont et al., 2022) | ✗ | ✓ | General | ✓ | ✗ | ✗ |
| | (Coeurdoux et al., 2024) | ✓ | ✓ | Linear | ✗ | ✗ | ✗ |
| | (Bouman & Buzzard, 2023) | ✗ | ✓ | Linear | ✓[2] | ✗ | ✗ |
| | (Sun et al., 2024) | ✓ | ✓ | General | ✓ | ✓ | ✗ |
| MCMC-based | **Ours** | ✓ | ✓ | General | ✓ | ✓ | ✓ |

[1]Requires $\boldsymbol{A}(\cdot)$ to be a Gaussian random matrix.  [2]Guarantees on asymptotic convergence.

provides a principled framework for posterior sampling, but it applies to inverse problems only when implemented within the EDM framework (Karras et al., 2022) as opposed to our method. Moreover, although PnP-DM establishes upper bounds on FI, it provides neither asymptotic convergence to the true posterior nor non-asymptotic guarantees in squared total variation distance. Consequently, it remains unclear whether samples produced after many iterations will coincide with those from the exact true posterior.

## B. Theoretical Results

**Notation.** Throughout the proof, we work within the probability space $(\Omega, \mathcal{F}, \mathbb{P})$, where $\Omega$ denotes the sample space, $\mathcal{F}$ the $\sigma$-algebra, and $\mathbb{P}$ the probability measure. We define the following filtration containing all randomness up to iteration $k$

$$\mathcal{F}_k := \sigma(\boldsymbol{x}_0, \boldsymbol{u}_0^i, \boldsymbol{Z}_0, \ldots, \boldsymbol{u}_k^i, \boldsymbol{Z}_k), \tag{15}$$

where $\boldsymbol{Z}_k := \boldsymbol{B}_{(k+1)\gamma} - \boldsymbol{B}_{k\gamma}$ and $\boldsymbol{u}_k^i$ denotes the collection of Gaussian vectors used in the ZO estimates at iteration $k$.

$\lceil \cdot \rceil$ is the ceiling function and it is defined as $\lceil x \rceil := \min\{n \in \mathbb{Z} : n \geq x\}$.

For two nonnegative sequences $(a_n)_{n \geq 0}$ and $(b_n)_{n \geq 0}$, the notation $a_n \lesssim b_n$ denotes that there exists a constant $C > 0$,

independent of $n$, such that

$$a_n \leq C b_n,$$

for all $n \geq 0$.

For a random variable $X : \Omega \to \mathbb{R}^n$, we write its expectation as

$$\mathbb{E}[X] = \int_\Omega X(\omega) \, \mathbb{P}(d\omega).$$

The posterior distribution of interest is of the form

$$\pi(\boldsymbol{x}|\boldsymbol{y}) \propto \ell(\boldsymbol{y}|\boldsymbol{x}) p(\boldsymbol{x}),$$

where we define $f(\boldsymbol{x})$ and $h(\boldsymbol{x})$ as potential functions of the likelihood $\ell(\boldsymbol{y}|\boldsymbol{x}) \propto \exp(-f(\boldsymbol{x}))$ and the prior $p(\boldsymbol{x}) \propto \exp(-h(\boldsymbol{x}))$, respectively. Moreover, we define $h_{\sigma_k}$ as the potential function of the perturbed prior given as $p_{\sigma_k}(\boldsymbol{x}) \propto \exp(-h_{\sigma_k}(\boldsymbol{x}))$, $p_\sigma(\boldsymbol{x}) := \int_{\mathbb{R}^d} p(\boldsymbol{z}) \mathcal{N}(\boldsymbol{x}|\boldsymbol{z}, \sigma^2 I) \, d\boldsymbol{z}$, where $p(\boldsymbol{x})$ is the unperturbed prior. For simplicity, we omit the explicit dependence on $\boldsymbol{y}$. Recall that

$$-\nabla \log \pi(\boldsymbol{x}) = \nabla f(\boldsymbol{x}) + \nabla h(\boldsymbol{x}). \tag{16}$$

We denote the zeroth-order approximation of the forward model gradient as follows

$$\widetilde{\nabla} f_\mu(\boldsymbol{x}_{k\gamma}, \boldsymbol{u}) := \frac{f(\boldsymbol{x}_{k\gamma} + \mu \boldsymbol{u}) - f(\boldsymbol{x}_{k\gamma})}{\mu} \boldsymbol{u}, \tag{17}$$

where $\boldsymbol{u} \sim \mathcal{N}(0, I)$ and $\mu > 0$. The expectation of the zeroth-order approximation is denoted as $\nabla f_\mu(\boldsymbol{x}_{k\gamma}) := \mathbb{E}_{\boldsymbol{u}}\left[\widetilde{\nabla} f_\mu(\boldsymbol{x}_{k\gamma}, \boldsymbol{u})\right]$. For notational convenience, we also define

$$\Delta_k := \mathbb{E}\left[\|\boldsymbol{x}_{(k+1)\gamma} - \boldsymbol{x}_{k\gamma}\|_2^2\right], \tag{18}$$

as the expected squared $\ell_2$-distance between consecutive iterates. For convenience, we recall the definition of the Kullback–Leibler (KL) divergence between two probability densities $\nu$ and $\pi$:

$$\mathrm{KL}(\nu\|\pi) = \int_{\mathbb{R}^n} \nu(x) \log \frac{\nu(x)}{\pi(x)} \, dx.$$

The Fisher information (FI) is given by

$$\mathrm{FI}(\nu\|\pi) = \int_{\mathbb{R}^n} \left\|\nabla \log \frac{\nu(x)}{\pi(x)}\right\|_2^2 \nu(x) \, dx = \int_{\mathbb{R}^n} \|\nabla \log \nu(x) - \nabla \log \pi(x)\|_2^2 \nu(x) \, dx.$$

Total variation (TV) distance between two probability measures $\mu$ and $\nu$ on a measurable space $(\mathcal{X}, \mathcal{F})$ is given b by

$$\|\nu - \pi\|_{\mathrm{TV}} := \sup_{A \in \mathcal{F}} |\nu(A) - \pi(A)| = \frac{1}{2} \int_{\mathcal{X}} |d\nu - d\pi|.$$

Unless otherwise stated, $\|\cdot\|$ denotes the squared $\ell_2$-norm, i.e. $\|\cdot\|_2$. $C_L^{1,1}$ denotes the class of differentiable functions $f : \mathbb{R}^d \to \mathbb{R}$ whose gradients are $L$-Lipschitz, i.e.,

$$\|\nabla f(\boldsymbol{x}) - \nabla f(\boldsymbol{y})\|_2 \leq L\|\boldsymbol{x} - \boldsymbol{y}\|_2, \quad \forall \boldsymbol{x}, \boldsymbol{y} \in \mathbb{R}^d. \tag{19}$$

### B.1. Lemmas

We begin by reviewing the key lemmas from the zeroth-order optimization and non-log-concave sampling literature. The following lemma summarizes key properties of zeroth-order approximations used in our analysis.

**Lemma 1** (Lemma 6.2 under Section 6.1.2.1 in Lan (2020)). *Suppose that $f(\boldsymbol{x}) \in C_L^{1,1}$, i.e., $f$ is continuously differentiable and its gradient is $L$-Lipschitz, and let $f_\mu(\boldsymbol{x}) := \mathbb{E}_{\boldsymbol{u}}[f(\boldsymbol{x} + \mu \boldsymbol{u})]$. Then the following statements hold:*

(a) $f_\mu \in C_{L_\mu}^{1,1}(\mathbb{R}^d)$, where $L_\mu \le L$,

(b) $\|\nabla f_\mu(\boldsymbol{x}) - \nabla f(\boldsymbol{x})\| \le \frac{1}{2}\mu L(d+3)^{\frac{3}{2}}$,

(c) $\mathbb{E}_{\boldsymbol{u}}[\|\widetilde{\nabla} f_\mu(\boldsymbol{x}, \boldsymbol{u})\|^2] \le \frac{1}{2}\mu^2 L^2 (d+6)^3 + 2(d+4)\|\nabla f(\boldsymbol{x})\|_2^2$,

where $\widetilde{\nabla} f_\mu(\boldsymbol{x}, \boldsymbol{u}) := \frac{f(\boldsymbol{x}+\mu\boldsymbol{u})-f(\boldsymbol{x})}{\mu}\boldsymbol{u}$ for $\boldsymbol{u} \sim \mathcal{N}(0, I)$ and any $\boldsymbol{x} \in \mathbb{R}^d$, $\mu > 0$.

The following lemma concerns the density evolution of an interpolated diffusion process.

**Lemma 2** (Lemma 23 in Balasubramanian et al. (2022)). *Consider the stochastic process defined by*

$$\boldsymbol{x}_t := \boldsymbol{x}_0 - t g_0 + \sqrt{2}\boldsymbol{B}_t, \quad \text{with } g_0 = g(\boldsymbol{x}_0), \quad \boldsymbol{x}_0 \sim \nu_0$$

*where $g_0$ is integrable and $\{B_t\}_{t\ge0}$ is a standard Brownian motion in $\mathbb{R}^d$ independent of $(\boldsymbol{x}_0, g_0)$. Then, writing $\nu_t$ for the probability density of $\boldsymbol{x}_t$, we have*

$$\frac{d}{dt}\mathrm{KL}(\nu_t\|\pi) \le -\frac{3}{4}\mathrm{FI}(\nu_t\|\pi) + \mathbb{E}\left[\|\nabla f(\boldsymbol{x}_t) - g_0\|^2\right],$$

*where we recall that $\pi \propto e^{-f}$, and the expectation in the last term is with respect to $x_0 \sim \nu_0$ and $x_t \sim \nu_t$.*

We also used the following lemma to bound the Fisher information, which is taken from (Chewi et al., 2024).

**Lemma 3** (Lemma 20 in Chewi et al. (2024)). *Assume that $\nabla \log \pi(\boldsymbol{x})$ is $L_\pi$-Lipschitz. For any probability measure $\nu$, it holds that*

$$\mathbb{E}_\nu\left[\|\nabla \log \pi(\boldsymbol{x})\|^2\right] \le \mathrm{FI}(\nu\|\pi) + 2dL_\pi.$$

We use the following lemma to derive an upper bound on total variation (TV) distance.

**Lemma 4** (Theorem 3.1 in Guillin et al. (2009)). *If $\pi$ satisfies a Poincaré inequality, i.e. for every smooth, compactly supported $f : \mathbb{R}^d \to \mathbb{R}$,*

$$\mathrm{Var}_\pi(f) \le C_{\mathrm{PI}}\mathbb{E}_\pi[\|\nabla f\|^2],$$

*then for any probability measure $\nu$,*

$$\|\nu - \pi\|_{\mathrm{TV}}^2 \le 4C_{\mathrm{PI}}\mathrm{FI}(\nu\|\pi).$$

## B.2. Proof of Proposition 1

**Proposition 1** *Suppose Assumption 1 holds, and let $(\boldsymbol{x}_{k\gamma})_{k\ge0}$ be generated by (9), where $\boldsymbol{g}_k$ is given by (8). Define $e_k^2 := \mathbb{E}[\|\boldsymbol{g}_k - \nabla f_\mu(\boldsymbol{x}_{k\gamma})\|_2^2]$. Then, for any $\gamma > 0$,*

$$e_{k+1}^2 \le \frac{p\sigma_{k+1}^2}{b} + (1-p)e_k^2 + \frac{4(1-p)dL_1^2\Delta_k}{\mu^2 b'}, \tag{20}$$

*where $\sigma_{k+1}^2 := \mathbb{E}[\|\widetilde{\nabla} f_\mu(\boldsymbol{x}_{(k+1)\gamma}, \boldsymbol{u}_{k+1}^i)\|^2]$ and $\Delta_k := \mathbb{E}[\|\boldsymbol{x}_{(k+1)\gamma} - \boldsymbol{x}_{k\gamma}\|^2]$*

*Proof.* Using the definition of $\boldsymbol{g}_k$ in (8), we can bound $e_{k+1}^2$ as follows

$$e_{k+1}^2 = p\,\mathbb{E}\left[\left\|\nabla f_\mu(\boldsymbol{x}_{(k+1)\gamma}) - \frac{1}{b}\sum_{i=1}^b \widetilde{\nabla} f_\mu(\boldsymbol{x}_{(k+1)\gamma}, \boldsymbol{u}_{k+1}^i)\right\|^2\right]$$

$$+ (1-p)\,\mathbb{E}\left[\left\|\nabla f_\mu(\boldsymbol{x}_{(k+1)\gamma}) - \boldsymbol{g}_k - \frac{1}{b'}\sum_{i=1}^{b'}\left(\widetilde{\nabla} f_\mu(\boldsymbol{x}_{(k+1)\gamma}, \boldsymbol{u}_{k+1}^i) - \widetilde{\nabla} f_\mu(\boldsymbol{x}_{k\gamma}, \boldsymbol{u}_{k+1}^i)\right)\right\|^2\right] \tag{21}$$

where $b'$, $b$ denote the small and large batch sizes, respectively, and $\boldsymbol{u}_{k+1}^i \sim \mathcal{N}(0, I)$ in $\mathbb{R}^d$. Here, $k$ and $i$ denote the iteration and sample indices, respectively, for the sampled Gaussian vectors. We can upper bound the first expectation as

$$\mathbb{E}\left[\left\|\nabla f_\mu(\boldsymbol{x}_{(k+1)\gamma}) - \frac{1}{b}\sum_{i=1}^{b}\widetilde{\nabla}f_\mu\big(\boldsymbol{x}_{(k+1)\gamma}, \boldsymbol{u}_{k+1}^i\big)\right\|^2\right] = \frac{1}{b}\mathbb{E}\left[\left\|\nabla f_\mu(\boldsymbol{x}_{(k+1)\gamma}) - \widetilde{\nabla}f_\mu(\boldsymbol{x}_{(k+1)\gamma}, \boldsymbol{u}_{k+1}^i)\right\|^2\right] \tag{22}$$

$$\leq \frac{1}{b}\underbrace{\mathbb{E}\left[\left\|\widetilde{\nabla}f_\mu(\boldsymbol{x}_{(k+1)\gamma}, \boldsymbol{u}_{k+1}^i)\right\|^2\right]}_{=\sigma_{k+1}^2}. \tag{23}$$

In (22), we expand the square and use $\mathbb{E}[\widetilde{\nabla}f_\mu(\boldsymbol{x}_{(k+1)\gamma}, \boldsymbol{u}_{k+1}^i)|\mathcal{F}_k] = \nabla f_\mu(\boldsymbol{x}_{(k+1)\gamma})$ with the fact that $\boldsymbol{u}_{k+1}^i$ are identically and independently distributed. In (23), we use the second-moment bound on the variance with definition of $\sigma_{k+1}^2$. Plugging this upper bound into (21), we get

$$e_{k+1}^2 \leq \frac{p\sigma_{k+1}^2}{b} + (1-p)\mathbb{E}\left[\left\|\nabla f_\mu(\boldsymbol{x}_{(k+1)\gamma}) - \boldsymbol{g}_k - \frac{1}{b'}\sum_{i=1}^{b'}\Big(\widetilde{\nabla}f_\mu(\boldsymbol{x}_{(k+1)\gamma}, \boldsymbol{u}_{k+1}^i) - \widetilde{\nabla}f_\mu(\boldsymbol{x}_{k\gamma}, \boldsymbol{u}_{k+1}^i)\Big)\right\|^2\right] \tag{24}$$

$$= \frac{p\sigma_{k+1}^2}{b} + (1-p)\mathbb{E}\left[\left\|\nabla f_\mu(\boldsymbol{x}_{k\gamma}) - \boldsymbol{g}_k + \nabla f_\mu(\boldsymbol{x}_{(k+1)\gamma}) - \nabla f_\mu(\boldsymbol{x}_{k\gamma})\right.\right.$$

$$\left.\left. - \frac{1}{b'}\sum_{i=1}^{b'}\Big(\widetilde{\nabla}f_\mu(\boldsymbol{x}_{(k+1)\gamma}, \boldsymbol{u}_{k+1}^i) - \widetilde{\nabla}f_\mu(\boldsymbol{x}_{k\gamma}, \boldsymbol{u}_{k+1}^i)\Big)\right\|^2\right] \tag{25}$$

$$= \frac{p\sigma_{k+1}^2}{b} + (1-p)e_k^2 + \frac{(1-p)}{b'}\mathbb{E}\left[\left\|\nabla f_\mu(\boldsymbol{x}_{(k+1)\gamma}) - \nabla f_\mu(\boldsymbol{x}_{k\gamma}) - \Big(\widetilde{\nabla}f_\mu(\boldsymbol{x}_{(k+1)\gamma}, \boldsymbol{u}_{k+1}^i) - \widetilde{\nabla}f_\mu(\boldsymbol{x}_{k\gamma}, \boldsymbol{u}_{k+1}^i)\Big)\right\|^2\right] \tag{26}$$

$$\leq \frac{p\sigma_{k+1}^2}{b} + (1-p)e_k^2 + \frac{(1-p)}{b'}\mathbb{E}\left[\left\|\widetilde{\nabla}f_\mu(\boldsymbol{x}_{(k+1)\gamma}, \boldsymbol{u}_{k+1}^i) - \widetilde{\nabla}f_\mu(\boldsymbol{x}_{k\gamma}, \boldsymbol{u}_{k+1}^i)\right\|^2\right] \tag{27}$$

$$= \frac{p\sigma_{k+1}^2}{b} + (1-p)e_k^2 + \frac{(1-p)}{\mu^2 b'}\mathbb{E}\left[\left\|\Big(f(\boldsymbol{x}_{(k+1)\gamma} + \mu\boldsymbol{u}_{k+1}^i) - f(\boldsymbol{x}_{k\gamma} + \mu\boldsymbol{u}_{k+1}^i)\Big)\right.\right.$$

$$\left.\left. - \Big(f(\boldsymbol{x}_{(k+1)\gamma}) - f(\boldsymbol{x}_{k\gamma})\Big)\right\|^2\left\|\boldsymbol{u}_{k+1}^i\right\|^2\right] \tag{28}$$

$$\leq \frac{p\sigma_{k+1}^2}{b} + (1-p)e_k^2 + \frac{4(1-p)L_1^2\Delta_k}{\mu^2 b'}\mathbb{E}\left[\left\|\boldsymbol{u}_{k+1}^i\right\|^2\right] \tag{29}$$

$$= \frac{p\sigma_{k+1}^2}{b} + (1-p)e_k^2 + \frac{4(1-p)dL_1^2\Delta_k}{\mu^2 b'} \tag{30}$$

where $\Delta_k := \mathbb{E}\left[\|\boldsymbol{x}_{(k+1)\gamma} - \boldsymbol{x}_{k\gamma}\|^2\right]$. Note that we add and subtract $\nabla f_\mu(\boldsymbol{x}_{k\gamma})$ in (25). To get (26), we use the fact that random variables $\boldsymbol{u}_{k+1}^i \sim \mathcal{N}(0, I)$ are i.i.d., and calculate the conditional expectations conditioned with respect to $\mathcal{F}_k$ and then use the definition of $e_k^2$. We use second-moment bound on variance in (27) and use the zeroth-order definition to get (28). Following these steps, we apply Assumption 1 and use the independence of $\boldsymbol{u}_{k+1}^i$ from $\boldsymbol{x}_{k\gamma}$ and $\boldsymbol{x}_{(k+1)\gamma}$ to obtain (29). This concludes our proof. $\qquad\square$

## B.3. Proof of Theorem 1

**Theorem 1** *Let $(\nu_t)_{t\geq 0}$ denote the law of interpolation (10), and let $\pi \propto \exp(-f)$ be the target distribution, where the potential function $f$ satisfies Assumptions 1 and 2 with Lipschitz constants $L_1$ and $L_2$, respectively. Then, for any step size $\gamma \in \left(0, \frac{1}{L_m\sqrt{52\phi(\mu)}}\right]$, where $\phi(\mu) := 1 + \frac{4(1-p)d}{p\mu^2 b'}$ and $L_m := \max\{L_1, L_2\}$, and for any $N \geq 1$, the Fisher information satisfies*

$$\frac{1}{N\gamma}\int_0^{N\gamma}\mathrm{FI}(\nu_t\|\pi)\,dt \leq \frac{C_0}{N\gamma} + \frac{13d(d+2)}{2b} + \frac{13}{8}\mu^2 L_2^2(d+3)^3 + 14\gamma L_m^2\phi(\mu)d, \tag{31}$$

*where $C_0 > 0$ is a numerical constant. Furthermore, let*

$$\gamma = \frac{1}{L_m N^{3/4} d^{7/4}}, \quad p = \frac{L_m}{N^{1/4} d^{1/4}} \quad b = \left\lceil \frac{1}{p} \right\rceil, \quad and \quad \mu^2 = \frac{1}{L_m N^{1/4} d^{5/4}} \tag{32}$$

*Then, to achieve $\mathrm{FI}(\bar{\nu}_{N\gamma} \| \pi) \le \varepsilon$ for the time averaged law $\bar{\nu}_{N\gamma} := \frac{1}{N\gamma} \int_0^{N\gamma} \nu_t \, dt$, $\mathcal{O}\left(\frac{d^7 L_m^4}{\varepsilon^4}\right)$ iterations with $\mathcal{O}(1)$ function evaluations per iteration is sufficient.*

*Proof.* At initialization, we choose $\gamma_0, b_0, \mu_0 > 0$ and $p_0 \in (0, 1]$, and define $\boldsymbol{g}_0$ as

$$\boldsymbol{g}_0 := \frac{1}{b_0} \sum_{i=1}^{b_0} \widetilde{\nabla} f_{\mu_0}(\boldsymbol{x}_0, \boldsymbol{u}_0^i), \quad \boldsymbol{u}_0^i \sim \mathcal{N}(0, I).$$

We recall that the interpolation argument (10) for the proposed algorithm is given as

$$\boldsymbol{x}_t := \boldsymbol{x}_{k\gamma} - (t - k\gamma)\boldsymbol{g}_k + \sqrt{2}(\boldsymbol{B}_t - \boldsymbol{B}_{k\gamma}), \tag{33}$$

where $(\boldsymbol{B}_t)_{t \ge 0}$ is Brownian motion and $\boldsymbol{g}_k$ is the proposed variance-reduced ZO estimator. Using this interpolation with Lemma 2, we obtain

$$\begin{aligned}
\frac{d}{dt} \mathrm{KL}(\nu_t \| \pi) &\le -\frac{3}{4} \mathrm{FI}(\nu_t \| \pi) + \mathbb{E}[\|\nabla f(\boldsymbol{x}_t) - \boldsymbol{g}_k\|^2] \\
&= -\frac{3}{4} \mathrm{FI}(\nu_t \| \pi) + \mathbb{E}[\|\nabla f(\boldsymbol{x}_t) - \nabla f(\boldsymbol{x}_{k\gamma}) + \nabla f(\boldsymbol{x}_{k\gamma}) - \nabla f_\mu(\boldsymbol{x}_{k\gamma}) + \nabla f_\mu(\boldsymbol{x}_{k\gamma}) - \boldsymbol{g}_k\|^2] \\
&\le -\frac{3}{4} \mathrm{FI}(\nu_t \| \pi) + 3L_2^2 \mathbb{E}[\|\boldsymbol{x}_t - \boldsymbol{x}_{k\gamma}\|^2] + \frac{3}{4} \mu^2 L_2^2 (d + 3)^3 + 3\mathbb{E}[\|\nabla f_\mu(\boldsymbol{x}_{k\gamma}) - \boldsymbol{g}_k\|^2]. \tag{34}
\end{aligned}$$

We use Jensen's inequality, Lemma 1, and Assumption 2 to obtain the last inequality. Let $e_k := \mathbb{E}[\|\nabla f_\mu(\boldsymbol{x}_{k\gamma}) - \boldsymbol{g}_k\|^2]$ be the mean squared estimation error. Then, using Proposition 1, we have

$$e_{k+1}^2 \le \frac{p\sigma_{k+1}^2}{b} + (1 - p)e_k^2 + \frac{4(1 - p)dL_1\Delta_k}{\mu^2 b'}, \tag{35}$$

where $\Delta_k := \mathbb{E}[\|x_{(k+1)\gamma} - x_{k\gamma}\|^2]$ and $\sigma_{k+1}^2 := \mathbb{E}[\|\widetilde{\nabla} f_\mu(x_{(k+1)\gamma}, \boldsymbol{u}_{k+1}^i)\|^2]$. We can further upper bound $\sigma_{k+1}^2$ by using the definition of $\widetilde{\nabla} f_\mu(\boldsymbol{x}_{(k+1)\gamma}, \boldsymbol{u}_{k+1}^i)$ with the Assumption 1 as follows

$$\begin{aligned}
\sigma_{k+1}^2 &= \mathbb{E}[\|\widetilde{\nabla} f_\mu(x_{(k+1)\gamma}, \boldsymbol{u}_{k+1}^i)\|^2] \\
&\le L_1^2 \mathbb{E}[\|\boldsymbol{u}_{k+1}^i\|^4] \\
&= L_1^2 d(d + 2). \tag{36}
\end{aligned}$$

Plugging this into (35), we have

$$e_{k+1}^2 \le \frac{pd(d + 2)L_1^2}{b} + (1 - p)e_k^2 + \frac{4(1 - p)dL_1\Delta_k}{\mu^2 b'} \tag{37}$$

for $k \ge 1$. Dividing both sides by $p$ and rearranging the terms, we get a recursive upper bound on the error term

$$e_k^2 \le \frac{d(d + 2)L_1^2}{b} + \left(\frac{1 - p}{p}\right) \frac{4dL_1^2}{\mu^2 b'} \Delta_k - \frac{1}{p}(e_{k+1}^2 - e_k^2). \tag{38}$$

Using the interpolation in (10), we can upper bound $\mathbb{E}[\|\boldsymbol{x}_t - \boldsymbol{x}_{k\gamma}\|^2]$ in (34) as

$$\begin{aligned}
\mathbb{E}[\|\boldsymbol{x}_t - \boldsymbol{x}_{k\gamma}\|^2] &= (t - k\gamma)\mathbb{E}[\|\boldsymbol{g}_k\|^2] + 2\gamma d \\
&\le \gamma^2 \mathbb{E}[\|\boldsymbol{g}_k\|^2] + 2\gamma d = \Delta_k. \tag{39}
\end{aligned}$$

Using this property and plugging the upper bound in (38) into (34), we get

$$\frac{d}{dt} \mathrm{KL}(\nu_t \| \pi) \le -\frac{3}{4} \mathrm{FI}(\nu_t \| \pi) + 3L_m^2 \phi(\mu) \Delta_k + \frac{3d(d + 2)L_1^2}{b} - \frac{3}{p}(e_{k+1}^2 - e_k^2) + \frac{3}{4} \mu^2 L_2^2 (d + 3)^3, \tag{40}$$

where $\phi(\mu) := 1 + \left(\frac{1-p}{p}\right) \frac{4d}{\mu^2 b'}$. We can upper bound the term involving $\Delta_k$ as

$$
\begin{aligned}
\Delta_k &:= \mathbb{E}[\|\boldsymbol{x}_{(k+1)\gamma} - \boldsymbol{x}_{k\gamma}\|^2] \\
&= \gamma^2 \mathbb{E}[\|\boldsymbol{g}_k\|^2] + 2\gamma d \\
&= \gamma^2 \mathbb{E}[\|\boldsymbol{g}_k - \nabla f_\mu(\boldsymbol{x}_{k\gamma}) + \nabla f_\mu(\boldsymbol{x}_{k\gamma}) - \nabla f(\boldsymbol{x}_{k\gamma}) + \nabla f(\boldsymbol{x}_{k\gamma}) - \nabla f(\boldsymbol{x}_t) + \nabla f(\boldsymbol{x}_t)\|^2] + 2\gamma d \\
&= 4\gamma^2 L_2^2 \Delta_k + \gamma^2 \mu^2 L_2^2 (d+3)^3 + 4\gamma^2 e_k^2 + 4\gamma^2 \mathbb{E}[\|\nabla f(\boldsymbol{x}_t)\|^2] + 2\gamma d \\
&\leq 4\gamma^2 L_m^2 \phi(\mu) \Delta_k + \frac{4\gamma^2 d(d+2) L_1^2}{b} - \frac{4\gamma^2}{p}(e_{k+1}^2 - e_k^2) + \gamma^2 \mu^2 L_2^2 (d+3)^3 + 4\gamma^2 \mathbb{E}[\|\nabla f(\boldsymbol{x}_t)\|^2] + 2\gamma d.
\end{aligned}
$$

$$(41)$$
$$(42)$$
$$(43)$$

We use interpolation (10) to obtain (41). Following that, we use Assumption 2 together with (39), Jensen's inequality, and apply Lemma 1 to obtain (42). Finally, we use the upper bound in (38) with the fact that $L_2 \leq L_m$ to obtain (43). Rearranging the terms, we get

$$
\left(1 - 4\gamma^2 L_m^2 \phi(\mu)\right) \Delta_k \leq \frac{4\gamma^2 d(d+2) L_1^2}{b} - \frac{4\gamma^2}{p}(e_{k+1}^2 - e_k^2) + \gamma^2 \mu^2 L_2^2 (d+3)^3 + 4\gamma^2 \mathbb{E}[\|\nabla f(\boldsymbol{x}_t)\|^2] + 2\gamma d.
$$

Let $\gamma \leq \frac{1}{L_m \sqrt{52\phi(\mu)}}$ and multiplying both sides by $\frac{13}{12}$, we get

$$
\Delta_k \leq \frac{13}{3} \frac{\gamma^2 d(d+2) L_1^2}{b} - \frac{13}{3} \frac{\gamma^2}{p} \left(e_{k+1}^2 - e_k^2\right) + \frac{13}{12}\gamma^2 \mu^2 L_2^2 (d+3)^3 + \frac{13\gamma^2}{3} \mathbb{E}[\|\nabla f(\boldsymbol{x}_t)\|^2] + \frac{13}{6}\gamma d
$$

Plugging this upper bound into (40), we obtain

$$
\begin{aligned}
\frac{d}{dt}\mathrm{KL}(\nu_t\|\pi) &\leq -\frac{3}{4}\mathrm{FI}(\nu_t\|\pi) + \left[3 + 13\gamma^2 L_m^2 \phi(\mu)\right] \frac{d(d+2)}{b} - \frac{1}{p}\left[3 + 13\gamma^2 L_m^2 \phi(\mu)\right] (e_{k+1}^2 - e_k^2) \\
&\quad + \left[\frac{3}{4} + \frac{13}{4}\gamma^2 L_m^2 \phi(\mu)\right] \mu^2 L_2^2 (d+3)^3 + 13\gamma^2 L_m^2 \phi(\mu)\mathbb{E}[\|\nabla f(\boldsymbol{x}_t)\|^2] + \frac{13}{2}\gamma L_m^2 \phi(\mu)d \\
&\leq -\frac{1}{2}\mathrm{FI}(\nu_t\|\pi) + \frac{13d(d+2)}{4b} - \frac{1}{p}\left[3 + 13\gamma^2 L_m^2 \phi(\mu)\right] (e_{k+1}^2 - e_k^2) + \frac{13}{16}\mu^2 L_2^2 (d+3)^3 \\
&\quad + \left[\frac{13}{2} + 26\gamma L_m\right] \gamma L_m^2 \phi(\mu)d.
\end{aligned}
$$

$$(44)$$

We use the bound $\gamma \leq \frac{1}{L_m \sqrt{52\phi(\mu)}}$ and apply Lemma 3 to obtain (44). We define

$$
\mathcal{L}_k := \mathrm{KL}(\nu_t\|\pi) + \frac{\gamma}{p}\left(3 + 13\gamma^2 L_m^2 \phi(\mu)\right) e_k^2.
$$

Integrating both sides of (44) and using the definition $\mathcal{L}_k$, we get

$$
\mathcal{L}_{k+1} - \mathcal{L}_k \leq -\frac{1}{2}\int_{k\gamma}^{(k+1)\gamma} \mathrm{FI}(\nu_t\|\pi)\,dt + \frac{13}{4}\frac{\gamma d(d+2)}{b} + \frac{13}{16}\gamma\mu^2 L_2^2 (d+3)^3 + 7\gamma^2 L_m \phi(\mu)d. \tag{45}
$$

Iterating this inequality for $k \in \{0, 1, \dots, N-1\}$, and rearranging the terms, we obtain

$$
\frac{1}{N\gamma}\int_0^{N\gamma} \mathrm{FI}(\nu_t\|\pi)\,dt \leq \frac{2\mathcal{L}_0}{N\gamma} + \frac{13}{2}\frac{d(d+2)}{b} + \frac{13}{8}\mu^2 L_2^2 (d+3)^3 + 14\gamma L_m^2 \phi(\mu)d, \tag{46}
$$

where

$$
2\mathcal{L}_0 = 2\mathrm{KL}(\nu_0\|\pi) + \frac{2\gamma}{p}\left(3 + 13\gamma^2 L_m^2 \phi(\mu)\right) e_0^2 \leq \underbrace{2\mathrm{KL}(\nu_0\|\pi) + \frac{26\gamma}{4p}e_0^2}_{:=C_0} < \infty,
$$

where we use $\gamma \leq \frac{1}{L_m \sqrt{52\phi(\mu)}}$ and $C_0 > 0$ is a numerical constant. This proves the upper bound on Fisher information in the theorem statement. Furthermore, since $p \in (0, 1]$, we obtain

$$
\phi(\mu) = 1 + \frac{(1-p)4d}{p\mu^2 b'} \leq 1 + \frac{4d}{p\mu^2 b'} = 1 + \frac{4d^{5/2} N^{1/2}}{b'}.
$$

We can choose $N \geq \frac{b'}{4d^{5/2}}$ and obtain

$$\phi(\mu) \leq 1 + \frac{4d^{5/2}N^{1/2}}{b'} \leq \frac{8d^{5/2}N^{1/2}}{b'} \tag{47}$$

Plugging this upper bound into (46), we obtain

$$\frac{1}{N\gamma} \int_0^{N\gamma} \mathrm{FI}(\nu_t\|\pi)\,dt \leq \frac{C_0}{N\gamma} + \frac{13d(d+2)}{2b} + \frac{13}{8}\mu^2 L_2^2(d+3)^3 + \frac{112\gamma L_m^2 d^{7/2}N^{1/2}}{b'}.$$

Using the convexity of Fisher information, we obtain

$$\mathrm{FI}(\bar{\nu}_{N\gamma}\|\pi) \leq \frac{C_0}{N\gamma} + \frac{13d(d+2)}{2b} + \frac{13}{8}\mu^2 L_2^2(d+3)^3 + \frac{112\gamma L_m^2 d^2}{p\mu^2 b'}. \tag{48}$$

Let

$$\gamma = \frac{1}{L_m N^{3/4}d^{7/4}}, \quad p = \frac{L_m}{N^{1/4}d^{1/4}} \quad b = \left\lceil\frac{1}{p}\right\rceil, \quad \text{and} \quad \mu^2 = \frac{1}{L_m N^{1/4}d^{5/4}}.$$

Plugging these values into (48), we obtain

$$\mathrm{FI}(\bar{\nu}_{N\gamma}\|\pi) \leq \frac{C_0 L_m d^{7/4}}{N^{1/4}} + \frac{13}{2}\frac{L_m(d+2)^{3/2}}{N^{1/4}} + \frac{13}{8}\frac{L_m(d+3)^3}{N^{1/4}d^{5/4}} + \frac{112d^{7/4}L_m}{b'N^{1/4}}.$$

This implies $\mathrm{FI}(\bar{\nu}_{N\gamma}\|\pi) = \mathcal{O}\left(\frac{d^{7/4}L_m}{N^{1/4}}\right)$. Therefore, to obtain $\mathrm{FI}(\bar{\nu}_{N\gamma}\|\pi) \leq \varepsilon$, the sufficient number of iterations is

$$N = \mathcal{O}\left(\frac{d^7 L_m^4}{\varepsilon^4}\right).$$

Note that the per-iteration cost is $pb + (1-p)b' = \mathcal{O}(1)$, therefore, the total number of function evaluation is $\mathcal{O}\left(\frac{d^7 L_m^4}{\varepsilon^4}\right)$ as well.

It remains to verify that the iteration complexity $N = \mathcal{O}\left(\frac{d^7 L_m^4}{\varepsilon^4}\right)$ satisfies the conditions

$$N \geq \frac{b'}{4d^{5/2}}, \quad p \in (0,1], \quad \gamma \leq \frac{1}{L_m\sqrt{52\phi(\mu)}}.$$

The first condition holds since $b' = \mathcal{O}(1)$ is a numerical constant. The second condition requires

$$N \geq \frac{L_m^4}{d},$$

which is satisfied by the derived iteration complexity. Finally, using the upper bound in (47), we choose $N$ such that

$$\gamma \leq \frac{1}{L_m\sqrt{\frac{416d^{5/2}N^{1/2}}{b'}}} \leq \frac{1}{L_m\sqrt{52\phi(\mu)}},$$

which verifies the last condition. Substituting the definition of $\gamma$ and simplifying yields the additional requirement

$$N \geq \frac{416}{b'd},$$

which is again satisfied by the derived iteration complexity. This completes the proof.

$\square$

## B.4. Proof of Corollary 1

**Corollary 1** *Let $(\nu_t)_{t\geq 0}$ denote the law of interpolation (10), and let $\pi \propto \exp(-f)$ be the target distribution, where the potential function $f$ satisfies Assumptions 1, 2 with Lipschitz constants $L_1$, $L_2$, respectively, and Assumption 3 with constant $C_{\mathrm{PI}}$. Then, for any step size $\gamma \in \left(0, \frac{1}{L_m\sqrt{52\phi(\mu)}}\right]$, where $\phi(\mu) := 1 + \frac{4(1-p)d}{p\mu^2 b'}$ and $L_m := \max\{L_1, L_2\}$, and for any $N \geq 1$, we have*

$$\|\bar\nu_{N\gamma} - \pi\|^2_{\mathrm{TV}} \leq \frac{4C_{\mathrm{PI}}C_0}{N\gamma} + \frac{26C_{\mathrm{PI}}d(d+2)}{b} + \frac{13}{2}C_{\mathrm{PI}}\mu^2 L_2^2(d+3)^3 + 56C_{\mathrm{PI}}\gamma L_m^2\phi(\mu)d, \tag{49}$$

*where $C_0 > 0$ is a numerical constant. Furthermore, let*

$$\gamma = \frac{1}{L_m N^{3/4}d^{7/4}}, \quad p = \frac{L_m}{N^{1/4}d^{1/4}} \quad b = \left\lceil\frac{1}{p}\right\rceil, \quad \text{and} \quad \mu^2 = \frac{1}{L_m N^{1/4}d^{5/4}} \tag{50}$$

*Then, to achieve $\|\bar\nu_{N\gamma} - \pi\|^2_{\mathrm{TV}} \leq \varepsilon$ for the time averaged law $\bar\nu_{N\gamma} := \frac{1}{N\gamma}\int_0^{N\gamma}\nu_t\,dt$, $\mathcal{O}\left(\frac{d^7 L_m^4 C_{\mathrm{PI}4}}{\varepsilon^4}\right)$ iterations with $\mathcal{O}(1)$ function evaluations per iteration is sufficient.*

*Proof.* Recall from Theorem 1, we have

$$\frac{1}{N\gamma}\int_0^{N\gamma}\mathrm{FI}(\nu_t\|\pi)\,dt \leq \frac{C_0}{N\gamma} + \frac{13}{2}\frac{d(d+2)}{b} + \frac{13}{8}\mu^2 L_2^2(d+3)^3 + 14\gamma L_m^2\phi(\mu)d. \tag{51}$$

Since $\mathrm{FI}(\cdot\|\pi)$ is convex with respect to its first argument, Jensen's inequality yields

$$\mathrm{FI}(\bar\nu_{N\gamma}\|\pi) \leq \frac{C_0}{N\gamma} + \frac{13}{2}\frac{d(d+2)}{b} + \frac{13}{8}\mu^2 L_2^2(d+3)^3 + 14\gamma L_m^2\phi(\mu)d. \tag{52}$$

Using Assumption 3 and invoking Lemma 4, we obtain

$$\|\bar\nu_{N\gamma} - \pi\|^2_{\mathrm{TV}} \leq \frac{4C_{\mathrm{PI}}C_0}{N\gamma} + 26C_{\mathrm{PI}}\frac{d(d+2)}{b} + \frac{13}{2}C_{\mathrm{PI}}\mu^2 L_2^2(d+3)^3 + 56C_{\mathrm{PI}}\gamma L_m^2\phi(\mu)d. \tag{53}$$

This completes the proof of the upper bound in the corollary. The convergence rate then follows by an argument similar to that of Theorem 1. We can choose $N \geq \frac{b'}{4d^{5/2}}$ and using the fact that $p \in (0,1]$, we have

$$\phi(\mu) := 1 + \frac{(1-p)4d}{p\mu^2 b'} \leq \frac{8d}{p\mu^2 b'}, \tag{54}$$

we obtain

$$\|\bar\nu_{N\gamma} - \pi\|^2_{\mathrm{TV}} \leq \frac{4C_{\mathrm{PI}}C_0}{N\gamma} + 26C_{\mathrm{PI}}\frac{d(d+2)}{b} + \frac{13}{2}C_{\mathrm{PI}}\mu^2 L_2^2(d+3)^3 + 448C_{\mathrm{PI}}\frac{\gamma L_m^2 d^2}{p\mu^2 b'}. \tag{55}$$

Similar to the previous part, letting

$$\gamma = \frac{1}{L_m N^{3/4}d^{7/4}}, \quad p = \frac{L_m}{N^{1/4}d^{1/4}} \quad b = \left\lceil\frac{1}{p}\right\rceil, \quad \text{and} \quad \mu^2 = \frac{1}{L_m N^{1/4}d^{5/4}}, \tag{56}$$

we obtain that $\|\bar\nu_{N\gamma} - \pi\|^2_{\mathrm{TV}} \leq \mathcal{O}\left(\frac{C_{\mathrm{PI}}d^{7/4}L_m}{N^{1/4}}\right)$. Therefore, to obtain $\|\bar\nu_{N\gamma} - \pi\|^2_{\mathrm{TV}} \leq \varepsilon$, the sufficient number of iterations is

$$N = \mathcal{O}\left(\frac{C_{\mathrm{PI}}^4 d^7 L_m^4}{\varepsilon^4}\right). \tag{57}$$

Note that the per-iteration cost is $pb + (1-p)b' = \mathcal{O}(1)$; therefore, the number of function evaluations is $\mathcal{O}\left(\frac{C_{\mathrm{PI}}^4 d^7 L_m^4}{\varepsilon^4}\right)$ as well. The remaining conditions on $N$ can be verified by following the same arguments as in the proof of Theorem 1. $\quad\square$

## B.5. Proof of Theorem 2

**Theorem 2** *Let $(\nu_t)_{t\geq 0}$ denote the law of interpolation (10), and let $\pi \propto \exp(-f)$ be the target distribution, where the potential function $f$ satisfies Assumptions 1 and 2 with Lipschitz constants $L_1$ and $L_2$, respectively. Define the time-varying parameters as follows*

$$\gamma_k = \frac{C_\gamma}{k^{3/2}}, \quad p_k = \frac{1}{2k^{1/2}}, \quad b_k = \left\lceil \frac{1}{p_k} \right\rceil, \quad and \quad \mu_k = \frac{C_\mu}{k^{1/8}}, \quad \forall k \geq 1, \tag{58}$$

*where $L_m := \max\{L_1, L_2\}$, and $C_\gamma, C_\mu > 0$ are numerical constants. Then, the time averaged law $\bar{\nu}_{\tau_n} := \frac{1}{\tau_n} \int_0^{\tau_n} \nu_t \, dt$, where $\tau_n := \sum_{k=1}^n \gamma_k$, converges weakly to $\pi$.*

*Proof.* Given time-varying parameters $\gamma_k, b_k, p_k, \mu_k$ at iteration $k$, define the cumulative time $\tau_n$ and averaged law $\bar{\nu}_{\tau_n}$ at iteration $n$ as

$$\tau_n := \sum_{k=1}^n \gamma_k, \qquad \bar{\nu}_{\tau_n} := \frac{1}{\tau_n} \int_0^{\tau_n} \nu_t \, dt,$$

where $\nu_t$ denotes the law of the process $x_t$ under the following continuous-time interpolation

$$x_t := x_{\tau_n} - (t - \tau_n) g_n + \sqrt{2} \left( B_t - B_{\tau_n} \right), \qquad t \in [\tau_n, \tau_{n+1}]. \tag{59}$$

$g_n$ is defined as follows

$$g_n := \begin{cases} \dfrac{1}{b_n} \displaystyle\sum_{i=1}^{b_n} \widetilde{\nabla} f_{\mu_n}(x_{\tau_n}, u_n^i) & \text{w.p. } p_n, \\[2ex] g_{n-1} + \dfrac{1}{b'} \displaystyle\sum_{i=1}^{b'} \left( \widetilde{\nabla} f_{\mu_n}(x_{\tau_n}, u_n^i) - \widetilde{\nabla} f_{\mu_n}(x_{\tau_{n-1}}, u_n^i) \right) & \text{w.p. } 1 - p_n, \end{cases} \tag{60}$$

for all $n \geq 1$. At initialization, we choose $\gamma_0, b_0, \mu_0 > 0$ and $p_0 \in (0, 1]$, and define $g_0$ as

$$g_0 := \frac{1}{b_0} \sum_{i=1}^{b_0} \widetilde{\nabla} f_{\mu_0}(x_0, u_0^i), \quad u_0^i \sim \mathcal{N}(0, I).$$

We establish weak convergence by adapting the proof of Theorem 1. To this end, we verify that the step sizes $(\gamma_k)_{k\geq 1}$ satisfy the step-size condition of Theorem 1, namely,

$$\gamma_k \in \left( 0, \frac{1}{L_m \sqrt{52\phi_k(\mu_k)}} \right], \quad \text{where} \quad \phi_k(\mu_k) = 1 + \frac{4(1 - p_k)d}{p_k \mu_k^2 b'} \tag{61}$$

for all $k \geq 1$. Using the definitions of $\mu_k$ and $p_k$, we have

$$\begin{aligned} \phi_k(\mu_k) &= 1 + \frac{4(1 - p_k)d}{p_k \mu_k^2 b'} \\ &\leq 1 + \frac{4d}{p_k \mu_k^2 b'} \\ &\leq 1 + \frac{8dk^{3/4}}{b' C_\mu^2} \\ &\leq \frac{16dk^{3/4}}{b' C_\mu^2}, \end{aligned} \tag{62}$$

where the last inequality is satisfied by selecting a constant $C_\mu$ such that

$$\sqrt{\frac{8d}{b'}} \leq C_\mu.$$

Then, we have

$$\frac{1}{L_m\sqrt{52\phi_k(\mu_k)}} \geq \frac{1}{L_m\sqrt{\frac{832dk^{3/4}}{b'C_\mu^2}}} = \frac{C_\mu}{L_mk^{3/8}}\sqrt{\frac{b'}{832d}}.$$

Choosing

$$C_\gamma = \frac{C_\mu}{L_m}\sqrt{\frac{b'}{832d}},$$

we have

$$\gamma_k = \frac{C_\gamma}{k^{3/2}} \leq \frac{C_\mu}{L_mk^{3/8}}\sqrt{\frac{b'}{832d}} \leq \frac{1}{L_m\sqrt{52\phi_k(\mu_k)}},$$

which verifies (61). By definition of $p_k = \frac{1}{2k^{1/2}}$, we have $p_k \in (0,1]$ for $k \geq 1$. Therefore, we can follow the same steps in Theorem 1 up to (45) since the same arguments remain valid for $t \in [\tau_{n-1}, \tau_n]$ with time-varying parameters. We obtain

$$\mathrm{KL}(\nu_{\tau_n}\|\pi) - \mathrm{KL}(\nu_{\tau_{n-1}}\|\pi) \leq -\frac{1}{2}\int_{\tau_{n-1}}^{\tau_n}\mathrm{FI}(\nu_t\|\pi)dt + \frac{13\gamma_nd(d+2)L_1^2}{4b} + \frac{13}{16}\gamma_n\mu_n^2L_2^2(d+3)^3$$
$$+ 7\gamma_n^2dL_m^2\phi_n(\mu_n) - \frac{\gamma_n}{p_n}\left(3 + 13\gamma_n^2L_m^2\phi_n(\mu_n)\right)\left(e_n^2 - e_{n-1}^2\right) \quad (63)$$

for $t \in [\tau_{n-1}, \tau_n]$. Plugging the definitions of $\gamma_n$, $b_n$, $p_n$, and $\mu_n$ into (63), we get

$$\mathrm{KL}(\nu_{\tau_n}\|\pi) - \mathrm{KL}(\nu_{\tau_{n-1}}\|\pi) \leq -\frac{1}{2}\int_{\tau_{n-1}}^{\tau_n}\mathrm{FI}(\nu_t\|\pi)dt + \frac{A_1}{n^2} + \frac{A_2}{n^{7/4}} + \frac{A_3}{n^{9/4}} - c_n\left(e_n^2 - e_{n-1}^2\right), \quad (64)$$

where

$$A_1 := \frac{13d(d+2)L_1^2C_\gamma}{8}, \quad A_2 := \frac{13L_2^2(d+3)^3C_\gamma C_\mu^2}{16}, \quad A_3 := \frac{112d^2L_m^2C_\gamma^2}{b'C_\mu^2},$$

and

$$c_n := \frac{\gamma_n}{p_n}\left(3 + 13\gamma_n^2L_m^2\phi_n(\mu_n)\right)$$

for $n \geq 1$. Iterating the bound in (64), we obtain

$$\mathrm{KL}(\nu_{\tau_n}\|\pi) \leq \mathrm{KL}(\nu_0\|\pi) - \frac{1}{2}\int_0^{\tau_n}\mathrm{FI}(\nu_t\|\pi)dt + A_1S_1 + A_2S_2 + A_3S_3 - \sum_{k=1}^n c_k\left(e_k^2 - e_{k-1}^2\right) \quad (65)$$

where we have

$$\sum_{k=1}^n k^{-2} \leq \underbrace{\sum_{k=1}^\infty k^{-2}}_{:=S_1} < \infty, \quad \sum_{k=1}^n k^{-7/4} \leq \underbrace{\sum_{k=1}^\infty k^{-7/4}}_{:=S_2} < \infty.$$

$$\sum_{k=1}^n k^{-9/4} \leq \underbrace{\sum_{k=1}^\infty k^{-9/4}}_{:=S_3} < \infty.$$

Thus, $A_1S_1$, $A_2S_2$, and $A_3S_3$ are bounded constants and are independent of $n$. Furthermore, if we assume that $(c_n)_{n\geq 1}$ is nonnegative and decreasing sequence (i.e. $0 \leq c_{n+1} < c_n$), we can bound the summation in (65) as follows

$$-\sum_{k=1}^n c_k\left(e_k^2 - e_{k-1}^2\right) = c_1e_0^2 + \sum_{k=1}^{n-1}(c_{k+1} - c_k)e_k^2 - c_ne_n^2 \leq c_1e_0. \quad (66)$$

Thus, it remains to show that $(c_n)_{n\geq 1}$ is a nonnegative and decreasing sequence. Substituting the parameters $\gamma_n$, $b_n$, $p_n$, and $\mu_n$ into the definition of $c_n$, we obtain

$$c_n = \frac{6C_\gamma}{n} + \frac{6L_m^2C_\gamma^3}{n^4} + \frac{48dL_m^2C_\gamma^3}{b'C_\mu^2}\frac{1}{n^{13/4}}\left(1 - \frac{1}{2n^{1/2}}\right) \quad (67)$$

for $n \geq 1$. For notational convenience, define $F := L_m^2 C_\gamma^3$, which is a numerical constant, then we rewrite $c_n$ as

$$c_n = \frac{6C_\gamma}{n} + \frac{F}{n^4} + \frac{8Fd}{b'C_\mu^2 n^{13/4}}\left(1 - \frac{1}{2n^{1/2}}\right).$$

Nonnegativity is immediate since $1 - \frac{1}{2n^{1/2}} > 0$ for $n \geq 1$. Therefore, all the terms in $c_n$ are nonnegative for $n \geq 1$. To show $(c_n)_{n\geq1}$ is decreasing, define the continuous extension $c(x)$, for $x \geq 1$, as

$$c(x) = \frac{6C_\gamma}{x} + \frac{F}{x^4} + \frac{8Fd}{b'C_\mu^2 x^{13/4}}\left(1 - \frac{1}{2x^{1/2}}\right).$$

Taking the derivative yields

$$c'(x) = -\frac{6C_\gamma}{x^2} - \frac{4F}{x^5} - \frac{26Fd}{b'C_\mu^2 x^{17/4}}\left(1 - \frac{15}{26x^{1/2}}\right).$$

For $x \geq 1$, we have

$$1 - \frac{15}{26x^{1/2}} > 0,$$

and hence $c'(x) < 0$. Therefore, $c(x)$ is decreasing on $[1, \infty)$, which implies that $(c_n)_{n\geq1}$ is decreasing. Hence, using the upper bound (66) in (65) yields

$$\mathrm{KL}(\nu_{\tau_n}\|\pi) \leq \mathrm{KL}(\nu_0\|\pi) - \frac{1}{2}\int_0^{\tau_n} \mathrm{FI}(\nu_t\|\pi)dt + A_1 S_1 + A_2 S_2 + A_3 S_3 + c_1 e_0^2 \tag{68}$$

Rearranging the terms, using the convexity of Fisher information with Jensen's inequality, and multiplying both sides by $2/\tau_n$, we get

$$\begin{aligned}
\mathrm{FI}(\bar{\nu}_{\tau_n}\|\pi) &\leq \frac{1}{\tau_n}\int_0^{\tau_n} \mathrm{FI}(\nu_t\|\pi)\,dt \\
&\leq \frac{2\,\mathrm{KL}(\nu_0\|\pi)}{\tau_n} + \frac{2}{\tau_n}\left(A_1 S_1 + A_2 S_2 + A_3 S_3 + c_1 e_0^2\right),
\end{aligned} \tag{69}$$

where $A_1 S_1 + A_2 S_2 + A_3 S_3 < \infty$ and does not depend on $n$. On the other hand, if $t \in [\tau_n, \tau_{n+1}]$, integrating (44) between $\tau_n$ and $t$ and dropping the negative integral over the Fisher information give us

$$\begin{aligned}
\mathrm{KL}(\nu_t\|\pi) &\leq \mathrm{KL}(\nu_{\tau_n}\|\pi) + \frac{13(t - \tau_n)d(d+2)L_{f_1}^2}{4b} + \frac{13}{16}(t - \tau_n)\mu^2 L_{f_2}^2(d+3)^3 \\
&\quad + 7(t - \tau_n)\gamma dL_m^2\phi(\mu) - \frac{(t - \tau_n)}{p_{n+1}}\left(3 + 13\gamma_{n+1}^2 L_m^2\phi_{n+1}(\mu_{n+1})\right)(e_{n+1}^2 - e_n^2) \\
&\leq \mathrm{KL}(\nu_0\|\pi) + 2A_1 S_1 + 2A_2 S_2 + 2A_3 S_3 + 2c_1 e_0^2 + c_{n+1}e_n^2.
\end{aligned} \tag{70}$$

To obtain the second inequality, we bound all terms except the KL divergence, $c_1 e_0^2$, and $c_{n+1}e_n^2$ by their finite limits as $n \to \infty$. We then apply the bound on $\mathrm{KL}(\nu_{\tau_n}\|\pi)$ derived from (68). By Assumption 1 and Lemma 1, the initial error satisfies $e_0 < \infty$, and hence $c_1 e_0^2 < \infty$. Therefore, to show that $\{\mathrm{KL}(\nu_t\|\pi) \mid t \geq 0\}$ is bounded, it remains to prove that $c_{n+1}e_n^2 < \infty$ for all $n \geq 1$. Since $(c_n)_{n\geq1}$ is decreasing, it is uniformly bounded. Therefore, it remains to show that $e_n^2 < \infty$ for all $n \geq 1$. To this end, we apply Proposition 1 for time-varying parameters

$$e_{n+1}^2 \leq (1 - p_{n+1})e_n^2 + \frac{p_{n+1}\sigma_{n+1}^2}{b_{n+1}} + \frac{4(1 - p_{n+1})dL_1^2}{\mu_{n+1}^2 b'}\mathbb{E}[\|\boldsymbol{x}_{\tau_{n+1}} - \boldsymbol{x}_{\tau_n}\|^2], \tag{71}$$

where

$$\sigma_{n+1}^2 := \mathbb{E}[\|\widetilde{\nabla}f_\mu(\boldsymbol{x}_{\tau_{n+1}}, \boldsymbol{u}_{n+1}^i)\|^2].$$

The second term can be bounded using Assumption 1 together with Lemma 1, which imply that $\sigma_n^2 \leq C_\sigma$ for some constant $C_\sigma > 0$. Hence,

$$\frac{p_{n+1}\sigma_{n+1}^2}{b_{n+1}} \lesssim p_{n+1}^2, \tag{72}$$

where $a_n \lesssim b_n$ denotes that there exists a constant $C > 0$, independent of $n$, such that $a_n \leq C b_n$.

To bound the last term in (71), we use the interpolation (10) to obtain

$$\frac{4(1 - p_{n+1}) d L_1^2}{\mu_{n+1}^2 b'} \mathbb{E}[\|\boldsymbol{x}_{\tau_{n+1}} - \boldsymbol{x}_{\tau_n}\|^2] \lesssim \frac{1 - p_{n+1}}{\mu_{n+1}^2} \left( \gamma_{n+1}^2 \mathbb{E}[\|\boldsymbol{g}_n\|^2] + \gamma_{n+1} \right) \tag{73}$$

$$\lesssim \frac{1}{\mu_{n+1}^2} \left( \gamma_{n+1}^2 e_n^2 + \gamma_{n+1}^2 \mathbb{E}[\|\nabla f_{\mu_n}(\boldsymbol{x}_{\tau_n})\|^2] + \gamma_{n+1} \right) \tag{74}$$

$$\lesssim \frac{\gamma_{n+1}^2}{\mu_{n+1}^2} (e_n^2 + 1) + \frac{\gamma_{n+1}}{\mu_{n+1}^2} \tag{75}$$

$$\lesssim (n+1)^{-11/4}(e_n^2 + 1) + (n+1)^{-5/4} \tag{76}$$

$$\lesssim p_{n+1}^{11/2}(e_n^2 + 1) + p_{n+1}^{5/2}. \tag{77}$$

We add and subtract $\nabla f_{\mu_n}(\boldsymbol{x}_{\tau_n})$ inside the squared norm in (73), apply the inequality $(a + b)^2 \leq 2a^2 + 2b^2$ and use $p_{n+1} \in (0, 1]$ to obtain (74). Since Lipschitz continuity is preserved under Gaussian smoothing, $f_{\mu_n}$ is Lipschitz continuous whenever $f$ is Lipschitz continuous. Applying this property yields (75). Substituting the definitions of $\gamma_{n+1}$ and $\mu_{n+1}$ into (75) gives (76), and using the definition of $p_{n+1}$ yields the final inequality.

Combining (72) and (77), we obtain, for some constant $G > 0$,

$$e_{n+1}^2 \leq \left( 1 - p_{n+1} + G p_{n+1}^{11/2} \right) e_n^2 + G \left( p_{n+1}^2 + p_{n+1}^{11/2} + p_{n+1}^{5/2} \right). \tag{78}$$

Since $p_n \to 0$, there exists $n_0 \geq 1$ such that

$$G p_{n+1}^{9/2} \leq \frac{1}{2}$$

for all $n \geq n_0$. Moreover, since $p_{n+1} \in (0, 1]$,

$$p_{n+1}^2 + p_{n+1}^{11/2} + p_{n+1}^{5/2} \leq 3 p_{n+1}.$$

Therefore,

$$e_{n+1}^2 \leq \left( 1 - \frac{p_{n+1}}{2} \right) e_n^2 + 3 G p_{n+1}, \qquad n \geq n_0.$$

Defining $M \coloneqq 6G$, we obtain

$$e_{n+1}^2 \leq \left( 1 - \frac{p_{n+1}}{2} \right) e_n^2 + \frac{M p_{n+1}}{2}.$$

Since the right-hand side is a convex combination of $e_n^2$ and $M$,

$$e_{n+1}^2 \leq \max\{e_n^2, M\}, \qquad n \geq n_0.$$

By induction,

$$e_n^2 \leq \max\{e_{n_0}^2, M\}, \qquad n \geq n_0.$$

For the finitely many indices $1 \leq n \leq n_0$, finiteness of $e_n^2$ follows recursively from (78). Consequently,

$$e_n^2 \leq \max\{e_0^2, \ldots, e_{n_0}^2, M\} < \infty \tag{79}$$

for all $n \geq 1$. That proves $c_{n+1} e_n^2 < \infty$ for all $n \geq 1$.

Combining this with (70) implies that $\{\mathrm{KL}(\nu_t\|\pi) \mid t \geq 0\}$ is uniformly bounded. By the convexity of the KL divergence, this implies that the sequence $\{\mathrm{KL}(\bar{\nu}_{\tau_n}\|\pi)\}_{n\in\mathbb{N}}$ is uniformly bounded as well. Since the sublevel sets of $\mathrm{KL}(\cdot\|\pi)$ are weakly compact, $(\bar{\nu}_{\tau_n})_{n\in\mathbb{N}}$ is tight. To establish that $\bar{\nu}_{\tau_n} \rightharpoonup \pi$ weakly, it suffices to verify that every cluster point of $(\bar{\nu}_{\tau_n})_{n\in\mathbb{N}}$ equal to $\pi$.

Consider a subsequence of $(\bar{\nu}_{\tau_n})_{n\in\mathbb{N}}$ converging to some limit $\bar{\nu}$. Taking $n \to \infty$ in (69) and noting that $\tau_n \to \infty$, we obtain $\mathrm{FI}(\bar{\nu}_{\tau_n}\|\pi) \to 0$. Therefore, the same holds along the subsequence. By the weak lower semicontinuity of the Fisher information along the subsequence, we have $\mathrm{FI}(\bar{\nu}\|\pi) = 0$. Writing $\psi := \frac{d\bar{\nu}}{d\pi}$, this means $\sqrt{\psi} \in \mathrm{dom}\, \mathcal{E}$ and $\mathcal{E}(\sqrt{\psi}) = 0$, where $\mathcal{E}$ denotes the Dirichlet energy (i.e., the squared $L^2(\pi)$-norm of the gradient; see Section 3 in Balasubramanian et al. (2022)). Since $\nabla \log \pi$ is Lipschitz by Assumption 2, $\pi$ has a continuous and strictly positive density on $\mathbb{R}^d$, so $\mathcal{E}(\sqrt{\psi}) = 0$, which implies that $\psi$ must be a constant $\pi$-a.e., hence $\bar{\nu} = \pi$. $\qquad\square$

## B.6. Proof of Theorem 3

**Theorem 3** *Let $\pi \propto \ell(\boldsymbol{y}|\boldsymbol{x})p(\boldsymbol{x})$ be the posterior with the likelihood $\ell(\boldsymbol{y}|\boldsymbol{x}) \propto \exp(-f(\boldsymbol{x}))$ and the prior $p(\boldsymbol{x}) \propto \exp(-h(\boldsymbol{x}))$. Suppose the likelihood potential $f$ satisfies Assumptions 1, 2 with Lipschitz constants $L_{f_1}$, $L_{f_2}$, respectively, the prior potential $h$ satisfies Assumption 2 with Lipschitz constant $L_h$ and Assumption 4, and the SGM satisfies Assumption 5 with decreasing error $\varepsilon_{\sigma_k} = \mathcal{O}(k^{-1/2})$ for $\sigma_k > 0$, and $k \geq 1$. Define $L_m := \max\{L_{f_2} + L_h, L_{f_1}\}$. Let $(\nu_t)_{t\geq 0}$ denote the law of interpolation generated by (14) with annealing and noise schedules defined in (13). For any step size $\gamma \in \left(0, \frac{1}{L_m\sqrt{85\phi(\mu)}}\right]$, where $\phi(\mu) := 1 + \frac{4(1-p)d}{p\mu^2 b'}$, and for any $N \geq 1$, the Fisher information satisfies*

$$\frac{1}{N\gamma}\int_0^{N\gamma} \mathrm{FI}(\nu_t\|\pi)\, dt \leq \frac{C_0}{N\gamma} + \frac{17d(d+2)L_{f_1}^2}{2b} + \frac{17\mu^2 L_{f_2}^2(d+3)^3}{8} + \bar{\sigma}^2 + \bar{\varepsilon}_\sigma^2 + \bar{\alpha}^2 + 32\gamma L_m^2 d\phi(\mu), \tag{80}$$

*where*

$$\bar{\sigma}^2 := \frac{51C_1^2}{2N}\sum_{k=0}^{N-1}\sigma_k^2, \quad \bar{\varepsilon}_\sigma^2 := \frac{51}{2N}\sum_{k=0}^{N-1}\varepsilon_{\sigma_k}^2, \quad \text{and} \quad \bar{\alpha}^2 := \frac{51C_2^2}{2N}\sum_{k=0}^{N-1}\frac{(\alpha_k-1)^2}{\sigma_k^2},$$

*and $C_0$, $C_1$, $C_2$ are positive constants. Furthermore, let*

$$\gamma = \frac{1}{L_m N^{3/4}d^{7/4}}, \quad p = \frac{L_m}{N^{1/4}d^{1/4}} \quad b = \left\lceil \frac{1}{p}\right\rceil, \quad \text{and} \quad \mu^2 = \frac{1}{L_m N^{1/4}d^{5/4}},$$

*and let the initial parameters satisfy $\sigma_0 \geq \sigma_{min}$, $\varepsilon_{\sigma_0} > 0$, $\alpha_0 \geq 1$, and $\sigma_{min} > 0$ is the minimum noise level. Then, to achieve $\mathrm{FI}(\bar{\nu}_{N\gamma}\|\pi) \leq \varepsilon$ for the time averaged law $\bar{\nu}_{N\gamma} := \frac{1}{N\gamma}\int_0^{N\gamma}\nu_t\, dt$, $\mathcal{O}\left(\frac{d^7 L_m^4}{\varepsilon^4}\right)$ iterations with $\mathcal{O}(1)$ function evaluations per iteration is sufficient.*

*Proof.* At initialization, we choose $\gamma_0, b_0, \mu_0 > 0$ and $p_0 \in (0,1]$, and define $\boldsymbol{g}_0$ as

$$\boldsymbol{g}_0 := \frac{1}{b_0}\sum_{i=1}^{b_0}\widetilde{\nabla}f_{\mu_0}(\boldsymbol{x}_0, \boldsymbol{u}_0^i), \quad \boldsymbol{u}_0^i \sim \mathcal{N}(0, I).$$

We recall the interpolation argument in (14) given as

$$\boldsymbol{x}_t := \boldsymbol{x}_{k\gamma} - (t - k\gamma)(\boldsymbol{g}_k - \alpha_k\mathcal{S}_\theta(\boldsymbol{x}_{k\gamma})) + \sqrt{2}(\boldsymbol{B}_t - \boldsymbol{B}_{k\gamma}) \quad \text{for} \quad t \in [k\gamma, (k+1)\gamma], \tag{81}$$

where $\boldsymbol{g}_k$ denotes variance-reduced zeroth-order estimate of the negative likelihood score $\nabla f(\boldsymbol{x}_{k\gamma})$ at iteration $k$, and $(\alpha_k)_{k=0}^{N-1}$ and $(\sigma_k)_{k=0}^{N-1}$ denote annealing and noise schedules, respectively. By Assumption 2, 4 and triangle inequality, we know that the posterior score function $\nabla\log\pi(\boldsymbol{x})$ is Lipschitz continuous with Lipschitz constant $L_p + L_{f_2}$. Furthermore, by Assumptions 4 and 5, the error between the negative prior score and and its estimate scaled by annealing parameter can be bounded by

$$\|\nabla h(\boldsymbol{x}_{k\gamma}) + \alpha_k\mathcal{S}_\theta(\boldsymbol{x}_{k\gamma})\| \leq \|\nabla h(\boldsymbol{x}_{k\gamma}) - \nabla h_{\sigma_k}(\boldsymbol{x}_{k\gamma}) + \nabla h_{\sigma_k}(\boldsymbol{x}_{k\gamma}) + \mathcal{S}_\theta(\boldsymbol{x}_{k\gamma}, \sigma_k) + (\alpha_k - 1)\mathcal{S}_\theta(\boldsymbol{x}_{k\gamma})\|$$
$$\leq \sigma_k C + \varepsilon_{\sigma_k} + (\alpha_k - 1)\sigma_k^{-1}C. \tag{82}$$

Recall that $\mathcal{S}_\theta(\boldsymbol{x}_{k\gamma}, \sigma_k)$ estimates the perturbed score $-\nabla h_{\sigma_k}(\boldsymbol{x}_{k\gamma})$. Applying the triangle inequality yields (82). We are now ready to prove the theorem.

Combining Lemma 2 with the interpolation argument in (81), it follows that for every $t \in [k\gamma, (k+1)\gamma]$,

$$\frac{d}{dt}\mathrm{KL}(\nu_t\|\pi) \leq -\frac{3}{4}\mathrm{FI}(\nu_t\|\pi) + \mathbb{E}\left[\|\nabla\log\pi(\boldsymbol{x}_t) + \boldsymbol{g}_k - \alpha_k\mathcal{S}_\theta(\boldsymbol{x}_{k\gamma}, \sigma_k)\|^2\right].$$

Adding and subtracting $\nabla f_\mu(\boldsymbol{x}_t)$, $\nabla f_\mu(\boldsymbol{x}_{k\gamma})$, $\nabla h(\boldsymbol{x}_{k\gamma})$, and $\nabla h_{\sigma_k}(\boldsymbol{x}_{k\gamma})$ inside the squared norm in expectation, and applying the inequality $(a + b + c + d)^2 \leq 4a^2 + 4b^2 + 4c^2 + 4d^2$ together with (82) and Lemma 1 yields

$$\frac{d}{dt}\mathrm{KL}(\nu_t\|\pi) \leq -\frac{3}{4}\mathrm{FI}(\nu_t\|\pi) + 4\mathbb{E}\left[\|\boldsymbol{g}_k - \nabla f_\mu(\boldsymbol{x}_{k\gamma})\|^2\right] + 4L_\pi^2\mathbb{E}\left[\|\boldsymbol{x}_t - \boldsymbol{x}_{k\gamma}\|^2\right]$$
$$+ \mu^2 L_{f_2}^2(d+3)^3 + 4(C_1\sigma_k + \varepsilon_{\sigma_k} + (\alpha_k - 1)C_2\sigma_k^{-1})^2, \tag{83}$$

where $L_\pi := L_{f_2} + L_h$. Using Proposition 1, and subsequently applying steps (36), (37), and (38), we can upper bound the mean squared estimation error $e_k^2 := \mathbb{E}[\|\boldsymbol{g}_k - \nabla f_\mu(\boldsymbol{x}_{k\gamma})\|^2]$ as follows

$$e_k^2 \leq \frac{d(d+2)L_{f_1}^2}{b} + \left(\frac{1-p}{p}\right)\frac{4dL_{f_1}^2}{\mu^2 b'}\Delta_k - \frac{1}{p}(e_{k+1}^2 - e_k^2) \tag{84}$$

Plugging this upper bound into (83), we get

$$\frac{d}{dt}\mathrm{KL}(\nu_t\|\pi) \leq -\frac{3}{4}\mathrm{FI}(\nu_t\|\pi) + 4L_\pi^2\mathbb{E}\left[\|\boldsymbol{x}_t - \boldsymbol{x}_{k\gamma}\|^2\right] + \mu^2 L_{f_2}^2(d+3)^2 + \frac{4d(d+2)L_{f_1}^2}{b}$$
$$+ \left(\frac{1-p}{p}\right)\frac{16dL_{f_1}^2}{\mu^2 b'}\Delta_k - \frac{4}{p}(e_{k+1}^2 - e_k^2) + 4(C_1\sigma_k + \varepsilon_{\sigma_k} + (\alpha_k - 1)C_2\sigma_k^{-1})^2 \tag{85}$$

By the interpolation argument in (81), we have

$$\mathbb{E}[\|\boldsymbol{x}_t - \boldsymbol{x}_{k\gamma}\|^2] = (t-k\gamma)^2\mathbb{E}\left[\|\boldsymbol{g}_k - \alpha_k\mathcal{S}_\theta(\boldsymbol{x}_{k\gamma})\|^2\right] + 2(t-k\gamma)d$$
$$\leq \gamma^2\mathbb{E}\left[\|\boldsymbol{g}_k - \alpha_k\mathcal{S}_\theta(\boldsymbol{x}_{k\gamma})\|^2\right] + 2\gamma d$$
$$= \mathbb{E}[\|\boldsymbol{x}_{(k+1)\gamma} - \boldsymbol{x}_{k\gamma}\|^2]$$
$$= \Delta_k \tag{86}$$

for $t \in [k\gamma, (k+1)\gamma]$. Substituting the bound in (86) into (85) yields

$$\frac{d}{dt}\mathrm{KL}(\nu_t\|\pi) \leq -\frac{3}{4}\mathrm{FI}(\nu_t\|\pi) + 4L_m^2\underbrace{\left[1 + \left(\frac{1-p}{p}\right)\frac{4d}{\mu^2 b'}\right]}_{:=\phi(\mu)}\Delta_k + \mu^2 L_{f_2}^2(d+3)^2 + \frac{4d(d+2)L_{f_1}^2}{b}$$
$$-\frac{4}{p}(e_{k+1}^2 - e_k^2) + 4(C_1\sigma_k + \varepsilon_{\sigma_k} + (\alpha_k - 1)C_2\sigma_k^{-1})^2 \tag{87}$$

where $L_m := \max\{L_{f_1}, L_\pi\}$. We next use the interpolation (81) to derive an upper bound on $\Delta_k$:

$$\Delta_k = \gamma^2\mathbb{E}[\|\boldsymbol{g}_k - \alpha_k\mathcal{S}_\theta(\boldsymbol{x}_{k\gamma}, \sigma_k)\|^2] + 2\gamma d$$
$$\leq 5\gamma^2 e_k^2 + \frac{5\gamma^2\mu^2 L_{f_2}^2(d+3)^3}{4} + 5\gamma^2 L_\pi^2\Delta_k + 5\gamma^2\mathbb{E}\left[\|\nabla\log\pi(\boldsymbol{x}_t)\|^2\right]$$
$$+ 5\gamma^2(C_1\sigma_k + \varepsilon_{\sigma_k} + (\alpha_k - 1)C_2\sigma_k^{-1})^2 + 2\gamma d \tag{88}$$

where we add and subtract $\nabla f_\mu(\mathbf{x}_{k\gamma})$, $\nabla f(\mathbf{x}_{k\gamma})$, $\nabla f(\mathbf{x}_t)$, $\nabla h(\mathbf{x}_t)$, $\nabla h(\mathbf{x}_{k\gamma})$, and $\nabla h_{\sigma_k}(\mathbf{x}_{k\gamma})$ inside the squared norm in expectation, and then apply the convexity of the $\ell_2$ norm together with the part (b) of Assumption 5 to obtain (88). Using the bound on $e_k^2$ in (84), we get

$$\Delta_k \leq 5\gamma^2 L_m^2\phi(\mu)\Delta_k + \frac{5\gamma^2 d(d+2)L_{f_1}^2}{b} + \frac{5\gamma^2\mu^2 L_{f_2}^2(d+3)^3}{4} + 5\gamma^2\mathbb{E}\left[\|\nabla\log\pi(\boldsymbol{x}_t)\|^2\right] - \frac{5\gamma^2}{p}\left(e_{k+1}^2 - e_k^2\right)$$
$$+ 5\gamma^2(\sigma_k C + \varepsilon_{\sigma_k} + (\alpha_k - 1)\sigma_k^{-1}C)^2 + 2\gamma d$$

Assume that $\gamma \leq \frac{1}{L_m\sqrt{85\phi(\mu)}}$, then rearranging the terms, we have

$$\frac{16}{17}\Delta_k \leq \frac{5\gamma^2 d(d+2)L_{f_1}^2}{b} + \frac{5\gamma^2\mu^2 L_{f_2}^2(d+3)^3}{4} + 5\gamma^2\mathbb{E}\left[\|\nabla\log\pi(\boldsymbol{x}_t)\|^2\right] - \frac{5\gamma^2}{p}\left(e_{k+1}^2 - e_k^2\right)$$
$$+ 5\gamma^2(C_1\sigma_k + \varepsilon_{\sigma_k} + (\alpha_k - 1)C_2\sigma_k^{-1})^2 + 2\gamma d$$

Multiplying both sides by $\frac{17}{16}$, we get

$$\Delta_k \leq \frac{85\gamma^2 d(d+2)L_{f_1}^2}{16b} + \frac{85\gamma^2\mu^2 L_{f_2}^2(d+3)^3}{64} + \frac{85}{16}\gamma^2\mathbb{E}\left[\|\nabla\log\pi(\boldsymbol{x}_t)\|^2\right] - \frac{85\gamma^2}{16p}\left(e_{k+1}^2 - e_k^2\right)$$
$$+ \frac{85}{16}\gamma^2(C_1\sigma_k + \varepsilon_{\sigma_k} + (\alpha_k - 1)C_2\sigma_k^{-1})^2 + \frac{17}{8}\gamma d$$

We can use Lemma 3 to put an upper bound on $\mathbb{E}[\|\nabla \log \pi(\boldsymbol{x}_t)\|^2]$

$$\Delta_k \leq \frac{85\gamma^2 d(d+2)L_{f_1}^2}{16b} + \frac{85\gamma^2\mu^2 L_{f_2}^2(d+3)^3}{64} + \frac{85}{16}\gamma^2 \mathrm{FI}(\nu_t\|\pi) + \frac{85}{8}\gamma^2 L_\pi d - \frac{85\gamma^2}{16p}\left(e_{k+1}^2 - e_k^2\right)$$
$$+ \frac{85}{16}\gamma^2(C_1\sigma_k + \varepsilon_{\sigma_k}^2 + (\alpha_k - 1)C_2\sigma_k^{-1})^2 + \frac{17}{8}\gamma d. \tag{89}$$

We can combine the terms $\frac{85}{8}\gamma^2 L_\pi d$ and $\frac{17}{8}\gamma d$ using the assumption $\gamma \leq \frac{1}{L_m\sqrt{85\phi(\mu)}}$. In particular,

$$\gamma \leq \frac{1}{\sqrt{85L_m^2\left(1 + \left(\frac{1-p}{p}\right)\frac{4d}{\mu^2 b'}\right)}} \leq \frac{1}{L_m\sqrt{85}},$$

where we use the fact that $\left(\frac{1-p}{p}\right)\frac{8d}{\mu^2 b'} > 0$. Consequently,

$$\frac{85}{8}\gamma^2 dL_\pi \leq \frac{\sqrt{85}\gamma dL_\pi}{8L_m} \leq \frac{\sqrt{85}}{8}\gamma d,$$

where we use $L_\pi \leq L_m$. Therefore,

$$\frac{85}{8}\gamma^2 L_\pi d + \frac{17}{8}\gamma d \leq \left(\frac{\sqrt{85}}{8} + \frac{17}{8}\right)\gamma d$$
$$\leq 4\gamma d.$$

Substituting this bound into (89) yields the simplified bound

$$\Delta_k \leq \frac{85\gamma^2 d(d+2)L_{f_1}^2}{16b} + \frac{85\gamma^2\mu^2 L_{f_2}^2(d+3)^3}{64} + \frac{85}{16}\gamma^2 \mathrm{FI}(\nu_t\|\pi) - \frac{85\gamma^2}{16p}\left(e_{k+1}^2 - e_k^2\right)$$
$$+ \frac{85}{16}\gamma^2(C_1\sigma_k + \varepsilon_{\sigma_k}^2 + (\alpha_k - 1)C_2\sigma_k^{-1})^2 + 4\gamma d. \tag{90}$$

Plugging (90) into (87), we get

$$\frac{d}{dt}\mathrm{KL}(\nu_t\|\pi) \leq \left(-\frac{3}{4} + \frac{85\gamma^2 L_m^2\phi(\mu)^2}{4}\right)\mathrm{FI}(\nu_t\|\pi) + \left(1 + \frac{85\gamma^2 L_m^2\phi(\mu)}{16}\right)\frac{4d(d+2)L_{f_1}^2}{b}$$
$$+ \left(1 + \frac{85\gamma^2 L_m^2\phi(\mu)}{16}\right)\mu^2 L_{f_2}^2(d+3)^3 - \frac{4}{p}\left(1 + \frac{85\gamma^2 L_m^2\phi(\mu)}{16}\right)\left(e_{k+1}^2 - e_k^2\right)$$
$$+ 4\left(1 + \frac{85\gamma^2 L_m^2\phi(\mu)}{16}\right)(\sigma_k C + \varepsilon_{\sigma_k} + (\alpha_k - 1)\sigma_k^{-1}C)^2 + 16\gamma dL_m^2\phi(\mu)$$
$$\leq -\frac{1}{2}\mathrm{FI}(\nu_t\|\pi) + \frac{17d(d+2)L_{f_1}^2}{4b} + \frac{17}{16}\mu^2 L_{f_2}^2(d+3)^3 - \frac{4}{p}\left(1 + \frac{85\gamma^2 L_m^2\phi(\mu)}{16}\right)\left(e_{k+1}^2 - e_k^2\right)$$
$$+ \frac{17}{4}(C_1\sigma_k + \varepsilon_{\sigma_k} + (\alpha_k - 1)C_2\sigma_k^{-1})^2 + 16\gamma dL_m^2\phi(\mu), \tag{91}$$

where we use the fact that $85\gamma^2 L_m^2\phi(\mu) \leq 1$ to get (91). Integrating both sides between $[k\gamma, (k+1)\gamma]$, we get

$$\mathrm{KL}(\nu_{(k+1)\gamma}\|\pi) - \mathrm{KL}(\nu_{k\gamma}\|\pi) \leq -\frac{1}{2}\int_{k\gamma}^{(k+1)\gamma}\mathrm{FI}(\nu_t\|\pi)dt + \frac{17\gamma d(d+2)L_{f_1}^2}{4b} + \frac{17}{16}\gamma\mu^2 L_{f_2}^2(d+3)^3$$
$$- \frac{4\gamma}{p}\left(1 + \frac{85\gamma^2 L_m^2\phi(\mu)}{16}\right)\left(e_{k+1}^2 - e_k^2\right)$$
$$+ \frac{51\gamma}{4}(C_1^2\sigma_k^2 + \varepsilon_{\sigma_k}^2 + (\alpha_k - 1)^2 C_2^2\sigma_k^{-2}) + 16\gamma^2 dL_m^2\phi(\mu), \tag{92}$$

where we use the inequality $(a + b + c)^2 \leq 3a^2 + 3b^2 + 3c^2$ to get the last term. Let $\mathcal{L}_k := \mathrm{KL}(\nu_{k\gamma}\|\pi) + \frac{4\gamma}{p}\left[1 + \frac{85}{16}\gamma^2 L_m^2\phi(\mu)\right]e_k^2$, iterating for $k = 0, \ldots, N-1$, multiplying both sides by $\frac{2}{N\gamma}$, rearranging the terms and using the fact that $\mathcal{L}_N \geq 0$, we get

$$\frac{1}{N\gamma}\int_0^{N\gamma}\mathrm{FI}(\nu_t\|\pi)dt \leq \frac{2\mathcal{L}_0}{N\gamma} + \frac{17d(d+2)L_{f_1}^2}{2b} + \frac{17}{8}\mu^2 L_{f_2}^2(d+3)^3 + \bar{\sigma}^2 + \bar{\varepsilon}_\sigma^2 + \bar{\alpha}^2 + 32\gamma L_m^2 d\phi(\mu), \quad (93)$$

where $\bar{\sigma}^2 := \frac{51C_1^2}{2N}\sum_{k=0}^{N-1}\sigma_k^2$, $\bar{\varepsilon}_\sigma^2 := \frac{51}{2N}\sum_{k=0}^{N-1}\varepsilon_{\sigma_k}^2$, and $\bar{\alpha} := \frac{51C_2^2}{2N}\sum_{k=0}^{N-1}\frac{(\alpha_k-1)^2}{\sigma_k^2}$. Since $85\gamma^2 L_m^2\phi(\mu) \leq 1$, we have

$$\mathcal{L}_0 = \mathrm{KL}(\nu_0\|\pi) + \frac{4\gamma}{p}\left(1 + \frac{85\gamma^2 L_m^2\phi(\mu)}{16}\right)e_0^2 \leq \mathrm{KL}(\nu_0\|\pi) + \frac{17\gamma e_0^2}{4p}.$$

We can define $C_0 := 2\mathrm{KL}(\nu_0\|\pi) + \frac{17\gamma e_0^2}{2p}$ which is a numerical constant. This completes the proof of the Fisher information upper bound.

To establish the iteration complexity, we observe that all terms in the upper bound of Theorem 1, except for the bias induced by the score network approximation error $\bar{\varepsilon}_\sigma^2$, and the bias due to the annealing and noise schedules $\bar{\alpha}^2$ and $\bar{\sigma}^2$, respectively, appear with different coefficients. Therefore, choosing the same parameter values as

$$\gamma = \frac{1}{L_m N^{3/4}d^{7/4}}, \quad p = \frac{L_m}{N^{1/4}d^{1/4}} \quad b = \left\lceil\frac{1}{p}\right\rceil, \quad \text{and} \quad \mu^2 = \frac{1}{L_m N^{1/4}d^{5/4}},$$

we obtain the following result

$$\frac{1}{N\gamma}\int_0^{N\gamma}\mathrm{FI}(\nu_t\|\pi)dt \leq \mathcal{O}\left(\frac{d^{7/4}L_m}{N^{1/4}}\right) + \bar{\sigma}^2 + \bar{\varepsilon}_\sigma^2 + \bar{\alpha}^2. \quad (94)$$

We recall that the noise $(\sigma_k)_{k=0}^{N-1}$ and annealing $(\alpha_k)_{k=0}^{N-1}$ schedules are defined as

$$\alpha_k = \max\{\alpha_0\rho_1^k, 1\} \quad \text{and} \quad \sigma_k = \max\{\sigma_0\rho_2^k, \sigma_{\min}\}, \quad (95)$$

where $\rho_1, \rho_2 \in (0, 1)$ denote decay rates, $\sigma_0 > 0$, $\alpha_0 > 0$, and $\sigma_{\min} > 0$ is the minimum noise level. By definition in (13), there exist indices $K_\alpha < N-1$ and $K_\sigma < N-1$ independent of $N$ such that $\alpha_k = 1$ for $\forall k \geq K_\alpha$ and $\sigma_k = \sigma_{\min}$ for $\forall k \geq K_\sigma$.

We next analyze the convergence behavior of the bias contributions due to the noise schedule, annealing schedule, and score network error separately.

To bound $\bar{\sigma}^2$, we proceed as

$$\bar{\sigma}^2 = \frac{51C_1^2}{2N}\sum_{k=0}^{N-1}\max\{\sigma_0^2\rho_2^{2k}, \sigma_{\min}^2\}$$

$$\stackrel{(*)}{=} \frac{51C_1^2}{2N}\sum_{k=0}^{K_\sigma-1}\sigma_0^2\rho_2^{2k} + \frac{51C_1^2}{2N}\sum_{k=K_\sigma}^{N-1}\sigma_{\min}^2$$

$$= \underbrace{\frac{51C_1^2\sigma_0^2(1-\rho_2^{2K_\sigma})}{2N(1-\rho_2^2)}}_{\mathcal{O}(N^{-1})} + \frac{51C_1^2(N-K_\sigma)\sigma_{\min}^2}{2N}. \quad (96)$$

where $(*)$ follows from the definition of the annealing and noise schedules (13). Setting

$$\sigma_{\min} = \mathcal{O}(N^{-1/8}),$$

we obtain

$$\bar{\sigma}^2 = \mathcal{O}(N^{-1}) + \mathcal{O}(N^{-1/4}) = \mathcal{O}(N^{-1/4}). \quad (97)$$

To bound $\bar{\alpha}^2$, we use that $(\sigma_k)_{k=0}^{N-1}$ is a nonincreasing sequence and that there exists a constant $K_\alpha < N - 1$, independent of $N$, such that $\alpha_k = 1$ for all $k \geq K_\alpha$. Thus, we obtain

$$
\begin{aligned}
\bar{\alpha}^2 &= \frac{51 C_2^2}{2N} \sum_{k=0}^{N-1} \frac{(\alpha_k - 1)^2}{\sigma_k^2} \\
&= \frac{51 C_2^2}{2N} \sum_{k=0}^{K_\alpha - 1} \frac{(\alpha_0 \rho_1^k - 1)^2}{\sigma_k^2} \\
&\leq \frac{51 C_2^2}{2N\sigma_{\min}^2} \sum_{k=0}^{K_\alpha - 1} (\alpha_0 \rho_1^k - 1)^2 \\
&\leq \frac{51 C_2^2}{2N\sigma_{\min}^2} \sum_{k=0}^{K_\alpha - 1} (\alpha_0^2 \rho_1^{2k} - 2\alpha_0 \rho_1^k + 1) \\
&\leq \frac{51 C_2^2 \alpha_0^2}{2N\sigma_{\min}^2} \sum_{k=0}^{K_\alpha - 1} \rho_1^{2k} + \frac{51 C_2^2 K_\alpha}{2N\sigma_{\min}^2} \\
&= \underbrace{\frac{51 C_2^2 \alpha_0^2 (1 - \rho_1^{2K_\alpha})}{2N\sigma_{\min}^2 (1 - \rho_1^2)}}_{\mathcal{O}(N^{-3/4})} + \underbrace{\frac{51 C_2^2 K_\alpha}{2N\sigma_{\min}^2}}_{\mathcal{O}(N^{-3/4})} \\
&= \mathcal{O}\left(N^{-3/4}\right)
\end{aligned}
\tag{98}
$$

Finally, to bound $\bar{\varepsilon}_\sigma^2$, under the assumption that the SGM estimation error decays as $\varepsilon_{\sigma_k}^2 \leq \frac{C'}{k}$ for some constant $C' > 0$, we proceed as

$$
\bar{\varepsilon}_\sigma^2 = \frac{51}{2N} \sum_{k=0}^{N-1} \varepsilon_{\sigma_k}^2 \leq \frac{51 \varepsilon_{\sigma_0}^2}{2N} + \frac{51 C'}{2N} \sum_{k=1}^{N-1} \frac{1}{k} = \mathcal{O}\left(\frac{\log N}{N}\right)
\tag{99}
$$

Combining the results (97), (98), and (99), we obtain

$$
\begin{aligned}
\frac{1}{N\gamma} \int_0^{N\gamma} \mathrm{FI}(\nu_t \| \pi) dt &= \mathcal{O}\left(\frac{d^{7/4} L_m}{N^{1/4}}\right) + \mathcal{O}\left(\frac{1}{N^{1/4}}\right) + \mathcal{O}\left(\frac{1}{N^{3/4}}\right) + \mathcal{O}\left(\frac{\log N}{N}\right) \\
&= \mathcal{O}\left(\frac{d^{7/4} L_m}{N^{1/4}}\right)
\end{aligned}
\tag{100}
$$

By the convexity of the Fisher information and Jensen's inequality, it follows that

$$
\mathrm{FI}(\bar{\nu}_{N\gamma} \| \pi) = \mathcal{O}\left(\frac{d^{7/4} L_m}{N^{1/4}}\right).
$$

Therefore, to obtain $\mathrm{FI}(\bar{\nu}_{N\gamma} \| \pi) \leq \varepsilon$, the sufficient number of iterations is

$$
N = \mathcal{O}\left(\frac{d^7 L_m^4}{\varepsilon^4}\right).
$$

Note that the mean per-iteration cost is $pb + (1 - p)b' = \mathcal{O}(1)$, therefore, the total number of function evaluations is $\mathcal{O}\left(\frac{d^7 L_m^4}{\varepsilon^4}\right)$ as well. In the proof of Theorem 1, we verify that $N \geq \mathcal{O}\left(\frac{d^7 L_m^4}{\varepsilon^4}\right)$ satisfies the conditions

$$
p \in (0, 1] \quad \text{and} \quad \gamma \leq \frac{1}{L_m \sqrt{85 \phi(\mu)}}.
\tag{101}
$$

Therefore, all required conditions on $N$ hold by the iteration complexity, completing the proof. $\square$

## B.7. Proof of Corollary 2

**Corollary 2** *Let $\pi \propto \ell(\boldsymbol{y}|\boldsymbol{x})p(\boldsymbol{x})$ be the posterior with the likelihood $\ell(\boldsymbol{y}|\boldsymbol{x}) \propto \exp(-f(\boldsymbol{x}))$ and the prior $p(\boldsymbol{x}) \propto \exp(-h(\boldsymbol{x}))$. Suppose the likelihood potential $f$ satisfies Assumptions 1, 2 with Lipschitz constants $L_{f_1}$, $L_{f_2}$, respectively, the target posterior $\pi$ satisfies Assumption 3 with constant $C_{\mathrm{PI}} > 0$, the prior potential $h$ satisfies Assumption 2 with Lipschitz constant $L_h$ and Assumption 4, and the SGM satisfies Assumption 5 with decreasing error $\varepsilon_{\sigma_k} = \mathcal{O}(k^{-1/2})$ for $\sigma_k > 0$, and $k \geq 1$. Define $L_m := \max\{L_{f_2} + L_h, L_{f_1}\}$. Let $(\nu_t)_{t\geq 0}$ denote the law of interpolation generated by (14) with annealing and noise schedules defined in (13). For any step size $\gamma \in \left(0, \frac{1}{L_m\sqrt{85\phi(\mu)}}\right]$, where $\phi(\mu) := 1 + \frac{4(1-p)d}{p\mu^2 b'}$, and for any $N \geq 1$, we have*

$$\|\bar{\nu}_{N\gamma} - \pi\|_{\mathrm{TV}}^2 \leq 16L_m\sqrt{\frac{C_0 C_{\mathrm{PI}} d\phi(\mu)}{N}} + \frac{34d(d+2)C_{\mathrm{PI}}L_{f_1}^2}{b} + \frac{17}{2}\mu^2 C_{\mathrm{PI}}L_{f_2}^2(d+3)^3 + \bar{\sigma}^2 + \bar{\varepsilon}_\sigma^2 + \bar{\alpha}^2,$$

*where*

$$\bar{\sigma}^2 := \frac{102C_{\mathrm{PI}}C_1^2}{N}\sum_{k=0}^{N-1}\sigma_k^2, \quad \bar{\varepsilon}_\sigma^2 := \frac{102C_{\mathrm{PI}}}{N}\sum_{k=0}^{N-1}\varepsilon_{\sigma_k}^2, \quad \text{and} \quad \bar{\alpha}^2 := \frac{102C_{\mathrm{PI}}C_2^2}{N}\sum_{k=0}^{N-1}\frac{(\alpha_k-1)^2}{\sigma_k^2},$$

*and $C_0, C_1, C_2$ are positive constants. Furthermore, let*

$$\gamma = \frac{1}{L_m N^{3/4}d^{7/4}}, \quad p = \frac{L_m}{N^{1/4}d^{1/4}} \quad b = \left\lceil\frac{1}{p}\right\rceil, \quad \text{and} \quad \mu^2 = \frac{1}{L_m N^{1/4}d^{5/4}},$$

*and let the initial parameters satisfy $\sigma_0 \geq \sigma_{min}$, $\varepsilon_{\sigma_0} > 0$, $\alpha_0 \geq 1$, and $\sigma_{min} > 0$ is the minimum noise level. Then, to achieve $\|\bar{\nu}_{N\gamma} - \pi\|_{\mathrm{TV}}^2 \leq \varepsilon$ for the time average law $\bar{\nu}_{N\gamma} := \frac{1}{N\gamma}\int_0^{N\gamma}\nu_t\,dt$, $\mathcal{O}\left(\frac{d^7 L_m^4 C_{\mathrm{PI}}^4}{\varepsilon^4}\right)$ iterations with $\mathcal{O}(1)$ function evaluations per iteration is sufficient.*

*Proof.* Recall that from Theorem 3 proof, we have inequality (93) as

$$\frac{1}{N\gamma}\int_0^{N\gamma}\mathrm{FI}(\nu_t\|\pi)dt \leq \frac{C_0}{N\gamma} + \frac{17d(d+2)L_{f_1}^2}{2b} + \frac{17}{8}\mu^2 L_{f_2}^2(d+3)^3 + \bar{\sigma}^2 + \bar{\varepsilon}_\sigma^2 + \bar{\alpha}^2 + 32\gamma L_m^2 d\phi(\mu),$$

where $C_0 = 2\mathrm{KL}(\nu_0\|\pi) + \frac{17\gamma e_0^2}{2p}$. By the convexity of the Fisher information, we have

$$\begin{aligned}
\mathrm{FI}(\bar{\nu}_{N\gamma}\|\pi) &\leq \frac{1}{N\gamma}\int_0^{N\gamma}\mathrm{FI}(\nu_t\|\pi)dt \\
&\leq \frac{C_0}{N\gamma} + \frac{17d(d+2)L_{f_1}^2}{2b} + \frac{17}{8}\mu^2 L_{f_2}^2(d+3)^3 + \bar{\sigma}^2 + \bar{\varepsilon}_\sigma^2 + \bar{\alpha}^2 + 32\gamma L_m^2 d\phi(\mu)
\end{aligned}$$

where $\bar{\nu}_{N\gamma} := \frac{1}{N\gamma}\int_0^{N\gamma}\nu_t dt$. By Assumption 3, we invoke Lemma 4 and get

$$\begin{aligned}
\|\bar{\nu}_{N\gamma} - \pi\|_{\mathrm{TV}}^2 &\leq 4C_{\mathrm{PI}}\mathrm{FI}(\bar{\nu}_{N\gamma}\|\pi) \\
&\leq \frac{4C_0 C_{\mathrm{PI}}}{N\gamma} + \frac{34d(d+2)C_{\mathrm{PI}}L_{f_1}^2}{b} + \frac{17}{2}\mu^2 C_{\mathrm{PI}}L_{f_2}^2(d+3)^3 + 4C_{\mathrm{PI}}(\bar{\sigma}^2 + \bar{\varepsilon}_\sigma^2 + \bar{\alpha}^2) \\
&\quad + 128\gamma C_{\mathrm{PI}}L_m^2 d\phi(\mu).
\end{aligned} \tag{102}$$

Let

$$\gamma = \frac{1}{L_m N^{3/4}d^{7/4}}, \quad p = \frac{L_m}{N^{1/4}d^{1/4}} \quad b = \left\lceil\frac{1}{p}\right\rceil, \quad \text{and} \quad \mu^2 = \frac{1}{L_m N^{1/4}d^{5/4}}.$$

Substituting these choices into (102) and following the derivation steps from (94) to (100) in Theorem 3, we obtain

$$\|\bar{\nu}_{N\gamma} - \pi\|_{\mathrm{TV}}^2 = \mathcal{O}\left(\frac{C_{\mathrm{PI}}d^{7/4}L_m}{N^{1/4}} + \frac{C_{\mathrm{PI}}\log N}{N}\right) \tag{103}$$

provided that the annealing and noise schedules are defined as (13) and the SGM estimation error obeys $\varepsilon_{\sigma_k} := \mathcal{O}(k^{-1/2})$ and for all $k \geq 1$ with initial value $\varepsilon_{\sigma_0} > 0$. Thereofer, to achieve $\|\bar{\nu}_{N\gamma} - \pi\|_{\mathrm{TV}}^2 \leq \varepsilon$, the sufficient number of iterations is

$$N = \mathcal{O}\left( \frac{C_{\mathrm{PI}}^4 d^7 L_m^4}{\varepsilon^4} \right).$$

Furthermore, the per-iteration computation cost is $pb + (1-p)b' = \mathcal{O}(1)$; therefore, the number of total function evaluations is $\mathcal{O}\left( \frac{C_{\mathrm{PI}}^4 d^7 L_m^4}{\varepsilon^4} \right)$ as well. $\qquad\square$

### B.8. Proof of Theorem 4

**Theorem 4** *Let $\pi \propto \ell(\boldsymbol{y}|\boldsymbol{x})p(\boldsymbol{x})$ be the posterior with the likelihood $\ell(\boldsymbol{y}|\boldsymbol{x}) \propto \exp(-f(\boldsymbol{x}))$ and the prior $p(\boldsymbol{x}) \propto \exp(-h(\boldsymbol{x}))$. Suppose the likelihood potential $f$ satisfies Assumptions 1, 2 with Lipschitz constants $L_{f_1}$, $L_{f_2}$, respectively, the prior potential $h$ satisfies Assumption 2 with Lipschitz constant $L_h$ and Assumption 4, and the SGM satisfies Assumption 5 with decreasing error $\varepsilon_{\sigma_k} = \mathcal{O}(k^{-1/2})$ for $\sigma_k > 0$, and $k \geq 1$. Define $L_m := \max\{L_{f_2} + L_h, L_{f_1}\}$. Let $(\nu_t)_{t \geq 0}$ denote the law of interpolation generated by (14) with annealing and noise schedules defined in (13). Define the time-varying parameters as follows*

$$\gamma_k = \frac{C_\gamma}{k^{3/2}}, \quad p_k = \frac{1}{2k^{1/2}}, \quad b_k = \left\lceil \frac{1}{p_k} \right\rceil, \quad \text{and} \quad \mu_k = \frac{C_\mu}{k^{1/8}},$$

*where $L_m := \max\{L_{f_2} + L_h, L_{f_1}\}$, and $C_\gamma, C_\mu$ are positive constants. Let the initial parameters satisfy $\sigma_0 \geq \sigma_{min}$, $\varepsilon_{\sigma_0} > 0$, $\alpha_0 \geq 1$, and $\sigma_{min} > 0$ is the minimum noise level. Then, the time-averaged law $\bar{\nu}_{\tau_n} := \frac{1}{\tau_n} \int_0^{\tau_n} \nu_t \, dt$, where $\tau_n := \sum_{k=1}^n \gamma_k$, converges weakly to $\pi$.*

*Proof.* Given time-varying parameters $\gamma_k, b_k, p_k, \mu_k$ at iteration $k$, define the cumulative time $\tau_n$ and averaged law $\bar{\nu}_{\tau_n}$ at iteration $n$ as

$$\tau_n := \sum_{k=1}^n \gamma_k, \qquad \bar{\nu}_{\tau_n} := \frac{1}{\tau_n} \int_0^{\tau_n} \nu_t \, dt,$$

where $\nu_t$ denotes the law of the process $\boldsymbol{x}_t$ under the following continuous-time interpolation

$$\boldsymbol{x}_t := \boldsymbol{x}_{\tau_{n-1}} - (t - \tau_{n-1})\left(\boldsymbol{g}_n - \alpha_n \mathcal{S}_\theta(\boldsymbol{x}_{\tau_n}, \sigma_n)\right) + \sqrt{2}\left(\boldsymbol{B}_t - \boldsymbol{B}_{\tau_{n-1}}\right), \qquad t \in [\tau_{n-1}, \tau_n]. \tag{104}$$

$g_k$ is defined as follows

$$\boldsymbol{g}_n := \begin{cases} \dfrac{1}{b_n} \sum_{i=1}^{b_n} \widetilde{\nabla} f_{\mu_n}(\boldsymbol{x}_{\tau_n}, \boldsymbol{u}_n^i) & \text{w.p. } p_n, \\[4mm] \boldsymbol{g}_{n-1} + \dfrac{1}{b'} \sum_{i=1}^{b'} \left( \widetilde{\nabla} f_{\mu_n}(\boldsymbol{x}_{\tau_n}, \boldsymbol{u}_n^i) - \widetilde{\nabla} f_{\mu_n}(\boldsymbol{x}_{\tau_{n-1}}, \boldsymbol{u}_n^i) \right) & \text{w.p. } 1 - p_n, \end{cases} \tag{105}$$

for all $n \geq 1$. At initialization, we choose $\gamma_0, b_0, \mu_0 > 0$ and $p_0 \in (0, 1]$, and define $\boldsymbol{g}_0$

$$\boldsymbol{g}_0 := \frac{1}{b_0} \sum_{i=1}^{b_0} \widetilde{\nabla} f_{\mu_0}(\boldsymbol{x}_0, \boldsymbol{u}_0^i), \quad \boldsymbol{u}_0^i \sim \mathcal{N}(0, I).$$

We establish weak convergence by adapting the proof of Theorem 3. To this end, we first verify that the step sizes $(\gamma_k)_{k \geq 1}$ satisfy the step-size condition of Theorem 3, namely,

$$\gamma_k \in \left( 0, \frac{1}{L_m \sqrt{85 \phi_k(\mu_k)}} \right], \quad \text{where} \quad \phi_k(\mu_k) = 1 + \frac{4(1 - p_k)d}{p_k \mu_k^2 b'} \tag{106}$$

for $k \geq 1$. Using the definitions of $\mu_k$ and $p_k$, we have

$$
\begin{aligned}
\phi_k(\mu_k) &= 1 + \frac{4(1 - p_k)d}{p_k \mu_k^2 b'} \\
&\leq 1 + \frac{4d}{p_k \mu_k^2 b'} \\
&\leq 1 + \frac{8dk^{3/4}}{b' C_\mu^2} \\
&\leq \frac{16dk^{3/4}}{b' C_\mu^2},
\end{aligned}
\tag{107}
$$

where the last inequality is satisfied by selecting a constant $C_\mu$ such that

$$
\sqrt{\frac{8d}{b'}} \leq C_\mu.
$$

Then, we have

$$
\frac{1}{L_m \sqrt{85 \phi_k(\mu_k)}} \geq \frac{1}{L_m \sqrt{\frac{1360 d k^{3/4}}{b' C_\mu^2}}} = \frac{C_\mu}{L_m k^{3/8}} \sqrt{\frac{b'}{1360 d}}.
$$

Choosing

$$
C_\gamma = \frac{C_\mu}{L_m} \sqrt{\frac{b'}{1360 d}},
$$

we have

$$
\gamma_k = \frac{C_\gamma}{k^{3/2}} \leq \frac{C_\mu}{L_m k^{3/8}} \sqrt{\frac{b'}{1360 d}} \leq \frac{1}{L_m \sqrt{85 \phi_k(\mu_k)}},
$$

which implies (106) holds for $k \geq 1$. Furthermore, we note that $p_k \in (0, 1]$ holds for all $k \geq 1$ by definition. Therefore, we can follow the same steps up to (92) in Theorem 3 since the same arguments remain valid for $t \in [\tau_{n-1}, \tau_n]$ with time-varying parameters. We obtain

$$
\begin{aligned}
\mathrm{KL}(\nu_{\tau_n} \| \pi) - \mathrm{KL}(\nu_{\tau_{n-1}} \| \pi) \leq {}& -\frac{1}{2} \int_{\tau_{n-1}}^{\tau_n} \mathrm{FI}(\nu_t \| \pi) dt + \frac{17 \gamma_n d(d+2) L_{f_1}^2}{4 b_n} + \frac{17}{16} \gamma_n \mu_n^2 L_{f_2}^2 (d+3)^3 \\
& - \frac{4 \gamma_n}{p_n} \left( 1 + \frac{85 \gamma_n^2 L_m^2 \phi_n(\mu_n)}{16} \right) \left( e_n^2 - e_{n-1}^2 \right) \\
& + \frac{51 \gamma_n}{4} (C_1^2 \sigma_n^2 + \varepsilon_{\sigma_n}^2 + (\alpha_n - 1)^2 C_2^2 \sigma_n^{-2}) + 16 \gamma_n^2 d L_m^2 \phi_n(\mu_n)
\end{aligned}
\tag{108}
$$

for $t \in [\tau_{n-1}, \tau_n]$. Plugging the definitions of $\gamma_k$, $b_k$, $p_k$, and $\mu_k$ given in Theorem 4 into (108), we get

$$
\begin{aligned}
\mathrm{KL}(\nu_{\tau_n} \| \pi) - \mathrm{KL}(\nu_{\tau_{n-1}} \| \pi) \leq {}& -\frac{1}{2} \int_{\tau_{n-1}}^{\tau_n} \mathrm{FI}(\nu_t \| \pi) dt + \frac{A_1}{n^2} + \frac{A_2}{n^{7/4}} + \frac{A_3}{n^{9/4}} - c_n \left( e_n^2 - e_{n-1}^2 \right) \\
& + \frac{51 \gamma_n}{4} (C_1^2 \sigma_n^2 + \varepsilon_{\sigma_n}^2 + (\alpha_n - 1)^2 \sigma_n^{-2} C_2^2),
\end{aligned}
\tag{109}
$$

where

$$
A_1 := \frac{17 d(d+2) L_{f_1}^2 C_\gamma}{8}, \quad A_2 := \frac{17 L_{f_2}^2 (d+3)^3 C_\gamma C_\mu^2}{16}, \quad A_3 := \frac{256 d^2 L_m^2 C_\gamma^2}{b' C_\mu^2},
$$

and

$$
c_n := \frac{\gamma_n}{p_n} \left( 4 + \frac{85 \gamma_n^2 L_m^2 \phi_n(\mu_n)}{16} \right),
$$

for $n \geq 1$. We use (107) to obtain $A_3$. Iterating the bound in (109), we obtain

$$
\mathrm{KL}(\nu_{\tau_n}\|\pi) \leq \mathrm{KL}(\nu_0\|\pi) - \frac{1}{2}\int_0^{\tau_n} \mathrm{FI}(\nu_t\|\pi)dt + A_1S_1 + A_2S_2 + A_3S_3 - \sum_{k=1}^n c_k\left(e_k^2 - e_{k-1}^2\right)
$$

$$
+ \frac{51}{4}\sum_{k=1}^n \gamma_k(\sigma_k^2 C^2 + \varepsilon_{\sigma_k}^2 + (\alpha_k - 1)^2\sigma_k^{-2}C^2), \tag{110}
$$

where we have

$$
\sum_{k=1}^n k^{-2} \leq \underbrace{\sum_{k=1}^\infty k^{-2}}_{:=S_1} < \infty, \quad \sum_{k=1}^n k^{-7/4} \leq \underbrace{\sum_{k=1}^\infty k^{-7/4}}_{:=S_2} < \infty.
$$

$$
\sum_{k=1}^n k^{-9/4} \leq \underbrace{\sum_{k=1}^\infty k^{-9/4}}_{:=S_3} < \infty.
$$

Thus, $A_1S_1$, $A_2S_2$, and $A_3S_3$ are uniformly bounded constants and are independent of $n$. Furthermore, if we assume that $c_n$ is nonnegative and decreasing sequence (i.e. $0 \leq c_{n+1} < c_n$), we can bound the summation in (110) as follows

$$
-\sum_{k=1}^n c_k\left(e_k^2 - e_{k-1}^2\right) = c_1 e_0^2 + \sum_{k=1}^{n-1}(c_{k+1} - c_k)e_k^2 - c_n e_n^2 \leq c_1 e_0. \tag{111}
$$

Thus, it remains to show that $(c_n)_{n\geq 1}$ is a nonnegative and decreasing sequence. Substituting the parameters $\gamma_n$, $b_n$, $p_n$, and $\mu_n$ into the definition of $c_n$, we obtain

$$
c_n = \frac{8C_\gamma}{n} + \frac{85L_m^2C_\gamma^3}{8}\frac{1}{n^4} + \frac{85dL_m^2C_\gamma^3}{b'C_\mu^2}\frac{1}{n^{13/4}}\left(1 - \frac{1}{2n^{1/2}}\right)
$$

for $n \geq 1$. Let $F := \frac{85L_m^2C_\gamma^3}{8}$, which is a numerical constant. Then, we have

$$
c_n = \frac{8C_\gamma}{n} + \frac{F}{n^4} + \frac{8Fd}{b'C_\mu^2 n^{13/4}}\left(1 - \frac{1}{2n^{1/2}}\right).
$$

Nonnegativity is immediate since $1 - \frac{1}{2n^{1/2}} > 0$ for $n \geq 1$. Therefore, all the terms in $c_n$ are nonnegative for $n \geq 1$. To show $(c_n)_{n\geq 1}$ is decreasing, define the continuous extension $c(x)$, for $x \geq 1$, as

$$
c(x) = \frac{8C_\gamma}{x} + \frac{F}{x^4} + \frac{8Fd}{b'C_\mu^2 x^{13/4}}\left(1 - \frac{1}{2x^{1/2}}\right).
$$

Taking the derivative yields

$$
c'(x) = -\frac{8C_\gamma}{x^2} - \frac{4F}{x^5} - \frac{26Fd}{b'C_\mu^2 x^{17/4}}\left(1 - \frac{15}{26x^{1/2}}\right).
$$

For $x \geq 1$, we have

$$
1 - \frac{15}{26x^{1/2}} > 0,
$$

and hence $c'(x) < 0$. Therefore, $c(x)$ is decreasing on $[1, \infty)$, which implies that $(c_n)_{n\geq 1}$ is decreasing. Hence, using the upper bound (111) in (110), we get

$$
\mathrm{KL}(\nu_{\tau_n}\|\pi) \leq \mathrm{KL}(\nu_0\|\pi) - \frac{1}{2}\int_0^{\tau_n}\mathrm{FI}(\nu_t\|\pi)dt + A_1S_1 + A_2S_2 + A_3S_3 + c_1e_0^2
$$

$$
+ \frac{51}{4}\sum_{k=1}^n \gamma_k(C_1^2\sigma_k^2 + \varepsilon_{\sigma_k}^2 + (\alpha_k - 1)^2C_2^2\sigma_k^{-2}). \tag{112}
$$

We next show that the last term in (112), capturing the effects of the noise and annealing schedules and the SGM error, is uniformly bounded.

For the noise schedule $(\sigma_k)_{k=1}^n$, using monotonicity, we obtain

$$
\begin{aligned}
\frac{51C_1^2}{4} \sum_{k=1}^n \gamma_k \sigma_k^2 &= \frac{51C_1^2\sigma_0^2}{4} \sum_{k=1}^n \gamma_k \\
&= \frac{51C_1^2\sigma_0^2 C_\gamma}{4} \sum_{k=1}^n \frac{1}{k^{3/2}} \\
&\leq \frac{51C_1^2\sigma_0^2 C_\gamma}{4} \underbrace{\sum_{k=1}^\infty \frac{1}{k^{3/2}}}_{:=C_\sigma} \tag{113} \\
&< \infty. \tag{114}
\end{aligned}
$$

For the annealing and noise schedules $(\alpha_k)_{k=1}^n$ and $(\sigma_k)_{k=1}^n$, respectively, using monotonicity together with the bounds $\alpha_k \leq \alpha_0$ and $\sigma_k \geq \sigma_{\min}$, we obtain

$$
\begin{aligned}
\frac{51C_2^2}{4} \sum_{k=1}^n \frac{\gamma_k(\alpha_k-1)^2}{\sigma_k^2} &\leq \frac{51C_2^2(\alpha_0-1)^2}{4\sigma_{\min}^2} \sum_{k=1}^n \gamma_k \\
&\leq \frac{51C_2^2(\alpha_0-1)^2}{4\sigma_{\min}^2} \underbrace{\sum_{k=1}^\infty \frac{1}{k^{3/2}}}_{:=C_\alpha} \tag{115} \\
&< \infty. \tag{116}
\end{aligned}
$$

Lastly, for the SGM error decay $(\varepsilon_{\sigma_k})_{k=1}^n$, we have

$$
\frac{51}{4} \sum_{k=1}^n \gamma_k \varepsilon_{\sigma_k}^2 \leq \frac{51}{4} \underbrace{\sum_{k=1}^\infty \mathcal{O}\left(\frac{1}{k^{5/2}}\right)}_{:=C_{\varepsilon_\sigma}} < \infty. \tag{117}
$$

Using the upper bounds (113), (115), and (117) to upper bound the last term in (112), we obtain

$$
\mathrm{KL}(\nu_{\tau_n}\|\pi) \leq \mathrm{KL}(\nu_0\|\pi) - \frac{1}{2}\int_0^{\tau_n} \mathrm{FI}(\nu_t\|\pi)dt + A_1S_1 + A_2S_2 + A_3S_3 + c_1e_0^2 + C_\alpha + C_\sigma + C_{\varepsilon_\sigma}. \tag{118}
$$

Rearranging the terms, using the convexity of Fisher information and multiplying both sides by $2/\tau_n$, we get

$$
\begin{aligned}
\mathrm{FI}(\bar{\nu}_{\tau_n}\|\pi) &\leq \frac{1}{\tau_n}\int_0^{\tau_n} \mathrm{FI}(\nu_t\|\pi)\,dt \\
&\leq \frac{2\,\mathrm{KL}(\nu_0\|\pi)}{\tau_n} + \frac{2}{\tau_n}\Big(A_1S_1 + A_2S_2 + A_3S_3 + c_1e_0^2 + C_\alpha + C_\sigma + C_{\varepsilon_\sigma}\Big), \tag{119}
\end{aligned}
$$

where $A_1S_1 + A_2S_2 + A_3S_3 + C_\alpha + C_{\varepsilon_\sigma} + C_\sigma < \infty$ and does not depend on $n$. On the other hand, if $t \in [\tau_n, \tau_{n+1}]$, integrating (91) between $\tau_n$ and $t$ and dropping the negative integral over the Fisher information give us

$$
\begin{aligned}
\mathrm{KL}(\nu_t\|\pi) &\leq \mathrm{KL}(\tau_n\|\pi) + \frac{17(t-\tau_n)d(d+2)L_{f_1}^2}{4b} + \frac{17}{16}(t-\tau_n)\mu^2 L_{f_2}^2(d+3)^3 \\
&\quad + 4(t-\tau_n)\gamma dL_m^2\phi(\mu) - \frac{4(t-\tau_n)}{p}\left(1 + \frac{85\gamma^2 L_m^2\phi(\mu)}{16}\right)(e_{n+1}^2 - e_n^2) \\
&\quad + \frac{51(t-\tau_n)}{4}(C_1^2\sigma_k^2 + \varepsilon_{\sigma_k}^2 + (\alpha_k-1)^2 C_2^2\sigma_k^{-2}) \\
&\leq \mathrm{KL}(\nu_0\|\pi) + 2A_1S_1 + 2A_2S_2 + 2A_3S_3 + 2c_1e_0^2 + c_{n+1}e_n^2 + 2C_\alpha + 2C_{\varepsilon_\sigma} + 2C_\sigma. \tag{120}
\end{aligned}
$$

We obtain (120) by bounding all terms except the KL divergence, $c_1 e_0^2$, and $c_n e_n^2$ by their finite limits as $n \to \infty$. We then apply the bound on $\mathrm{KL}(\nu_{\tau_n} \| \pi)$ derived from (118). By Assumption 1 and Lemma 1, the initial error satisfies $e_0 < \infty$, and hence $c_1 e_0^2 < \infty$. In addition, following the steps from (78) to (79) in the proof of Theorem 2, we have $e_n^2 < \infty$. Combining these with (120) implies that $\{\mathrm{KL}(\nu_t \| \pi) \mid t \geq 0\}$ is uniformly bounded. By the convexity of the KL divergence, this implies that the sequence $\{\mathrm{KL}(\bar{\nu}_{\tau_n} \| \pi)\}_{n \in \mathbb{N}}$ is uniformly bounded as well. Since the sublevel sets of $\mathrm{KL}(\cdot \| \pi)$ are weakly compact, $(\bar{\nu}_{\tau_n})_{n \in \mathbb{N}}$ is tight. To establish that $\bar{\nu}_{\tau_n} \rightharpoonup \pi$ weakly, it suffices to verify that every cluster point of $(\bar{\nu}_{\tau_n})_{n \in \mathbb{N}}$ equal to $\pi$.

Consider a subsequence $(\bar{\nu}_{\tau_n})_{n \in \mathbb{N}}$ converging to some limit $\bar{\nu}$. Taking $n \to \infty$ in (119) and noting that $\tau_n \to \infty$, we obtain $\mathrm{FI}(\bar{\nu}_{\tau_n} \| \pi) \to 0$. Therefore, the same holds along the subsequence. By the weak lower semicontinuity of the Fisher information along the subsequence, we have $\mathrm{FI}(\bar{\nu} \| \pi) = 0$. Writing $\psi := \frac{d\bar{\nu}}{d\pi}$, this means $\sqrt{\psi} \in \mathrm{dom}\, \mathcal{E}$ and $\mathcal{E}(\sqrt{\psi}) = 0$, where $\mathcal{E}$ denotes the Dirichlet energy (i.e., the squared $L^2(\pi)$-norm of the gradient; see Section 3 in Balasubramanian et al. (2022)). Since $\nabla \log \pi$ is Lipschitz by Assumption 2, $\pi$ has a continuous and strictly positive density on $\mathbb{R}^d$, so $\mathcal{E}(\sqrt{\psi}) = 0$, which implies that $\psi$ must be a constant $\pi$-a.e., hence $\bar{\nu} = \pi$. $\qquad\square$

# C. Extended Experimental Results

In this section, we provide additional details on our experiments. The code is publicly available at `https://github.com/mberk-sahin/zo-posterior-sampling`, where additional implementation details can be found.

### C.1. Numerical Validation

Similar to the related works (Sun et al., 2024; Song & Ermon, 2020), we run ZO-APMC with an exponential annealing schedule:

$$\sigma_k := \max\{\sigma_0 \rho_2^k, \sigma_{\min}\}, \quad \alpha_k = \max\{\alpha_0 \sigma_k^2, 1\}, \tag{121}$$

where $\rho_2$ is the decay rate and $k$ is the step index. We always choose $\alpha_0 \leq 1/\sigma_{\min}^2$ so that $\alpha_k$ converges to 1. We note that our definition of annealing and noise schedules (13) with two different independent decay rates, which are $\rho_1$ and $\rho_2$, for each schedule is more general than this definition since one could easily recover (121) by setting $\rho_2$, $\alpha_0$, $\sigma_0$, and $\sigma_{\min}$ properly. For our numerical validation results, we set $\sigma_0 = 10$, $\alpha_0 = 10$, $\rho_2 = 0.975$, $\sigma_{\min} = 0$ and $\gamma = 0.1$. For ZO estimator, we choose the smoothing parameter as $\mu = 10^{-4}$. We run ZO-APMC with 1000 sample points initialized with uniform distribution $\mathrm{U}[-50, 50]^2$ on $[-50, 50]^2$ grid for $N = 2000$ iterations. At each step, we use a Gaussian mixture model (GMM) to fit a distribution to the samples at intermediate steps, which allows us to compute the probability of an arbitrary value on $[-50, 50]^2$ grid. Then, for each intermediate GMM distribution, we calculate the empirical Fisher information relative to target posterior whose analytical posterior can be calculated. We discretize the grid to $1000 \times 1000$ unit areas in $[-50, 50]^2$ and calculate the Fisher information for each unit area. The total sum over the grid gives us the approximate relative Fisher information. As stated in Section 4.1, all results in the numerical validation experiments are obtained by repeating the same experimental procedure across 20 random seeds, each corresponding to a different randomly generated forward operator. Reported Fisher information and KL divergence values are averaged over these runs.

**Convergence in Fisher Information.** We provide additional experiments illustrating the effect of $p$ on FI convergence in Fig. 6. At each iteration, ZO-APMC performs a zero-order estimate with a large batch size $b = 10$ with probability $p$, while with probability $1 - p$ it uses a smaller batch size $b' = 1$, whose gradient estimate is aggregated with the previous step's update.

To further demonstrate the benefit of the variance reduction mechanism, we compare naive ZO-APMC approach ($p = 1$) discussed in after Proposition 1 with ZO-APMC with $p < 1$. We use the same inverse problem setting and analytical prior explained in Section 4.1. We note that we approximate FI accurately because our generated samples live in $\mathbb{R}^2$. In higher dimensions, this approximation quickly becomes unreliable and expensive. Importantly, in $\mathbb{R}^2$, ZO estimators do not suffer from the curse of dimensionality, which is a primary motivation for our variance-reduction mechanism. To better mimic high-dimensional behavior and induce higher variance in ZO estimates, we inject synthetic noise into the likelihood score estimates with standard deviation $\sigma_{\mathrm{noise}} = 10$. Given a fixed budget of 6 score likelihood approximations on average, which corresponds to 12 forward model evaluations, we run ZO-APMC with different pairs of $(p, b) \in \{(1, 6), (0.4, 9), (0.2, 14), (0.1, 24)\}$. For each pair, we choose $b'$ such that $(1 - p)b' + pb \leq 6$ and the algorithm remains within the per-iteration budget. This ensures a fair comparison between ZO-APMC with $p < 1$ and the naive ZO-APMC ($p = 1$) approach. In addition to the Gaussian noise model, we also consider a Laplace noise model, which is commonly

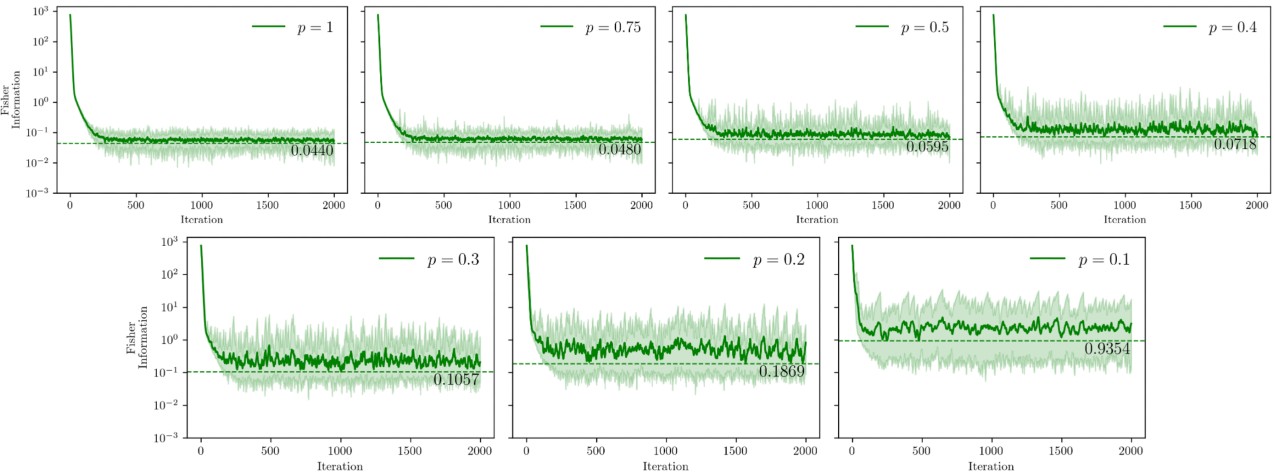

*Figure 6.* Effect of $p$ on the convergence of ZO-APMC to the true posterior distribution in terms of relative Fisher information. The solid lines show the mean values and shaded areas show the minimum and maximum ranges.

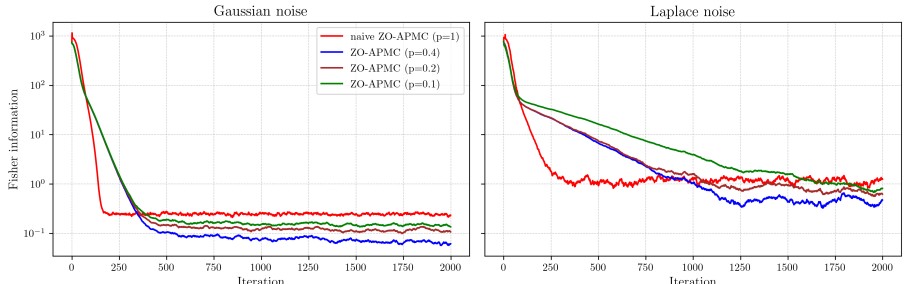

*Figure 7.* Comparison of naive ZO-APMC ($p = 1$) and ZO-APMC with $p < 1$ for Fisher information convergence under Gaussian (left) and Laplace (right) measurement noise types in an inverse problem. Each plot performs the same number of forward model evaluations per iteration on average.

used in practice. Accordingly, we conduct experiments under both noise modeling assumptions. We present the results in Fig. 7. They verify our theoretical results in Theorem 3. When $p = 1$, ZO-APMC converges to the largest suboptimal point. This suggests that when $b$ is not large enough, the upper bound in Theorem 3 becomes suboptimal and it can be made optimal only by increasing $b$. In contrast, when $p < 1$, we can increase $b$ without changing the per-iteration budget. Doing so improves the convergence of the FI because ZO-APMC with different values of $p < 1$ converge to smaller values. We further note that the convergence for $p = 0.1$ is worse than $p = 0.4$. This is consistent with Theorem 3 because as $p$ reduces, it increases the discretization error (second term) at the upper bound. We present a similar comparison for statistical validation in the subsequent section after weak convergence results.

**Weak Convergence.** To verify the weak convergence of ZO-APMC to the target posterior, we consider the same synthetic inverse problem setting as in the previous experiments, but use time-varying schedules for the step size $\gamma_k$, refresh probability $p_k$, batch size $b_k$, and ZO smoothing parameter $\mu_k$. Following Theorem 4, we employ decreasing schedules for $\gamma_k$, $p_k$, and $\mu_k$, together with an increasing schedule for $b_k$. Specifically, letting $t = k/N \in [0, 1]$ denote the normalized iteration index, where $N$ is the total number of iterations, we linearly decay the step size from $\gamma_0 = 0.1$ to $\gamma_{\min} = 0.001$, the refresh probability from its initial value $p_0$ (reported in Fig. 8) to $p_{\min} = 0.01$, and the smoothing parameter from $\mu_0 = 10^{-4}$ to $\mu_{\min} = 10^{-5}$. Consistent with the theoretical relation $b_k = \lceil 1/p_k \rceil$, the batch size is increased linearly from $b_0 = 10$ to $b_{\max} = 100$. Unlike the asymptotic schedules in Theorem 4, where $\gamma_k, p_k, \mu_k \to 0$ and $b_k \to \infty$, we use bounded practical schedules for numerical stability and finite computational cost. Moreover, consistent with Theorem 4, the SGM estimation error is linearly decreased from $\varepsilon_{\sigma_0} = 2.5$ to 0 throughout sampling. Fig. 8 shows that, across different initial values of $p_0$, ZO-APMC achieves near-zero KL divergence, providing empirical support for the weak convergence of the generated sample distribution toward the target posterior predicted by Theorem 4. Despite using only ZO likelihood score estimates (8), ZO-APMC closely matches the convergence behavior of APMC using exact likelihood scores.

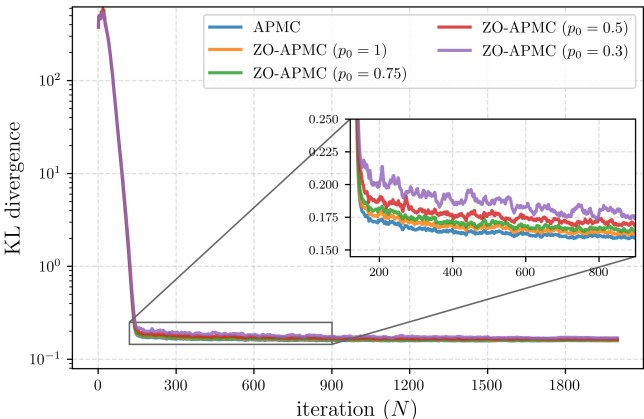

*Figure 8.* Convergence of APMC and ZO-APMC to the target posterior in KL divergence for varying initial values $p_0$. The probability parameter $p$ is decreased throughout sampling, and the reported values denote its initialization. ZO-APMC converges for all initial values of $p_0$.

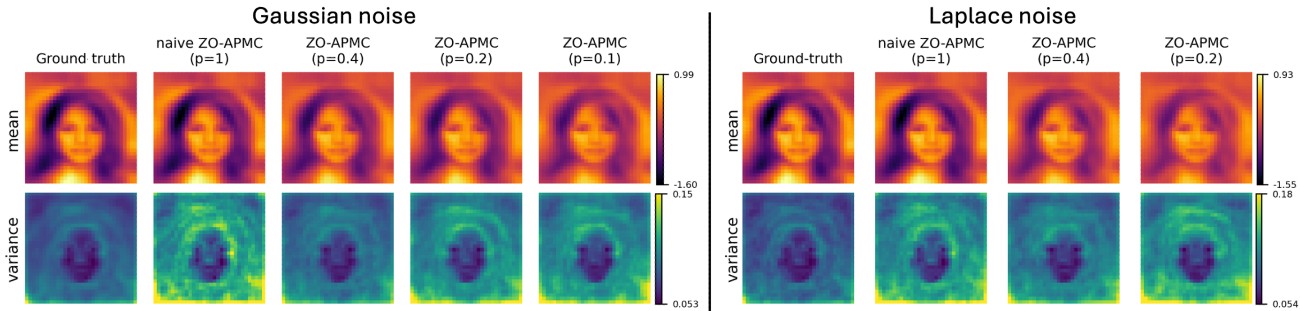

*Figure 9.* Comparison of naive ZO-APMC ($p = 1$) and ZO-APMC with $p < 1$ for estimating the ground truth posterior mean and variance statistics under Gaussian (left) and Laplace (right) measurement noise types. First row shows the posterior means and the second row shows the posterior variance. The colorbar for each statistic is shown at the end of its corresponding row. Each column shows the statistics estimated by ZO-APMC with specified probability $p$.

### C.2. Statistical Validation

The SGM used in this experiment is U-Net taken from (Nichol & Dhariwal, 2021) with some of its layers removed to process the $32 \times 32$ images, which are taken from CelebA (Liu et al., 2015) dataset. Each image is normalized to $[-1, 1]$ and downscaled to $32 \times 32$ pixels for simplicity. The forward operator is generated as random Gaussian matrix and for each test image, we inject a Gaussian noise with variance 0.01 as a measurement noise. We construct a bimodal distribution by selecting male and female images from the CelebA dataset and fitting a Gaussian mixture model (GMM) to the combined data. To ensure adequate separation, the two modes are shifted by $+1$ and $-1$. The SGM prior is then trained on samples drawn from this synthetic multimodal distribution. Because the synthetic Gaussian images lack the structural richness of natural images, the score network's results on this dataset should not be taken as representative of its performance on real-world data. For comparison, we compute the target modes and posterior statistics using the statistics derived from male and female images in the CelebA dataset.

In addition to the statistical validation experiments in Section 4.1, we compare naive ZO-APMC ($p\!=\!1$) with ZO-APMC for $p\!<\!1$ under Gaussian and Laplace noise modeling, which are two widely used noise modelings. We run each ZO-APMC algorithm for $N = 5000$ iterations to generate 1000 samples with different values of $(p, b) \in \{(1, 2), (0.4, 4), (0.2, 7), (0.1, 12)\}$ and $b'$ is selected such that the number of gradient approximations per-iteration satisfies $(1 - p)b' + pb \leq 2$. Because the Laplace likelihood and Gaussian prior are not a conjugate pair, closed-form posterior statistics are not available. Therefore, when modeling Laplace noise, we estimate the ground truth posterior statistics numerically by running standard Langevin Monte Carlo and get an empirical estimate of mean and variance. In this procedure, we use the likelihood score directly and do not apply any annealing schedule to the sampling parameters. We present the results in Fig. 9. Under Gaussian measurement noise, the naive ZO-APMC has significantly higher variance than the ground truth variance, which would

require increasing the batch size and therefore the number of forward evaluations per iteration. Instead, our ZO-APMC with $p < 1$ reduces this variance by using a larger batch size $b$ together with a small $p$ as explained previously. Fig. 9 shows that choosing $p = 0.4$ reduces the sample variance significantly and it looks very similar to the ground truth variance. However, the variance reduction is not monotonic in $p$. For $p = 0.2$ and $p = 0.1$, we observe a slight increase in variance, although it remains lower than that obtained with $p = 1$. This phenomenon follows from the trade-off characterized in Proposition 1 between variance reduction and propagated estimation error. Specifically, decreasing $p$ increases the influence of accumulated errors from previous iterations, leading to larger zeroth-order estimation error and preventing convergence to the target statistics. We observe a similar trend under Laplace noise modeling. The choice $p = 0.4$ yields the most accurate posterior variance estimate, whereas decreasing $p$ further to $p = 0.2$ leads to increased variance. Lastly, across all the experiments, we keep the ZO smoothing parameter fixed $\mu = 10^{-4}$ to isolate its effect. This results in large batch sizes being used less frequently, thereby increasing the bias-induced error predicted by Proposition 1. We observe this effect at $p = 0.1$ in the estimated posterior mean under both noise models. The effect is stronger under Laplace noise modeling because the Fisher information upper bound in Theorem 3 depends on the smoothness properties of the log-likelihood. While $\|\boldsymbol{y} - \boldsymbol{A}\boldsymbol{x}\|_2^2$ is globally smooth with Lipschitz gradient constant $2\|\boldsymbol{A}^\top \boldsymbol{A}\|_{\mathrm{op}}$, $\|\boldsymbol{y} - \boldsymbol{A}\boldsymbol{x}\|_1$ is non-smooth and only differentiable almost everywhere. Consequently, the zeroth-order smoothing bias is larger under Laplace noise modeling, although this effect can be mitigated by choosing a smaller value of $\mu$.

### C.3. MRI Reconstruction

We evaluate the reconstruction quality of the samples generated by ZO-APMC and other baselines methods by using peak signal to noise (PSNR) ratio, structural similarity index measure (SSIM), normalized root mean square error (NRMSE), and mean standard deviation (SD). Given an estimate $\hat{\boldsymbol{x}} \in \mathbb{R}^d$ and the ground truth $\boldsymbol{x}_{\mathrm{GT}} \in \mathbb{R}^d$, we define the error metrics as

$$\mathrm{MSE}(\hat{\boldsymbol{x}}, \boldsymbol{x}_{\mathrm{GT}}) := \frac{1}{d}\|\hat{\boldsymbol{x}} - \boldsymbol{x}_{\mathrm{GT}}\|_2^2, \quad \mathrm{NRMSE}(\hat{\boldsymbol{x}}, \boldsymbol{x}_{\mathrm{GT}}) := \frac{\|\hat{\boldsymbol{x}} - \boldsymbol{x}_{\mathrm{GT}}\|_2}{\|\boldsymbol{x}_{\mathrm{GT}}\|_2},$$

$$\mathrm{PSNR}(\hat{\boldsymbol{x}}, \boldsymbol{x}_{\mathrm{GT}}) := 10\log_{10}\left(\frac{\max(\boldsymbol{x}_{\mathrm{GT}})^2}{\mathrm{MSE}(\hat{\boldsymbol{x}}, \boldsymbol{x}_{\mathrm{GT}})}\right).$$

where $d$ is the dimensionality of $\boldsymbol{x}$, and $\max$ denotes the maximum possible value of the signal (e.g., 1 for normalized data or 255 for 8-bit images). In addition, we compute SD as follows: for each test image, we first calculate the standard deviation across generated samples at each pixel location, and then average these pixel-wise standard deviations to obtain a mean standard deviation for the generated image. We run ZO-APMC and APMC using the annealing and noise schedules defined in (121), with step size $\gamma = 5 \times 10^{-6}$, initial noise and annealing parameters $\sigma_0 = 348$ and $\alpha_0 = 10^4$, minimum noise level $\sigma_{\min} = 0.01$, decay rate $\rho_2 = 0.99$, and ZO smoothing parameter $\mu = 10^{-4}$. Unless otherwise specified, these hyperparameters follow the implementation and settings of Sun et al. (2024). For all other baselines, we utilize the implementations and parameters provided by InverseBench (Zheng et al., 2025b). While our theoretical results do not imply gains in sampling speed, for completeness, we report runtimes of each method to generate a sample. The per-sample runtimes (seconds) are as follows: PnPDM: 68.3, DPS: 25.4, APMC: 23.77, Forward-GSG: 38.12, Central-GSG: 37.83, DPG: 60.54, and ZO-APMC: 50.53, measured on an NVIDIA H100 GPU.

**Ablation Study.** Among the inverse problems considered in this work, MRI reconstruction involves the largest image size $(256 \times 256)$, which necessitates a larger batch size in our ZO estimator to accurately compute the forward model gradient. To identify the optimal value of $p$, we subsample examples from the validation set of FastMRI (Zbontar et al., 2019) and evaluate reconstruction quality across different values, $p \in \{0.1, 0.2, 0.4, 0.5\}$, as illustrated in Fig. 10. As indicated by the orange arrow, reducing $p$ excessively while keeping $b$ fixed produces visible artifacts in the generated samples. ZO-APMC maintains reasonable reconstruction quality down to $p = 0.2$. Even when using a smaller batch size of $b' = 10^3$, an order of magnitude lower than $b$, in about half of the iterations on average, ZO-APMC maintains high reconstruction quality that is very close both visually and quantitatively to the reconstruction of APMC. As opposed to APMC, ZO-APMC achieves this without any gradient information and uses only forward model function evaluations. Because the performance gain beyond $p = 0.2$ is not significant and the gap between $p = 0.5$ and $p = 0.2$ can be further reduced by averaging multiple parallel outputs, we set $p = 0.2$ for our brain MRI inverse problem experiments.

Moreover, ZO estimators are widely recognized in the literature for exhibiting high variance in high-dimensional settings, as they rely on first-order approximations of the function along random directions. To evaluate our proposed variance-reduction mechanism, we compare the reconstructions of our method with DPS and APMC, which do not assume black-box setting and have access to gradients of the forward model. Results in Fig. 11 show that although our proposed method ZO-APMC

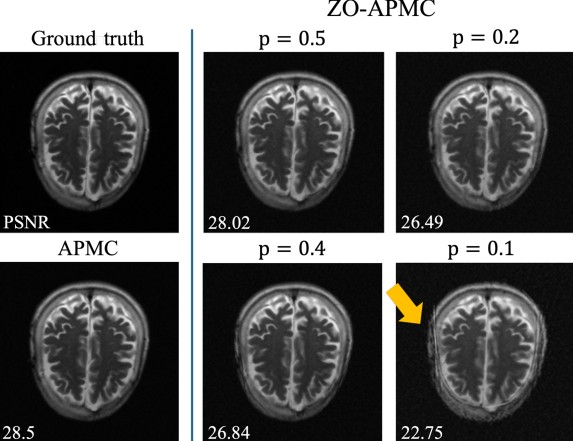

*Figure 10.* Comparison of the ground-truth brain MRI with APMC and ZO-APMC reconstructions for various probabilities $p \in \{0.1, 0.2, 0.4, 0.5\}$, using a large batch size of $b = 10^4$ and a small batch size of $b' = 10^3$. PSNR values for each reconstruction are displayed in the lower-left corner of the corresponding image.

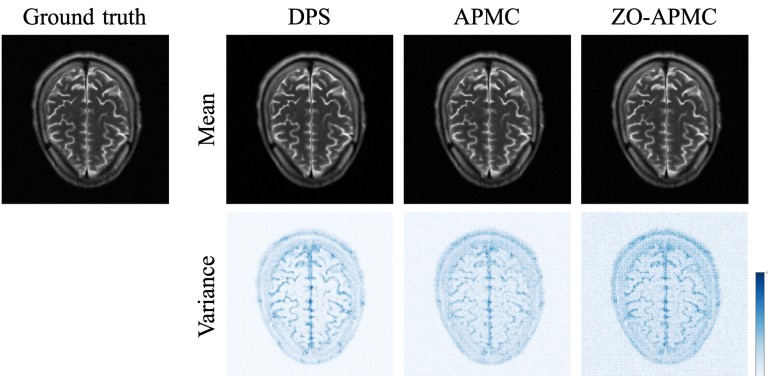

*Figure 11.* Comparison of the ground-truth brain MRI with reconstructions from ZO-APMC and the gradient-based approaches DPS and APMC. Each method generates 20 samples from the same measurements; the first row shows the mean reconstructions and the second row shows the corresponding variance maps. Owing to its variance-reduction mechanism, ZO-APMC produces variance maps comparable to those of the gradient-based algorithms despite relying on noisy evaluations of the forward model.

assumes black-box setting and uses noisy forward model evaluations to approximate the gradient of the forward model, it has similar variance compared to DPS and APMC, which assumes access to the gradients, thanks to our proposed variance-reduction mechanism.

### C.4. Black-Hole Imaging

In black-hole imaging, very long baseline interferometry (VLBI) uses an array of ground-based telescopes. Each telescope pair $(a, b)$ at time $t$ produces a complex visibility $V_t^{a,b}$. To mitigate atmospheric and thermal phase errors, visibilities are combined into noise-robust *closure* measurements (Chael et al., 2018): closure phases $\boldsymbol{y}_{t,(a,b,c)}^{\mathrm{cph}}$ and log-closure amplitudes $\boldsymbol{y}_{t,(a,b,c,d)}^{\mathrm{camp}}$. Following Sun & Bouman (2021); Zheng et al. (2025a), we use the following likelihood model:

$$\ell(\boldsymbol{y} \mid \boldsymbol{x}) = \sum_t \frac{\left\| \mathcal{A}_t^{\mathrm{cph}}(\boldsymbol{x}) - \boldsymbol{y}_t^{\mathrm{cph}} \right\|_2^2}{2\beta_{\mathrm{cph}}^2} + \sum_t \frac{\left\| \mathcal{A}_t^{\mathrm{camp}}(\boldsymbol{x}) - \boldsymbol{y}_t^{\mathrm{camp}} \right\|_2^2}{2\beta_{\mathrm{camp}}^2} + \frac{\rho}{2} \left\| \sum_i x_i - y^{\mathrm{flux}} \right\|_2^2. \tag{122}$$

Here, $\mathcal{A}_t^{\mathrm{cph}}$ and $\mathcal{A}_t^{\mathrm{camp}}$ map an image $\boldsymbol{x}$ to predicted closure phases and log-closure amplitudes, respectively; $\beta_{\mathrm{cph}}$ and $\beta_{\mathrm{camp}}$ are instrument-specific noise scales. The first two sums act as chi-squared penalties for the closure measurements, while the final term enforces the total-flux constraint with weight $\rho$ and target flux $y^{\mathrm{flux}}$. For our experiments, we use the GRMHD dataset (Wong et al., 2022), the pre-trained SGM prior (Sun et al., 2024), and the forward model implementation

and baseline methods provided by Zheng et al. (2025b). For EnKG, we adopt the hyperparameter settings recommended by Zheng et al. (2025a), while for the baseline methods we use the hyperparameters provided by Zheng et al. (2025b). For ZO-APMC, we use a batch size of $b = 1024$, set $p = 1$, and choose a ZO smoothing parameter of $\mu = 0.01$. We use $p = 1$ because the image size is relatively small ($64 \times 64$), making the large batch size practical. The per-sample runtimes (seconds) of each method are PnPDM: 25.84, DPS: 15.02, APMC: 14.32, Forward-GSG: 156.72, Central-GSG: 155.66, SCG: 213.78, DPG: 152.86, EnKG: 422.25, and ZO-APMC: 154.22, measured on an NVIDIA H100 GPU.

### C.5. Navier–Stokes Equation

In our experiments, we study the two-dimensional Navier–Stokes equations for a viscous, incompressible fluid in vorticity form on a torus. Let $u \in C([0, T]; H_{\text{per}}^r((0, 2\pi)^2, \mathbb{R}^2))$ for any $r > 0$ denote the velocity field, and let $w = \nabla \times u$ be the vorticity. The initial vorticity is $w_0 \in L_{\text{per}}^2((0, 2\pi)^2; \mathbb{R})$, the viscosity coefficient is $\nu \in \mathbb{R}_+$, and the forcing term is $f \in L_{\text{per}}^2((0, 2\pi)^2; \mathbb{R})$. The solution operator $\mathcal{G}$ maps the initial vorticity to the vorticity at time $T$, i.e. $\mathcal{G} : w_0 \mapsto w_T$. In our experiments, we implement $\mathcal{G}$ using a pseudo-spectral solver following He & Sun (2007):

$$\partial_t w(x, t) + u(x, t) \cdot \nabla w(x, t) = \nu \Delta w(x, t) + f(x), \qquad x \in (0, 2\pi)^2, \, t \in (0, T], \qquad (123)$$

$$\nabla \cdot u(x, t) = 0, \qquad x \in (0, 2\pi)^2, \, t \in (0, T], \qquad (124)$$

$$w(x, 0) = w_0(x), \qquad x \in (0, 2\pi)^2. \qquad (125)$$

The task is to infer the initial vorticity field from noisy and sparsely observed vorticity data at time $T = 1$. Since Eq. (21) admits no closed-form solution, the corresponding derivative of the solution operator is also unavailable. Furthermore, the computation of accurate numerical derivatives via automatic differentiation through the solve is challenging since the extensive computation graph can span thousands of discrete time steps.

We follow the approach in Zheng et al. (2025a;b) and first solve the equation up to time $T = 5$ starting from random Gaussian initial conditions, which are highly nontrivial due to the nonlinearity of the Navier–Stokes equations. We use the SGM-prior, which was pre-trained over 20,000 vorticity fields, and use the test set consisting of 10 samples with size of $128 \times 128$ from InverseBench. For EnKG, we use the hyperparameters recommended by Zheng et al. (2025a), while for the baseline methods we adopt the settings provided by Zheng et al. (2025b). For ZO-APMC, we set the ZO smoothing parameter to $\mu = 0.01$, batch size to $b = 1024$ and use $p = 1$. We observed degraded performance for $p < 1$, which may arise from the violation of Assumption 1, as the Navier–Stokes forward operator exhibits strong nonlinearity arising from the underlying PDE dynamics. Quantitative results are presented in Table 4. Our method outperforms Forward-GSG, Central-GSG, and SCG, and achieves performance comparable to DPG, while additionally providing theoretical guarantees of convergence to the target posterior that the baseline methods lack. Table 4 also shows that EnKG outperforms ZO-APMC in this setting. One possible explanation is that ZO-APMC estimates gradients using Gaussian perturbations, which may move inputs off the data manifold and adversely affect solver stability in highly nonlinear PDE-based problems such as Navier–Stokes. In contrast, EnKG avoids such perturbations and appears more robust in this regime. We refer the reader to Section 4.2 of Zheng et al. (2025a) for a more detailed discussion. The per-sample runtimes (seconds) of each method are Forward-GSG: 4536.87, Central-GSG: 4534.32, SCG: 2112.72, DPG: 5133.93, EnKG: 3422.25, and ZO-APMC: 4531.61, measured on an NVIDIA H100 GPU.

*Table 4.* Quantitative results for the Navier–Stokes inverse problem. For noise level $\sigma_{\text{noise}}$, the best-performing method is shown in **bold**. Baseline results are taken from Zheng et al. (2025a).

| | NRMSE ($\sigma_{\text{noise}} = 0$)↓ | NRMSE ($\sigma_{\text{noise}} = 1$)↓ | NRMSE ($\sigma_{\text{noise}} = 2$)↓ |
|---|---|---|---|
| Forward-GSG | 1.687 | 1.612 | 1.454 |
| Central-GSG | 2.203 | 2.117 | 1.746 |
| SCG | 0.908 | 0.928 | 0.966 |
| DPG | 0.325 | 0.408 | 0.466 |
| EnKG | **0.120** | **0.191** | **0.294** |
| ZO-APMC (Ours) | 0.459 | 0.463 | 0.472 |

