# OpenReview forum: "Zeroth-Order Non-Log-Concave Sampling with Variance Reduction and Applications to Inverse Problems"
_ICML.cc/2026/Conference — ICML 2026 regular_

### Official Review · Reviewer_62CN · 2026-02-18

**Soundness:** 3
**Presentation:** 3
**Significance:** 4
**Originality:** 3
**Overall Recommendation:** 5
**Confidence:** 3

**Summary:**

The paper studies the theory and algorithm of zeroth-order Langevin Monte Carlo (LMC) sampling from a non-log-concave density. They first propose a variance reduction zeroth-order sampling method that leverages the history of gradient approximations, and plug this into standard LMC for posterior sampling in diffusion to achieve ZO-APMC. The authors empirically show in diverse inverse problems that ZO-APMC achieves superior performance compared to prior art in black box inverse problem solving. Further, the authors provide rigorous theoretical (convergence) guarantees.

**Compliance With Llm Reviewing Policy:**

Affirmed.

**Final Justification:**

All concerns have been resolved. I increased my score from 4 to 5.

**Key Questions For Authors:**

1. Can the authors comment on the choice of p, and why it deviates from theoretical insights? Also, the hyperparameter values are tuned differently for each problem. It would also be valuable to add a discussion on the guidance for choosing hyperparameters, along with the stability.

2. I understand that the ZO estimate in (8) may be theoretically more desirable under the assumptions. However, currently, the intuitive motivation for such design is lacking. How is it exactly motivated from Li et al. 2021, and how does it differ?

**Limitations:**

Yes

**Strengths And Weaknesses:**

### Strenghts

1. The paper is well-written and well-motivated. The main manuscript contains just enough information to understand the arguments intuitively, yet with theoretical insights.

2. Unlike existing methods, ZO-APMC provides rigorous theoretical guarantees.

3. The experiments are complete, starting from toy experiments and extending to linear and non-linear inverse problems, finally even to PDE inverse problems.

4. The performance is strong, largely outperforming previous zeroth-order gradient methods, and on par with those that require access to gradients.

### Weaknesses

1. From the paper, it is unclear how much compute (in wall-clock time) ZO-APMC requires compared to other zeroth-order and gradient-based approaches. This should be included in the evaluation table.

2. The theory says that one should choose p << 1, but in practice relatively high values of p are selected, hinting there may be deviations from theory. For black-hole imaging, it is stated that p = 1.

3. I understand that the ZO estimate in (8) may be theoretically more desirable under the assumptions. However, currently, the intuitive motivation for such design is lacking. How is it exactly motivated from Li et al. 2021, and how does it differ?

---

> ### Author Rebuttal · Authors · 2026-03-31
>
> We thank the reviewer for their time and constructive feedback, and are happy to hear that they found our work well-written and significant. We have incorporated their suggestions into the revised manuscript and address the remaining concerns below.
>
> **Weakness 1**
>
> We add a comparison of wall-clock time to generate a sample for each method. We rerun each method to generate 100 samples for MRI and black-hole imaging, and 10 samples for the Navier-Stokes problem, and report average runtimes (seconds) in Table 1. All experiments are conducted on a single 80GB NVIDIA H100 GPU. Results marked "*" correspond to the experiments that are still running and will be included upon completion.
>
> **Table 1**: Average wall-clock time (seconds) to generate a sample for each method across inverse problems.
> | Method          | MRI  | Black-hole Imaging | Navier-Stokes |
> |----------------|:----:|:------------------:|:-------------:|
> | PnPDM           | 68.3 | 25.84              | -             |
> | DPS             | 25.44| 15.02              | -             |
> | APMC            | 23.77| 14.32              | -             |
> | Forward-GSG     | 38.12| 156.72            | 4536.87        |
> | Central-GSG     | 37.83| 155.66             | 4534.32       |
> | SCG             | *    | 213.78             | 2112.72       |
> | DPG             | 60.54| 152.86             | 5133.93       |
> | EnKG            | *    | 422.25             | 3422.25       |
> | ZO-APMC (ours)  | 50.53| 154.22             | 4531.61       |
>
> **Weakness 2**
>
> The statement $p\ll 1$ was intended to indicate that smaller values of $p$ reduce the need for large batch sizes, but it should not be interpreted as requiring extremely small (e.g., 0.01). In fact, very small $p$ leads to more conservative step sizes due to to the bound on $\gamma$ in Theorems 1 and 3, resulting in slower convergence. Therefore, moderate values of $p$ provide a better practical trade-off. We replace the symbol "$\ll$" with "<" and clarify this in the revision.
>
> In the black-hole imaging experiments, we set $p=1$ as it yielded slightly better performance. One explanation is that the nonlinear forward model can have a large Lipschitz constant. Although its gradients are bounded due to the compact domain (images are normalized to $[-1,1]$) and differentiability, a large Lipschitz constant makes it more difficult to control the estimation error in Proposition 1. This further slows convergence, since the iteration complexity scales with $L_m^4$.
>
> Moreover, our theoretical results also cover the case $p=1$, albeit with batch sizes that scale with the dimension, which is computationally feasible in lower-dimensional settings such as black-hole imaging. In contrast, for high-dimensional problems such as MRI, choosing $p<1$ is crucial to achieve a good performance.
>
> **Weakness 3**
>
> **Q1:** Intuitive design is lacking. How is (8) exactly motivated from Li et al. 2021?
>
> The ZO estimate in (8) is motivated by the variance reduction idea of in Li et al 2021. Intuitively, computing a fresh gradient estimate at every iteration ($p=1$) requires a large batch size that scales with dimension $d$. To avoid this, our estimator (8) and Li et al 2021 alternates between (i) computing a fresh estimate and (ii) reusing the previous estimate (e.g., $g_{k-1}$) while estimating the change in gradients between successive iterates using a smaller batch. When the potential function values between consecutive iterates do not change significantly (Assumption 1), this difference of gradients can be estimated accurately, reducing computational cost per iteration.
>
> **Q2:** How does (8) differ from Li et al. 2021?
>
> Compared to Li et al. 2021, our estimator extends this idea from non-convex first-order optimization to the more challenging zeroth-order, non-log-concave sampling setting. In this setting, gradients are replaced by noisy, biased, and high-variance estimates constructed from function evaluations, which are further coupled with the discretization error of Langevin dynamics. In particular, ZO estimates introduce a bias-variance trade-off governed by the smoothing parameter $\mu$: as $\mu\to 0$, the bias vanishes (Lemma 1b), but the variance, and consequently the discretization error in Proposition 1 (Term 2), increases. Our method and analysis account for this trade-off and establish convergence guarantees with per-iteration cost that does not scale with the dimension.
>
> **Key questions for authors:**
>
> **Q1:** To select $p$, we first performed a grid search over $b\in${$10^2, 10^3, 10^4$} for the MRI experiments and chose $b=10^4$, as $10^3$ performed poorly. We then tuned $p$, starting from $p=0.5$ and decreasing it in steps of 0.1. We observed a significant performance drop below $p=0.2$ (see Fig. 9 in the Appendix). Based on this, we selected $p=0.2$. We include additional discussion on hyperparameter selection for each experiment in the revision.
>
>
> **Q2:** We answered this question under *Weakness 3*.

---

> > ### Author Rebuttal · Reviewer_62CN · 2026-04-01
> >
> > My concerns have been addressed and I will keep my positive score. I would advise the authors to include the response to W3 in the revised manuscript.

---

> > > ### Author Response · Authors · 2026-04-03
> > >
> > > Thank you for your positive feedback and for recognizing the significance of our work. We appreciate your constructive suggestions and we are glad our responses addressed your concerns. We will include all suggestions in the final version.

---

### Official Review · Reviewer_DQ7h · 2026-02-23

**Soundness:** 3
**Presentation:** 3
**Significance:** 3
**Originality:** 3
**Overall Recommendation:** 5
**Confidence:** 5

**Summary:**

In this paper the authors propose a Langevin sampling method to sample from distributions of the form $p(x)\propto e^{-f(x)}$. Specifically, they are interested in cases where the gradient $\nabla f$ is not directly accessible and is therefore approximated using a stochastic finite differences approximation. Since the accuracy of a "regular" stochastic finite differences approximation scales poorly with the dimension, the authors consider an iterative update of the gradient approximation. In each iteration, with probability $p$ the gradient is estimated using the (Monte Carlo) average of a large batch of finite difference approximations is used, and with probability $1-p$ the previous estimate is corrected using a cheaper correction. Assuming Lipschitz continuity of $f$ and $\nabla f$, convergence in Fisher divergence of the ergodic mean of the distributions is proven and, under a log Sobolev inequality, also in TV distance.

As a second contribution the authors combine the approach explained above with a score based prior. In this case, similar convergence results are provided assuming sufficient (uniform) accuracy of the score estimate.

The authors provide multiple different experiments comparing the proposed zero order method for different parameter choices as well as with other methods.

**Compliance With Llm Reviewing Policy:**

Affirmed.

**Final Justification:**

All my concerns have been adressed.

The addressed problem in the paper is relevant and interesting (zero order Langevin sampling, i.e., when gradients cannot be computed directly for some reason). The solution (based on other works) is to use a finite differences type estimator for the gradient where at random either a high accuracy but expensive estimate is used or a lower accuracy cheap estimate. In total this leads to a balance of accuracy and computational effort. The paper is technically strong and correct and I think provides a good contribution!

**Key Questions For Authors:**

1. It would be great if assumption 1 could somehow be relaxed. I understand, that this might be difficult, but linear growth of $f$ is rather restrictive. Especially, whenever dealing with squared L2 losses as is customary when assuming inverse problems with Gaussian noise, or in general whenever having any Gaussian components, $f$ will grow quadratically in some directions, At least this should be noted as a limitation and clearly stated in the numerical experiments, whenever this assumption is violated. In such cases, the experiments, while interesting, are not covered by the theory.

2. When briefly thinking about it, it seems to me that the standard case for assumption 4 to be fulfilled is Lipschitz continuity of $\nabla h$. I would suggest to either state this or provide an example where $\nabla h$ is not Lipschitz but assumption 4 still holds.

3. In eq (28). I think $Cov(e_{k-1}| F_{k-1})$ should be $Cov(e_{k-1}| F_{k-1}, x_k)$ and it should anyway be zero since conditioning on $F_{k-1}$, renders $e_{k-1}$ deterministic, I suppose.

4. I think equations (48)-(54) can entirely be removed and simply be replaced by the result of prop 1 and an estimate on $\sigma$. Similarly equations (112)-(118) are simply a copy of the previous ones. It was very confusing to me reading the proof and wondering why prop 1 is not used. Also I do not see where the factor (1-p) in (55) is coming from (same for (119) )

5. In equation (60) it is unclear to me where $\nabla f(x_t)$ is coming from since $g_k$ only makes use of $\nabla f$ evaluated at $x_{k\gamma}$ and $x_{(k-1)\gamma}$. If there is an additional $\pm$ of a term this is worth mentioning in more detail. (Same for eq (125), (126) with $\nabla \log \pi(x_t)$ )

6. The argument after eq (96) is not correct. The fact that $c_k = O(k^{-1/2})$ does not imply that it is monotonically increasing. In particular since the expression in (96) contains an increasing term $1-k^{-1/2}$ this should be argued in more detail. For instance one could replace $k$ by a real variable $t$, compute the derivatives of the terms with $k$ resp. $t$ and show that they are non-positive. (Same for (164))

8. Smaller remarks:
* In section 2.1 paragraph after (3) one expectation is wrt x but should be wrt u
* In eq (8) the indexing of $g$ is inconsistent. Either always use only natural indexes or always real indexes.
* In the statement of corollary 2 in the main part the Fisher divergence should probably be the TV norm.
* I cannot understanf what exactly the shift by $\pm 1$ in the paragraph "Statistical evaluation" is supposed to mean. In particular since in Fig 1 b, the images are all scaled to roughly $[-1, 1]$
* I do not understand what the red crosses in Fig 1 (b) precisely represent. Also the figures are poorly formatted (font way too small)
* (20) $\mu$ and $\nu$ should be the same.
* In the conditional expectation after (26) there is one | too many.
* I would suggest writing (34) already in traces as it makes the argument easier (no need for ordering matrices). Also in (35) then the traces are missing according to the preceding sentence.
* In (93) the term $(e_k^2-e_{k-1}^2)$ should be removed.
* $\mathcal{E}$ in the paragraphs after (99) is never introduced. Also the last argument is a little rushed. For instance, I would have though, $FI(\nu|\mu)=0$ implies that $d\nu / d\mu = 0$ only $\nu$-a.e. etc. Maybe this can be detailed a little. (Same at page 32)
* In Section C.1 it is not mentioned what actually $f(x)$ is I think.
* Please add *exact* references. I find it quite unsettling, when authors cite specific results and reference an entire book (!) wiithout any information as to where the result can be found (This concerns Lemmas 1-4).

**Limitations:**

In my opinion, the authors should address that limitations imposed by assumption 1 which is in particular violated in the shown numerical experiments.

**Strengths And Weaknesses:**

Soundness:
The paper is mostly technically sound. I found several smaller issues in the proofs which are listed  below in questions.

Presentation:
The presentation in the main paper is good. However, in the appendix sometimes I found parts of the proofs a little confusing which is also related to the remarks made below.

Significance:
I think the paper is significant. As the authors mention there are many interesting use-cases, where the gradients cannot be directly evaluated. The significance might, however, be limited by the assumption of Lipschitz continuity of $f$ which is violated very often for inverse problems and by the dimensionality scaling of $d^7$. However, I believe that the results are relevant nonetheless.

Originality:
My experience with zero order methods is limited which is why I cannot judge the originality very strongly. However, I found the ideas to be interesting and novel. While the treatment of the estimated score seemed rather standard and "easy" to me, I found the proposed gradient estimate in (8) to keep the complexity of the gradient update limited very nice.

---

> ### Author Rebuttal · Authors · 2026-03-31
>
> We thank the reviewer for their time and constructive feedback, and are happy to hear that they found our work interesting and novel. We have incorporated their suggestions into the revised manuscript and address the remaining concerns below.
>
> **Q1: ...linear growth of $f$ is rather restrictive...should be noted as a limitation...**
>
> **A:** We agree that Assumption 1 is restrictive, and in the revised manuscript we explicitly identify the experiments that violate this assumption and discuss this as a limitation.
>
> Regarding the applicability of Assumption 1, we note that in image processing the target distribution $\pi$ is supported on a compact set $\mathcal{M}$ [1,2] as images are normalized to $[0,1]$ or $[-1,1]$. Under this condition, if the forward model $\mathbf{A}$ is continuously differentiable, then the likelihood potential has bounded gradient on $\mathcal{M}$, i.e., it is Lipschitz continuous. Thus, Assumption 1 holds in our synthetic and MRI reconstruction experiments (with linear forward models), as well as in the black hole imaging experiments with a differentiable nonlinear forward model. However, for the Navier-Stokes problem, where the forward model is not differentiable, the assumption is not valid.
>
> A possible relaxation is to replace Lipschitz continuity with $\alpha$-Hölder continuity ($\alpha\in(0,1]$). Balasubramanian et al. [3] derive iteration complexity results for LMC with Gaussian-smoothed potentials under this condition. Extending these results to stochastic gradient and/or zeroth-order setting with variance reduction is nontrivial, and we leave this for future work.
>
> **Q2: ..assumption 4...Lipschitz continuity of $\nabla h$...**
>
> **A:** Lipschitz continuity of $\nabla h$ is stronger assumption than Assumption 4. In the revised manuscript, we discuss a counterexample that verifies Assumption 4 but does not satisfy Lipschitz continuity.
>
> **Q3&4: ...why prop 1 is not used...**
>
> **A:** In the revised manuscript, we replace (48)-(54) and (112)-(118) with Prop. 1 in our derivations and make the corresponding corrections.
>
> **Q5: ...unclear to me where $\nabla f(\mathcal{x}_t)$ is coming from...**
>
> **A:** We proceeded as follows:
>
> $E[||g_k||^2] = E[||g_k - \tilde{\nabla} f_\mu(x_{k\gamma}, u_k) + \tilde{\nabla} f_\mu(x_{k\gamma}, u_k) - \nabla f(x_{k\gamma}) + \nabla f(x_{k\gamma}) - \nabla f(x_{t}) + \nabla f(x_{t})||^2]$
>
> and then we used Cauchy-Schwarz inequality. For (126), in addition to these terms, we added and subtracted $\nabla h(x_t)$, $\nabla h(x_{k\gamma})$, and $\nabla h_{\sigma_k}(x_{k\gamma})$. We clarify this in the revision.
>
> **Q6: ... $c_k=O(k^{-1/2})$ does not imply non-increasing sequence...**
>
> **A:** In the revised manuscript, we precisely show that $c_k$ is non-increasing sequence as follows
>
> Since $k\in ℕ_{+}$, we have $1 - \frac{1}{k^{1/2}} \leq 1$. Similarly, $3 + \frac{b'}{2dk^2} \leq 3 + \frac{b'}{2d}$. Thus,
> \begin{equation}
>     c_k \leq \frac{1}{k^{1/2}}\sqrt{\frac{b'}{680L_m^2d}}\left(3+\frac{b'}{2d}\right) + \frac{1}{k^{9/4}}\sqrt{\frac{b'}{680L_m^2d}}\frac{85b'}{12\times 680d}.
> \end{equation}This implies that $c_k$ is a non-negative, non-increasing sequence for $k\in ℕ_{+}$.
>
> **Q7, Remark 4: ...shift by ±1...**
>
> **A:** The ±1 shift separates the two Gaussian modes (male and female), which would otherwise overlap after normalization to $[-1,1]$. The posterior in Fig. 1b appears roughly in $[-1,1]$ due to the selected test image.
>
> **Q7, Remark 5: ...red crosses in Fig. 1b...**
>
> **A:** Fig. 1 shows that, for a fixed per-iteration cost (e.g., $p\times b$), the Fisher information falls below a threshold (e.g., $ε = 0.01$). Each red cross corresponds to ZO-APMC with 1000 samples over 20 trials, reporting averaged Fisher information is below the threshold. We increase the font sizes and size of the figure to improve readability in the revision.
>
>
>
>
> **References**
>
> [1] Marien Renaud, Jean Prost, Arthur Leclaire, and Nicolas Papadakis. Plug-and-play image restoration with stochastic denoising regularization. In Proceedings of the 41st International Conference on Machine Learning. JMLR.org, 2024.
>
> [2] Convergence of denoising diffusion models under the manifold hypothesis, De Bortoli, Valentin, Transactions on Machine Learning Research, 2022.
>
> [3] Balasubramanian, K., Chewi, S., Erdogdu, M. A., Salim, A., and Zhang, S. Towards a theory of non-log-concave sampling: first-order stationarity guarantees for langevin monte carlo. In Conference on Learning Theory, pp. 2896–2923. PMLR, 2022.

---

> > ### Author Rebuttal · Reviewer_DQ7h · 2026-04-01
> >
> > Some remaining remarks.
> >
> > > Regarding the applicability of Assumption 1, we note...
> >
> > I do not think this works. Your paper relies on the assumption that the potential function $f:\mathbb{R}^d\rightarrow\mathbb{R}$. If you formally want to work with data supported on a bounded set, you would have to allow $f(x)=\infty$.
> >
> > ** Q.6** This is still not a valid argument: A sequence which is upper bounded by a decreasing sequence does not need to be decreasing itself. Please correct this for the final version (as previously stated: simply in the terms containing $k$ replace by $x\in\mathbb{R}$ and check the derivative)

---

> > > ### Author Response · Authors · 2026-04-03
> > >
> > > Thank you again for your constructive feedback and for engaging with the theoretical parts of our work. We appreciate your suggestions for improving clarity and rigor in the proofs, and we will incorporate these refinements carefully in the final revision. Here are our responses to the remaining remarks.
> > >
> > > Regarding the Assumption 1, we acknowledge the gap between theory and experiments. This can be mitigated by enforcing a compact domain with probability 1 via projection or clipping at each LMC iteration, as explored in prior work [1]. In the revision, we will add a remark clarifying this limitation and outline how such a projection could be incorporated for a more rigorous treatment in future work.
> > >
> > > Regarding **Q6**, we agree with the reviewer and will revise the argument by introducing a continuous variable $t$ and analyzing its derivative to show that $c_k$ forms a decreasing sequence.
> > >
> > > **References**
> > >
> > > [1] Bubeck, Sébastien, Ronen Eldan, and Joseph Lehec. "Sampling from a log-concave distribution with projected Langevin Monte Carlo." Discrete & Computational Geometry 59.4 (2018): 757-783.

---

### Official Review · Reviewer_EWYy · 2026-03-05

**Soundness:** 3
**Presentation:** 1
**Significance:** 3
**Originality:** 3
**Overall Recommendation:** 5
**Confidence:** 5

**Summary:**

This paper develop a sampling strategy based on Langevin dynamic without evaluating the gradient of the potential, only evaluating the potential itself. It also adapt this strategy for inverse problem in imaging incorporating forward model in the process. Convergence guarantees are provided in a non-log-concave setting.

**Compliance With Llm Reviewing Policy:**

Affirmed.

**Key Questions For Authors:**

Questions:
- Could you comment Assumption 1 ? What is the restriction of it ?
- Could you comment Assumption 2 ? What kind of function verify it ?
- In Theorem 1 (fully stated in Appendix B.3), how the condition on $\gamma$ impose a condition on $N$ ?
- Could you comment Assumption 3 ? Which kind of functions verify it ? Why Assumption 3 is needed to obtain a result in TV norm ?
- Could you comment the link between assumption 4 and bounded denoiser [1] ?
- How do you justify Assumption 5 when [2] show in Lemma C.1 that if the prior is supported in a bounded set then the dependency is $1/\sigma^2$ ?
- Figure 1(a), why do you use $20$ random seeds ? Your convergence results doesn't work with a unique seed ? Is the ZZO-APMC ergodic ?
- Have you compare experimentally your method with the ZO-LMC described in equation (5) ?

[1] Stanley H Chan, Xiran Wang, and Omar A Elgendy. “Plug-and-playADMM for image restoration: Fixed-point convergence and applica-tions”. In: IEEE Transactions on Computational Imaging 3.1 (2016),pp. 84–98.

[2] Convergence of denoising diffusion models under the manifold hypothesis, De Bortoli, Valentin, Transactions on Machine Learning Research, 2022

**Limitations:**

The limitations are not disccused in details. A paragraph should be added at the end of the paper with the clear limitations of the paper.

**Strengths And Weaknesses:**

I reviewed in details the paper but not the proofs in Appendices.

Strength:
- The motivation of the paper on the importance of zero-order methods is well explained.
- The experiments are strong and diverse.
- The approach is original and inovative.


Major Weaknesses:
- All the theoretical statement are stated with an "informal" style making them unclear. As the main contribution of the paper is theoretical, I do not support this choice. The theoretical results should be stated precisely.
- The computational cost of methods is not discussed in details, in the experimental part there is no quantitative comparison. I suggest to add the computational costs to Table 1-2.
- The paper is not well written with many typos and imprecisions (some are repported below).
- Assumptions are not commented; In particular, the restriction implies by the assumptions are not stated.
- There are unprecise references : After Assumption 1, it is claims that [4] study L-Lipschitz objective function where it focuses on L-smooth objective function. In section 4.2, Zbontar et al. is said to be written in "1811".

Minor Weaknesses/Typos:
- In the introduction, please precise what do you call a "smooth" $f$.
- Equation (1): there is no square root on the Brownian motion for Langevin dynamic.
- Algorithm (2) is more commonly called Unajusted Langevin Algorithm (ULA) [1]. I recommand to adopt this terminology.
- Just after equation (3), it must be $\nabla f_{\mu}(x) = \mathbb{E}_{u}(\tilde \nabla f_{\mu}(x,u))$.
- For Lemma 1, please precise that this is in Appendix.
- Lemma 1(a), it must be $C_{L_{\mu}}^{(1,1)}$.
- 5 line after equation (3) : it should be $\tilde \nabla f_{\mu}(x_{k\gamma}, u_{k})$ or $\nabla f_{\mu}(x_{k\gamma})$ but $\tilde \nabla f(x_{k\gamma})$ has not been introduced. Same error three line after equation (5).
- For the Tweedie formula please refer to the initial works that introduce this formula [2,3] and not only to the recent Efron work.
- Line 4 of section 3.1 : "computationally"
- Line 7 after Assumption 1 : error on the equation number, (16) is in the Appendices.
- The reference "Liu et al." after Assumption 1 is not precised, which reference are you refer to ? There are 4 differents papers with this first author last name in the bibliography.
- The construction in equation (10) is not precise. For which $t$ this equation is defined ?
- The term "Gaussian-smoothed score" is used in Proposition 1 but not defined before, what are you refer to ?
- In Proposition 1 : the quantity $p$ is not defined.
- Theorem 1 : the quantities $N_{\gamma}$ and $\bar{\nu_{N_{\gamma}}$ are not defined.
- I suggest to temper the statement after Assumption 5. This assumption has been used by Sun and al., 2024, but it is not standard in Langevin dynamics study. This assumption is very restrictive.
- I suggest to add in Table 3 two aditionnal MCMC, the historical [5] and the more recent [6]. I suggest to comment these references [5,6] also in section A.2 together with the Gibbs sampling references (Coeurdoux and al., 2024).
- Section 4.1: I do not understand the dimension of $A$ to be $155\times 1024$ could you explain that ?

[1] Exponential convergence of Langevin distributions and their discrete approximations, Roberts, Gareth O and Tweedie, Richard L, 1996

[2] An empirical Bayes approach to statistics, Robbins, Herbert E, In Proceedings of the Third Berkeley Symposium on Mathematical Statistics and Probability, 1956

[3] An empirical Bayes estimator of the mean of a normal population, Miyasawa, Koichi and others, Bull. Inst. Internat. Statist, 1961

[4] PAGE: A simple and optimal probabilistic gradient estimator for nonconvex optimization, Li, Zhize and Bao, Hongyan and Zhang, Xiangliang and Richt{\'a}rik, Peter, International conference on machine learning, 2021

[5] Efficient bayesian computation by proximal markov chain monte carlo: when langevin meets moreau, Durmus, Alain and Moulines, Eric and Pereyra, Marcelo, SIAM Journal on Imaging Sciences, 2018

[6] From stability of Langevin diffusion to convergence of proximal MCMC for non-log-concave sampling, Renaud, Marien and de Bortoli, Valentin and Leclaire, Arthur and Papadakis, Nicolas, NeurIPS 2025-39th Annual Conference on Neural Information Processing Systems

---

> ### Author Rebuttal · Authors · 2026-03-31
>
> We thank the reviewer for their time and constructive feedback, and are pleased that they found our work original and innovative. We have included their suggestions into the revised manuscript and address the remaining concerns below.
>
> **Weaknesses**
>
> 1. We understand the reviewer's point and state all theoretical results formally in the revised manuscript. To maintain readability and flow of the main text, we present the iteration complexities in the main text and upper bounds on FI and TV square in Appendix.
>
> 2. We added a quantitative comparison. Please see our response to Reviewer 4 (62CN, Weakness 1).
>
> 3. We have carefully proofread the manuscript and corrected all identified typos and imprecisions, including those noted by all reviewers. We will also perform an additional round of proofreading in the final version.
>
> 4. Please refer to our detailed responses under "Key questions".
>
> 5. Regarding the reference [1] after Assumption 1, our intent was to compare the assumptions required for variance control in first-order vs. zeroth-order settings. In [1], the analysis relies on $L$-smoothness, i.e., Lipschitz continuity of the stochastic gradients $\nabla f(x,u)$ with respect to $x$ (where $u$ denotes the source of stochasticity). In contrast, we work with ZO estimates $\tilde{\nabla}f_\mu(x,u)$ constructed from function evaluations. As a result, variance control relies on Lipschitz continuity of the function evaluations, rather than Lipschitz continuity of the stochastic gradients. We clarify this distinction in the revision. We also correct the fastMRI reference with year "1811", which was due to a BibTeX formatting issue.
>
> **Minor Weaknesses**
>
> **Q13:** By "Gaussian-smoothed", we referred to $\nabla f_\mu (x_{k\gamma})$ and clarify this part.
>
> **Q18:** By "dimension of $A$ to be $155\times 1024$", we meant that $A\in \mathbb{R}^{155 \times 1024}$, which makes the inverse problem ill-posed as $y=Ax$ does not have a unique solution.
>
> **Key questions:**
>
> **Q1:** We acknowledge that Assumption 1 is restrictive and may not hold globally for simple distributions such as Gaussian. However, it holds for differentiable potentials on compact domains, which commonly arise in practice via normalization or gradient clipping. In our synthetic, MRI, and black hole imaging experiments, the forward model is differentiable and images are normalized to $[-1,1]$, so Assumption 1 holds. This assumption ensures that the correction-term estimates are sufficiently accurate to control the error.
>
> **Q2:** This is standard in sampling literature, ensuring controlled variation of the gradient term and stability of the discretization in Langevin dynamics. It holds for common potentials such as quadratic (Gaussian) potentials. In the context of inverse problems, it holds for linear inverse problems with Gaussian noise but does not hold for nonlinear inverse problems in general.
>
> **Q3:** The bound on $\gamma$ and $p\in(0,1]$ imposes $N \geq$ max{$L_m^4/d, 416/(b'd)$}, which is satisfied by the derived iteration complexity. We add this verification to our proofs in revision.
>
> **Q4:** Assumption 3 enforces concentration by requiring sufficiently growth of the potential at infinity, preventing heavy tails. It holds for a broad class of distributions such as log-concave, as well as certain non-log-concave cases (e.g., Gaussian convolutions of bounded-support distributions) [2]. It is needed because FI provides only local (gradient-based) control and does not imply global closeness in TV. Assumption 3 converts this into global concentration.
>
> **Q5:** Assumption 4 implies that, under an MMSE denoiser, the score induced by the denoiser converges to the true score as $\sigma_k\to 0$. In contrast, the bounded denoiser condition only controls the magnitude of the residual and does not ensure alignment with true score, and is therefore weaker.
>
> **Q6:** The bound on score network norm is only for discrete noise levels (i.e., $\sigma_k$), whereas Lemma C.3 provides a stronger uniform bound over continuous time. Moreover, our results can still be obtained by assuming $C/\sigma_k^2$ with different $\sigma_k$ schedule.
>
> **Q7:** ZO-APMC operates with a single seed. We average over 20 runs to estimate FI due to inherent randomness, similar to [3]. It is ergodic in the sense that it converges in time-averaged FI.
>
> **Q8:** Yes, see Fig. 7 and Fig. 8 in Appendix.
>
> **References**
>
> [1] Li, Z., Bao, H., Zhang, X., and Richtarik, P. Page: A simple and optimal probabilistic gradient estimator for nonconvex optimization. In International conference on machine learning, pp. 6286–6295. PMLR, 2021.
>
> [2] S. Chewi, M. A. Erdogdu, M. B. Li, R. Shen, and M. Zhang. “Analysis of Langevin Monte Carlo
> from Poincar´e to log-Sobolev”. In: arXiv e-prints, arXiv:2112.12662 (2021).
>
> [3] Sun, Y., Wu, Z., Chen, Y., Feng, B. T., and Bouman, K. L. Provable probabilistic imaging using score-based generative priors. IEEE Transactions on Computational Imaging, 2024.

---

> > ### Author Rebuttal · Reviewer_EWYy · 2026-04-01
> >
> > My main concern was on the informal formulation of the theorem, the computational cost of the methods and the assumption discussion.
> > All these point have been well answer by the authors.
> > I strongly recommend to include these modifications in the final version : formal theorems, table of computational costs (not only in Appendix but in the main text in Table 1-2) and the discussion on the assumptions that is well explained in the authors' answers.
> >
> > Under these modifications, I think that this paper is valuable and should to be published.

---

> > > ### Author Response · Authors · 2026-04-03
> > >
> > > Thank you for your detailed and constructive feedback. We appreciate your careful reading and helpful suggestions. We are glad that our responses addressed your concerns and will ensure all revisions are incorporated into the final manuscript.

---

### Official Review · Reviewer_5jpk · 2026-03-06

**Soundness:** 4
**Presentation:** 3
**Significance:** 3
**Originality:** 3
**Overall Recommendation:** 5
**Confidence:** 4

**Summary:**

The paper analyses convergence of the Langevin Monte Carlo algorithm under zero-order approximations, i.e., when one has to approximate the derivative part on the score of the target distribution using only black box access to the evaluation of the log likelihood.
It starts by proposing an estimate of the score via a mechanism that allows for a trade of in computational efficiency and variance of the estimator. They propose a bound on the MSE of the estimator and then analyse the impact on the convergence of the LMC, with an increasing number of hypothesis.

Then, the paper generalizes those theorems to the case of posterior sampling with a score-based generative prior and also provide numerical evaluation in several pertinent cases.

**Compliance With Llm Reviewing Policy:**

Affirmed.

**Final Justification:**

I had already a positive opinion which was confirmed by the communication with the authors and other reviewers.

**Key Questions For Authors:**

Main questions:
1. Why did the authors did not provide a numerical evidence of weak convergence in the numerical parts? It seems that this could be feasible on top of the convergence of the FI for the synthetic experiment.
2. Why does EnKG works so much better than the proposed approach for the navier stokes problem (Table 4)? I would like to see a more detailed discussion of such behavior, specially that in the main this issue is avoided.
3. Why not use proposition 1 for bounding $e_k$ page 17?


Minor questions, typos and suggestions:
1. equation (1) has the brownian inside the square root.
2. I believe in the last paragraph of page two, the expected value is w.rt. u and not x.
3. Below eq 6, the authors define $\pi_{\sigma_k}^{\sigma_k}$ while i believe it must be $\pi_{\sigma_k}^{\alpha_k}$.
4. In the first paragraph of section 2.3, FI is not yet defined.
5. There is an error in the gradient $\upsilon$ of the KL after equation (7).
6. Section 4.2 has a citation dating from 1811. While not impossible, I don't believe arxiv existed back then.
7. $C_\mu^{1, 1}$ is not defined in lemma 1.
8. Typo in the text right after eq 26
9. After equation 31 $\mathbf{\delta}_k$ becomes $\delta_k$ (loose boldness).
10. Are the covariances in (34) conditional on $x_k, x_{k+1}$?
11. It seems that $\mathbb{E}[Cov(\delta_k)]$ should be replaced by $Cov(\delta_k)$ in eq 33.
12. I don't think there is a $d$ in eq 34. If that was true, in eq 36 one would have $d^2$.
13. Eq (37) should have a $\upsilon_t$.
14. Equation (93) should not have $e_k^2 - e_{k-1}^2$

**Strengths And Weaknesses:**

* Soundness: The paper is overall technically sound. While there are some small mistakes or typos on the proofs in the appendix (some also in the main), they are minor and the reader is able to follow the proofs.  The numerical part also investigates several important problems where the unavailability of the gradients of the "simulator" are real.

* Presentaiton: The paper is clearly written and abeilt some minor typos and small imprecisions, which I point out in the following sections.

* Significance: The paper adresses a main challenge in solving ill-posed posterior sampling coming from diverse "science" domains.  The hypothesis concerning the score-based generative model are a bit too restrictive, namely Assumption 4 and 5, that is bounded error for the score approximation over the whole space w.r.t. what is actually the state of the art on convergence of score-based generative models [2], [3], but as the current work is the first one to propose an actual end-to-end error estimation of the procedure, I still consider the paper an important work in the field.

* Originality: While the theoretical analysis follows closely [1] (as mentionned in the paper), the paper applies it to a previously unexplored method (ZO-LMC) while also proposing an estimator, that while also linked to the one in [4] renders the approach original. Furthermore, the application to posterior sampling with score-based generative models add to the originality of the paper.

[1] Balasubramanian, K., Chewi, S., Erdogdu, M.A., Salim, A. &amp; Zhang, S.. (2022). Towards a Theory of Non-Log-Concave Sampling:First-Order Stationarity Guarantees for Langevin Monte Carlo. <i>Proceedings of Thirty Fifth Conference on Learning Theory</i>, in <i>Proceedings of Machine Learning Research</i> 178:2896-2923 Available from https://proceedings.mlr.press/v178/balasubramanian22a.html.

[2] Silveri, M. G., & Ocello, A. (2025, January). Beyond log-concavity and score regularity: Improved convergence bounds for score-based generative models in W2-distance. In Forty-second International Conference on Machine Learning.

[3] Gao, X., Nguyen, H. M., & Zhu, L. (2025). Wasserstein convergence guarantees for a general class of score-based generative models. Journal of machine learning research, 26(43), 1-54.

[4]Li, Z., Bao, H., Zhang, X. &amp; Richtarik, P.. (2021). PAGE: A Simple and Optimal Probabilistic Gradient Estimator for Nonconvex Optimization. <i>Proceedings of the 38th International Conference on Machine Learning</i>, in <i>Proceedings of Machine Learning Research</i> 139:6286-6295 Available from https://proceedings.mlr.press/v139/li21a.html.

---

> ### Author Rebuttal · Authors · 2026-03-31
>
> We thank the reviewer for their time and constructive feedback, and are pleased that they found our work technically sound, clearly presented, and important work in the field. We have included their suggestions into the revised manuscript and address the remaining concerns below.
>
> **Q1: Why did the authors not provide a numerical evidence of weak convergence?**
>
> **A:** We thank the reviewer for this suggestion. While we did not include experiments on weak convergence, as we focused on Fisher information, we agree that numerical evidence of weak convergence would further strengthen the validation of our theoretical results. We therefore rerun the numerical validation experiments in Fig. 1b, using decreasing schedules for $\gamma_k$, $p_k$, $ε_{σ_k }$, $\mu_k$, and an increasing schedule for $b_k$, and compute the approximate KL divergence between the sample distribution and the target distribution on a grid. For numerical stability, we impose lower limits on the decreasing schedules and an upper limit on $b_k$. The results are presented in a figure available at this link: https://imgur.com/a/mIeDOUi. They show that ZO-APMC converges in KL divergence with different probability values of $p\in${$0.3, 0.5, 0.75, 1.0$}, which also imply weak convergence. We include these results and additional details in the revised manuscript.
>
> **Q2: Why does EnKG work so much better for the Navier-Stokes problem?**
>
> **A:** We have two justifications for this.
>
> 1. The accuracy of gradient estimate $\nabla f(x_{(k+1)\gamma})$ in the correction branch of $g_k$ (used with prob. $1-p$) depends on Assumption 1, which requires the likelihood potential to be Lipschitz continuous. When violated, the estimate for $\nabla f_\mu(x_{(k+1)\gamma}) \approx g_k + \tilde{\nabla}f_{\mu}(x_{(k+1)\gamma}, u_{k+1}) - \tilde{\nabla}f_{\mu}(x_{k\gamma}, u_{k+1})$ becomes inaccurate and its error cannot be controlled. In our numerical, MRI, and black hole imaging experiments, the forward model is differentiable and supported on a compact domain (images normalized to $[-1,1]$), so Assumption 1 holds and performance is strong. In contrast, for Navier-Stokes problem, the forward model is not necessarily continuously differentiable [1,2], degrading the correction step and leading to errors. While using $p=1$ (i.e., no correction branch) with large batch size can improve accuracy, it is computationally expensive due to the computational cost of PDE solves at each function evaluation.
>
> 2. ZO-APMC uses zeroth-order gradient estimates via Gaussian perturbations. These perturbations may move inputs off the data manifold and can violate solve stability in PDE-based problems such as Navier-Stokes. In contrast, EnKG avoids such perturbations and is more robust in this regime. Please see Section 4.2 in [1] for more discussion of this.
>
> Overall, ZO-APMC is most effective in settings when the forward model satisfies Assumption 1, as demonstrated in our experiments, and provides theoretical convergence guarantees to the target distribution. We include this discussion in the revised manuscript.
>
> **Q3: Why not use proposition 1 for bounding $e_k$ page 17?**
>
> **A:** We originally included the alternative derivation to provide additional insights but we agree that using Proposition 1 improves clarity. We now use Proposition 1 in our derivations and avoid any repetition.
>
> **Minor questions, typos and suggestions:**
>
> 10. No, they are not. From (32) to (33), we took the expectation of both sides with respect to $F_{k-1}$ and we used the fact that Cov($e_{k-1}$) = $\mathbb{E}[e_{k-1}e_{k-1}^T]$.
>
> 11. This cannot be done because the expectation is with respect to $x_k$ and $δ_k$ depends on $x_k$, so the two are not independent.
>
> 12. Yes, we revise this part.
>
> 10,11,12: To improve clarity in the proof of Proposition 1, we replace the covariance-based steps with (43)-(54) and omit (46)-(47) in the revision.
>
> **References**
>
> [1] Zheng, H., Chu, W., Wang, A., Kovachki, N. B., Baptista, R., and Yue, Y. Ensemble kalman diffusion guidance: A derivative-free method for inverse problems. Transactions on Machine Learning Research, 2025a. ISSN 2835-8856. URL https://openreview.net/forum?id=XPEEsKneKs.
>
> [2] Zheng, H., Chu, W., Zhang, B., Wu, Z., Wang, A., Feng, B. T., Zou, C., Sun, Y., Kovachki, N., Ross, Z. E., et al. Inversebench: Benchmarking plug-and-play diffusion priors for inverse problems in physical sciences. arXiv preprint arXiv:2503.11043, 2025b.

---

> > ### Author Rebuttal · Reviewer_5jpk · 2026-04-01
> >
> > My concerns have been addressed and I keep my positive score.

---

> > > ### Author Response · Authors · 2026-04-03
> > >
> > > Thank you again for your thoughtful feedback, and for recognizing the technical soundness and significance of our work. We appreciate your constructive suggestions and are glad that our revisions addressed your concerns. We will incorporate all improvements into the final version.

---

### Decision · Program_Chairs · 2026-04-30

**Decision:**

Accept (regular)

**Comment:**

This paper studies zeroth-order Langevin Monte Carlo sampling fro non-log-concave density.   The authors first propose a variance reduced gradient estimate that leverages the history of gradient approximations, and use it for posterior sampling with diffusion models.  As one of the main contributions, theoretical convergence guarantee is provided for non-log-concave density.  In experiments,  the performance of the proposed method is demonstrated in many inverse problems, where the proposed method performs significantly better than the gradient-free baselines, and comparably to the gradient-based baselines.

The reviewers acknowledged significant contributions, and the authors addressed their concerns.